# Selective pressures of platinum compounds shape the evolution of therapy-related myeloid neoplasms

Eline J. M. Bertrums [1,2,3,10], Jurrian K. de Kanter [1,2,10], Lucca L. M. Derks [1,2], Mark Verheul[1,2], Laurianne Trabut[1,2], Markus J. van Roosmalen [1,2], Henrik Hasle [4], Evangelia Antoniou[5,6], Dirk Reinhardt [5,6], Michael N. Dworzak[7,8], Nora Mühlegger[7], Marry M. van den Heuvel-Eibrink [1,9], C. Michel Zwaan[1,3], Bianca F. Goemans [1] & Ruben van Boxtel [1,2] ✉

Therapy-related myeloid neoplasms (t-MN) arise as a complication of chemo- and/or radiotherapy. Although t-MN can occur both in adult and childhood cancer survivors, the mechanisms driving therapy-related leukemogenesis likely vary across different ages. Chemotherapy is thought to induce driver mutations in children, whereas in adults pre-existing mutant clones are selected by the exposure. However, selective pressures induced by chemotherapy early in life are less well studied. Here, we use single-cell whole genome sequencing and phylogenetic inference to show that the founding cell of t-MN in children starts expanding after cessation of platinum exposure. In patients with Li-Fraumeni syndrome, characterized by a germline *TP53* mutation, we find that the t-MN already expands during treatment, suggesting that platinum-induced growth inhibition is *TP53*-dependent. Our results demonstrate that germline aberrations can interact with treatment exposures in inducing t-MN, which is important for the development of more targeted, patient-specific treatment regimens and follow-up.

Most chemotherapies act by fatally damaging or inhibiting the synthesis of DNA of cancer cells[1]. However, normal cells are also exposed during treatment, which can promote new carcinogenesis. Therapy-related myeloid neoplasms (t-MN), which include myelodysplastic syndrome (MDS) and acute myeloid leukemia (AML), are hematological disorders that typically occur within 10 years after treatment with cytotoxic therapy for a primary cancer or autoimmune disease[2–4]. Patients with t-MN have a poor prognosis compared to treatment-naïve AML or MDS[5,6], urging the need to develop preventive strategies. t-MN can occur at all ages but remains understudied in children.

Nevertheless, it is one of the most prevalent subsequent neoplasms after childhood cancer treatment, besides radiotherapy-induced breast cancers, mostly occurring in females[4]. Previous research that investigated the effects of cytostatic treatment in the hematopoietic system of cancer patients focused on therapy-related clonal hematopoiesis (t-CH) and t-MN in adults[7–11]. Together, these studies propose a model in which chemotherapy exposure induces leukemogenesis mainly by selecting mutated clones that in most cases predated the treatment exposure[7,10,12], and only partly by induction of mutations[12]. Indeed, t-MN driver mutations can already be observed in bone

[1]Princess Máxima Centrum for pediatric oncology, Utrecht, the Netherlands. [2]Oncode Institute, Utrecht, the Netherlands. [3]Department of Pediatric Oncology/Hematology, Erasmus Medical Center – Sophia Children's Hospital, Rotterdam, the Netherlands. [4]Department of Pediatrics, Aarhus University Hospital, Aarhus, Denmark. [5]Clinic of Pediatrics III, University Hospital of Essen, Essen, Germany. [6]AML-BFM Study Group, Essen, Germany. [7]St. Anna Children's Cancer Research Institute, Vienna, Austria. [8]St. Anna Children's Hospital, Department of Pediatrics and Adolescent Medicine, Medical University of Vienna, Vienna, Austria. [9]Utrecht University, Utrecht, the Netherlands. [10]These authors contributed equally: Eline J. M. Bertrums, Jurrian K. de Kanter. ✉e-mail: R.vanBoxtel@prinsesmaximacentrum.nl

marrow or peripheral blood of adult patients isolated before exposure to cytotoxic therapy[7]. Also, in retrospective studies, specific genes were found to drive t-CH depending on the preceding treatment exposures and which also varied between distinct exposures[10,13]. For example, mutations in DNA damage response (DDR) genes, such as *TP53* and *CHEK2*, were significantly enriched in clones of adults specifically after exposure to cytotoxic therapies, such as platinum drugs and topoisomerase II inhibitors (TOP2i)[10].

Besides selective pressure, DNA damage induced by chemotherapy in healthy cells can increase the genetic diversity in normal tissues during exposure. This can contribute to the development of t-MN by increasing the chance that healthy cells acquire a cancer driving event[14–16]. In contrast to adult t-MN, studies suggest that this mechanism occurs often in pediatric t-MN patients, as t-MN driver mutations could not be detected by ultra-deep sequencing methods in samples predating the start of treatment[16,17]. In addition, mutational signature analysis in pediatric t-MN has shown that some driver mutations are directly induced by cytotoxic treatments, such as platinum drugs and thiopurines[14,16,18]. Furthermore, exposure to topoisomerase II inhibitors (TOP2i) is associated with *KMT2A* rearrangements with breakpoints in TOP2-binding regions[19,20]. This association indicates that these rearrangements, that are often found in pediatric t-MN, are likely directly induced by TOP2i. Yet, the evolutionary pressures induced by exposure to these agents and subsequent selection of malignant clones in the blood of children remain unclear.

Here, we present a model explaining pediatric t-MN development and we subsequently compare this model to the etiology of t-MN in adults. Therefore, we analyze the genomes of t-MN in 44 children. Using single-cell whole genome sequencing (WGS), we show that although chemotherapy exposure induces the genetic aberrations that drive leukemogenesis, the expansion of the t-MN clone is inhibited by platinum drugs. Subsequently, after finalization of treatment with platinum drugs, the leukemic cell of origin rapidly expands. In Li-Fraumeni Syndrome (LFS) patients, *TP53*-deficient leukemic clones can expand during platinum drugs exposure, suggesting that platinum-induced growth inhibition is TP53-dependent. Indeed, in LFS patients the t-MN shows a developmental trajectory that is more like that observed in adults.

## Results

### Pediatric t-MN patient cohort

We included 18 Dutch (Cancer Discovery 2022)[14], one Austrian, and 25 German t-MN patient samples that were obtained via a collaboration with the International Berlin-Frankfurt-Münster AML Study Group (I-BFM AML SG). The patients had a variety of first diagnoses and were exposed to different treatment regimens (Fig. 1a, Supplementary Table 1). Most of these children had a primary cancer diagnosis, but four patients received treatment for other underlying diseases (Supplementary Table 1). Except for two patients, all children received chemotherapy (Fig. 1a). Although patients UPN009 and IBFM29 were included in this cohort based on clinical diagnosis, their t-MN cannot be specified as chemotherapy-induced, as they received only local radiation or immunosuppressants. Due to the inclusion criteria of our study, 43 patients presented with t-AML and only one with t-MDS. The mean age at t-MN diagnosis was 10.9 years (range 2.4–18.4 years). The latency time between first diagnosis and t-MN varied from 0.5 – 11 years (mean 3.6 [95% CI 2.7-4.4], Fig. 1b, c). Patients with a hematological malignancy as a first diagnosis had a longer latency time compared to patients with any other primary diagnosis ($p = 0.013$, Supplementary Fig. 1a). Unfortunately, due to the variety of treatment regimens included in our cohort, this difference in latency time could not be linked to a specific chemotherapeutic compound. We performed WGS on bulk t-MN blasts and used mesenchymal stromal cells

(MSCs) or bulk-sorted B-cells of the same patient as a germline control (Methods, Supplementary Fig. 2, 3).

In general, the driver events observed in our cohort were in line with previous studies[14,16] (Fig. 1d, e, Supplementary Fig. 1b). Most recurrent genetic aberrations in the pediatric t-MN samples were structural variants (SVs). Fusions were found in *KMT2A* (59%), *RUNX1* (7%), *MLLT10* (5%) and *MECOM* (2%) (Fig. 1d, Supplementary Data 1). Breakpoints of 9 out of 26 *KMT2A* fusions (35%) overlapped with the 11 bp topoisomerase-associated breakpoint hotspot (Supplementary Fig. 1c)[19,20]. The t-MN with oncogenic fusions usually had fewer chromosomal aberrations than t-MN without fusions (0.4 vs 4.2, $p = 1.6*10^{-5}$) except for *RUNX1* alterations which co-occurred with chromosome 21 gains in four patients. This co-occurrence has also been described in treatment-naïve AML[21]. Recurrent losses were observed in chromosome 6, 7, and 11 (in 9%, 9% and 7% of the patients, respectively; Fig. 1f). Compared to adult t-MN, our cohort had a paucity of *TP53* aberrations. A somatic *TP53* mutation was only found in one t-MN (2%), compared to 33% of adult t-MN[7]. Also in de novo pediatric AML, *TP53* aberrations are rare[22].

In the German I-BFM patients, we could investigate germline predisposition variants for which we used a previously published set of childhood cancer-associated genes[23]. Likely pathogenic germline mutations were found in 6/25 t-MN patients (24%), which is higher than previously published in pediatric cancer and pediatric t-MN[16,24], but comparable to adult t-MN[25]. We found a germline aberration in five patients in the genes encoding *BLM, WRN*, and *WT1* in one patient each, and *FANCA* in two patients. Patient IBFM22 had a germline aberration in both *TP53* and *NF1*, indicating that this patient had two tumor predisposition syndromes: LFS and neurofibromatosis type 1 (NF1). In the t-MN blasts, the wild-type allele of *TP53* was lost and the *NF1* mutation was duplicated. In the Dutch cohort and Austrian patient sample we were not able to search for predisposition genes due to ethical constrains, yet the only known germline aberration, based on diagnostic data, was a mutation in *CHEK2*.

### Mutational processes underlying t-MN development

A previously established baseline of mutation accumulation during healthy life showed that hematopoietic stem and progenitor cells (HSPCs) normally acquire mutations at a constant rate of 14 to 16 single base substitutions and approximately one insertion or deletion (indel) per year[26,27]. We compared the mutation load of the t-MN blasts to this baseline and found a significant increase (1005 additional mutations, $p < 10^{11}$; Supplementary Fig. 4a), similarly to what was previously reported[14].

To elucidate the mutational processes underlying the additional mutations after treatment exposure, we extracted and refitted mutational signatures (Fig. 2a, Methods). In 13 out of 44 t-MN cases (30%), all mutations could be exclusively explained by clock-like signatures SBS5 and HSPC, while in the other patients at least 150 mutations could be attributed to an additional signature (Fig. 2a–c). We identified the platinum-associated signature SBS31 in platinum-exposed patients[28], and the thiopurine-associated signature SBS87 in thiopurine-exposed patients[29]. In addition, we identified SBS17a/b in the t-MN of one patient (IBFM9) in our cohort, who developed the t-MN after a primary acute lymphoblastic leukemia (ALL). SBS17a/b mutations have previously been attributed to exposure to the drug 5-FU as well as misincorporation of oxidized guanines opposite a thymine in the DNA template during replication[30]. Since 5-FU is not used in ALL treatment protocols, and this patient did not have a treatment history of 5-FU, the latter process seems more likely.

Furthermore, we identified a signature, here named SBSD, which resembles a carboplatin-associated signature previously described by Pich et al. in metastases of adult solid tumors[31] (sbs25, cosine similarity 0.89, Fig. 2d). The 96-trinucleotide profile has similarities to SBS31 and

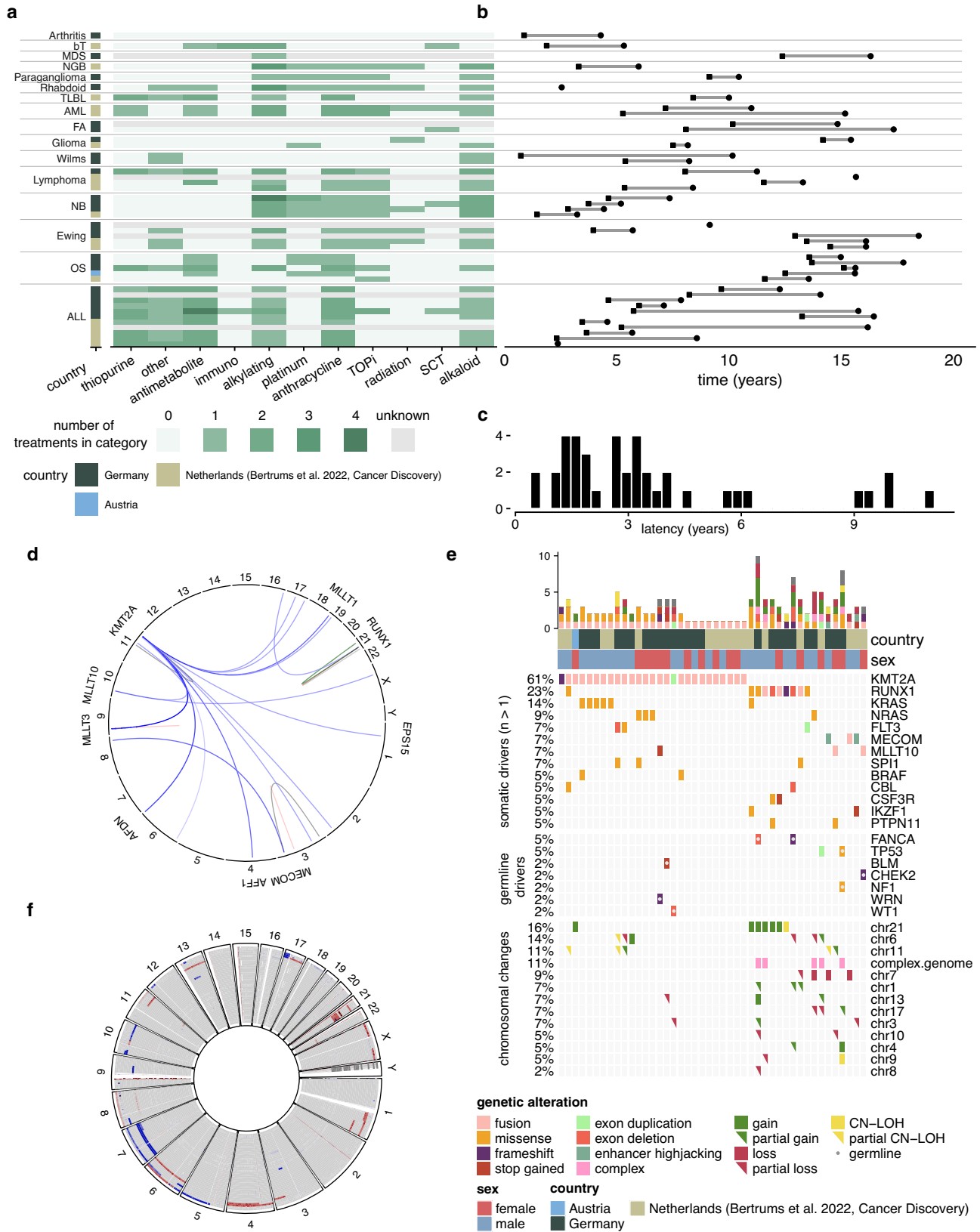

SBS35, which are both platinum-induced signatures. SBS31 and SBSD both co-occur with a platinum-induced double base substitution signature DBS5 in our dataset (Supplementary Fig. 4b-d). SBSD was mainly present in four patients (IBFM22, IBFM38, IBFM14 and IBFM28). Interestingly, IBFM22, who was treated with carboplatin for NF1-related opticus glioma, also harbored a germline *TP53* mutation,

characteristic of LFS. TP53 is essential in the G1/S cell cycle checkpoint and is activated upon a variety of cellular stresses, including DNA damage[32]. *TP53* is the most commonly mutated gene in human cancer[33] and mutations have been associated with platinum-resistance[34], potentially explaining the distinct signature. Interestingly, the t-MN of IBFM14 had a heterozygous loss of chromosome 17p that harbors the

**Fig. 1 | Pediatric t-MN (n = 44) is mainly driven by *KMT2A* fusions. a** A table depicting the different first diagnoses of t-MN patients and the treatment categories that each patient received. ALL: acute lymphoblastic leukemia; AML: acute myeloid leukemia; bT: beta-thalassemia; FA: Fanconi anemia; MDS: myelodysplastic syndrome; NB: neuroblastoma; NGB: neuroganglioblastoma; OS: osteosarcoma; SCT: allogenic stem cell transplantation; TLBL: T-cell lymphoblastic lymphoma; TOPi: topoisomerase inhibitors. The country in which the sample was collected is indicated in the left side of the table. **b** Per-patient timelines depicting the latency time between the first diagnosis and the t-MN diagnosis. Rows per patient match with (**a**). **c** Distribution of latency times in years. **d** Circos plot of the structural variants (n = 72) in t-MN patients that involved at least one cancer gene. **e** Oncoprint depicting the clonal driver events that were present in t-MN samples. The bar plots on top represent the number of driving events present in each sample. Small drivers are only included if they occurred in more than one patient. CN-LOH: copy neutral loss of heterozygosity. **f** Circos plot depicting the copy number profiles of all t-MN samples. *Source data are provided as a Source Data file.*

*TP53* gene and IBFM38 had a germline *WT1* mutation, which has been described to impact downstream factors of *TP53*[35], and thus likely increases resistance to DNA damaging agents. Like IBFM22, IBFM28 (atypical rhabdoid tumor) was treated with carboplatin. Carboplatin treatment was also highly likely for IBFM38 (Wilms tumor), according to the applicable treatment protocol at the time the patient was treated, whereas this was unclear for IBFM14 (Ewing sarcoma). Notably, also patient IBFM21 (*TP53*[+/+]) received carboplatin therapy, and this patient did not show SBSD-related mutations. These findings suggest that the type of carboplatin-induced mutations might be influenced by *TP53* function.

To further validate this, we compared *TP53*[+/+], *TP53*[+/-] and *TP53*[-/-] metastases of cancers that had been treated with carboplatin or cisplatin from a cohort of 4853 metastases from 4,711 patients previously described by Priestley et al.[36]. We checked the contribution of the SBS31, SBS35 and the carboplatin-associated sbs25 that was found by Pich et al. and is similar to SBSD in this cohort. After carboplatin exposure, fewer *TP53*[-/-] and *TP53*[+/-] tumors contained SBS35 mutations compared to *TP53*[+/+] tumors (22%, 44% and 55% respectively, not significant). In contrast, more *TP53*[-/-] and *TP53*[+/-] tumors had sbs25 (SBSD) mutations than *TP53*[+/+] tumors (55%, 38% and 28%, respectively). After cisplatin exposure SBS31 was the dominant signature, independent of *TP53* status (Supplementary Fig. 5). Although not confirmative, these results are in line with the idea that *TP53* status influences the type of mutations that accumulate during carboplatin exposure.

Finally, we identified three thus far unknown signatures with unique 96-trinucleotide profiles, which we here named SBSE, SBSF and SBSG (Fig. 2d). Interestingly, SBSG was identified in three t-MN that developed after a Ewing sarcoma, suggesting a potential association with the treatment regimen. As SBSE and SBSF only occurred in the t-MN of one patient each (IBFM28 and IBFM21, respectively), it remains challenging to elucidate the underlying process. While the profile of SBSE shows mainly C > G changes, in SBSF mainly T > C changes are seen, which is in line with exposure to alkylating agents[37].

We used a previously established method to calculate the probability that the t-MN driving mutations were caused by each detected signature[38,39]. This analysis revealed that clock-like signatures could explain 29% of the single nucleotide variant drivers. On the other hand, 45% of the driver mutations could be attributed to a non-clock-like signature with > 70% likelihood (Fig. 2e), showing that also in our cohort treatments did induce small drivers.

### Clonal evolution of t-MN under platinum exposure

To study the selective pressure of platinum compounds in the blood of t-MN patients, we performed retrospective lineage tracing in multiple patients. WGS was performed on DNA directly amplified from single cells using primary template-directed amplification (PTA; referred to as "single cells"). Or, when possible, WGS was also performed for clonally expanded HSPCs (referred to as "HSPC clones"; Supplementary Data 2 and 3)[40]. By comparing the mutations shared between all sequenced cells and clones from the same patient, we could construct phylogenetic trees in which each split of a branch represents a cell division (Fig. 3). These trees can be used to study the evolution of the t-MN over time, by assessing the moment of the t-MN expansion and the mutational processes, which were active before, during and after

this expansion. *Clonal* mutations were shared between all the t-MN blasts and accumulated before expansion of the initial leukemic cell, *subclonal* mutations were shared between a subset of single t-MN blasts and accumulated during expansion of the leukemic clone, and *private* mutations were unique to single blasts and accumulated most recently during t-MN expansion. The HSPCs were used to check if the same mutational processes were active in normal cells. We validated the shape of the tree, the *clonal*, *subclonal*, and *private* labels for branches, and the most common recent ancestor of the bulk t-MN by investigating the variant allele frequencies (VAF) of the tree mutations in the matching bulk t-MN data, and by investigating mutations present in the bulk t-MN, but not present in the tree (Supplementary Fig. 6). Of note, HSPCs and t-MN cells were sorted and labeled in the manuscript based on their immunophenotype. In four patients some of the immunophenotypically HSPC-like cells shared all the t-MN drivers and other clonal mutations with the bulk t-MN blasts and were thus in the phylogenetic analysis considered t-MN blasts.

We ran our updated pipeline (Methods) on previously published data of UPN008[14], a patient with a t-MN after platinum-treatment (n = 12 HSPCs). The bulk t-MN blasts of this patient harbored many *clonal* platinum-related mutations, only few platinum-related *subclonal* mutations (17), and no platinum-related *private* mutations (Fig. 3a, b, Supplementary Fig. 7a). This observation suggested that the t-MN clone started expanding after the end of platinum treatment. The few *subclonal* platinum-related mutations were likely acquired due to unrepaired platinum-induced lesions being passed on to daughter cells in the first division after treatment[41]. The fact that the *subclonal* branches consisted of very few mutations suggests that the cell divisions happened in a very short timeframe. The phylogenies of patients IBFM32 (n = 3 single HSPCs, n = 3 blasts), IBFM42 (n = 2 single HSPCs, n = 1 HSPC clone, n = 4 single blasts), and IBFM67 (n = 2 single HSPCs, n = 5 single blasts), all treated with cisplatin, showed a similar pattern. For these patients, besides HSPCs, we also included single blasts in our analysis. Apart from a single division detected in the middle of the t-MN evolution of IBFM32, all three patients showed long *clonal* branches and short *subclonal* branches (Fig. 3c–h, Supplementary Fig. 7b–d). Of note, in IBFM32, a *KRAS* pG12V mutation was present in the bulk t-MN at high VAF, that was not present in the tree, indicating that a t-MN clone went unsampled (Supplementary Fig. 6c). Similar to patient UPN008, the expansion of the t-MN clones in all three patients happened after the end of platinum treatment. Indeed, most *clonal*, but only few *subclonal* and *private* mutations were platinum-related. Single t-MN blasts of IBFM67 harbored C > A mutations in a profile that was similar to SBS29 and SBS18. These mutations have an unknown cause, but a possible explanation could be in vitro oxidative stress. These results support the idea that platinum exposure inhibits the expansion of the initial leukemic clone, and that this expansion thus starts when the exposure to platinum ends.

### The role of *TP53* in clonal evolution under the selective pressure of platinum exposure

As discussed above, we found that *TP53*-deficient cells that were exposed to carboplatin accumulated a different mutational signature

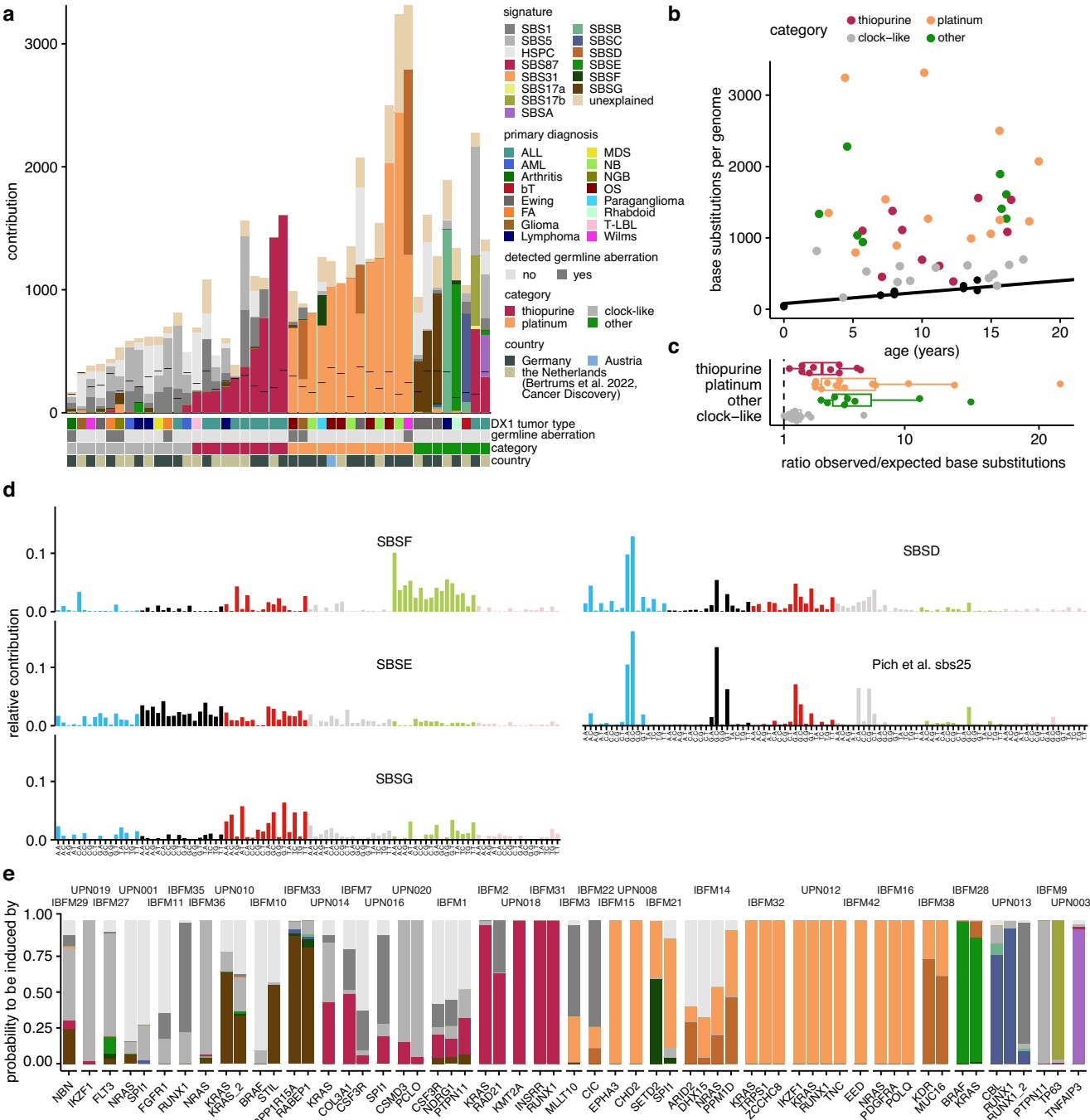

**Fig. 2 | Mutational processes underlying the increased mutation load in pediatric t-MN (*n* = 44). a** The contribution of each single base substitution signature to t-MN blasts of each patient, obtained after bootstrapped (*n* = 100) refitting of signatures that were extracted by non-negative matrix factorization. The first bar below the plot represents the first diagnosis (abbreviations conform Fig. 1a), the second bar notes if a pathogenic germline mutation was found, the third bar represents the treatment category (>150 mutations of that treatment type, or otherwise "clock-like") and the last bar indicates the country in which the sample was collected. **b** Mutation accumulation of t-MN (colored dots) compared to the baseline of healthy blood cells (black dots). The color is similar to the grouping in

(**a**). **c** The ratio of the number of observed versus expected single base substitutions in all t-MN samples within a specific signature-category (as in (**b**)). *N* = 44. Here and in all other figures, the box plots depict the median (center line), 25th and 75th percentiles (box), and the largest values, no more than 1.5 * the interquartile range (whiskers). **d** The 96-trinucleotide single base substitution profiles of SBSD-G and the profile of the previously defined signature sbs25 (Pich et al.[31]). **e** The probability that different driver mutations (*n* = 59) were caused by treatment-related or clock-like signatures. Colors are similar to signatures in (**a**). *Source data are provided as a Source Data file.*

(SBSD) compared to *TP53*-proficient cells. In addition, previous literature indicates a clonal advantage of *TP53*-mutated cells under platinum treatment[34]. Therefore, we investigated the clonal evolution of t-MN in the carboplatin-treated LFS patient IBFM22 and compared this to *TP53* wild-type platinum-treated t-MN patients.

In the phylogeny of patient IBFM22 (*n* = 5 single HSPCs, *n* = 5 single blasts), fewer *clonal* mutations were present. In addition, only few of these trunk mutations could be attributed to platinum treatment, indicating that the t-MN clone started expanding very early during treatment (Fig. 4a, b, Supplementary Fig. 7e). In contrast to the

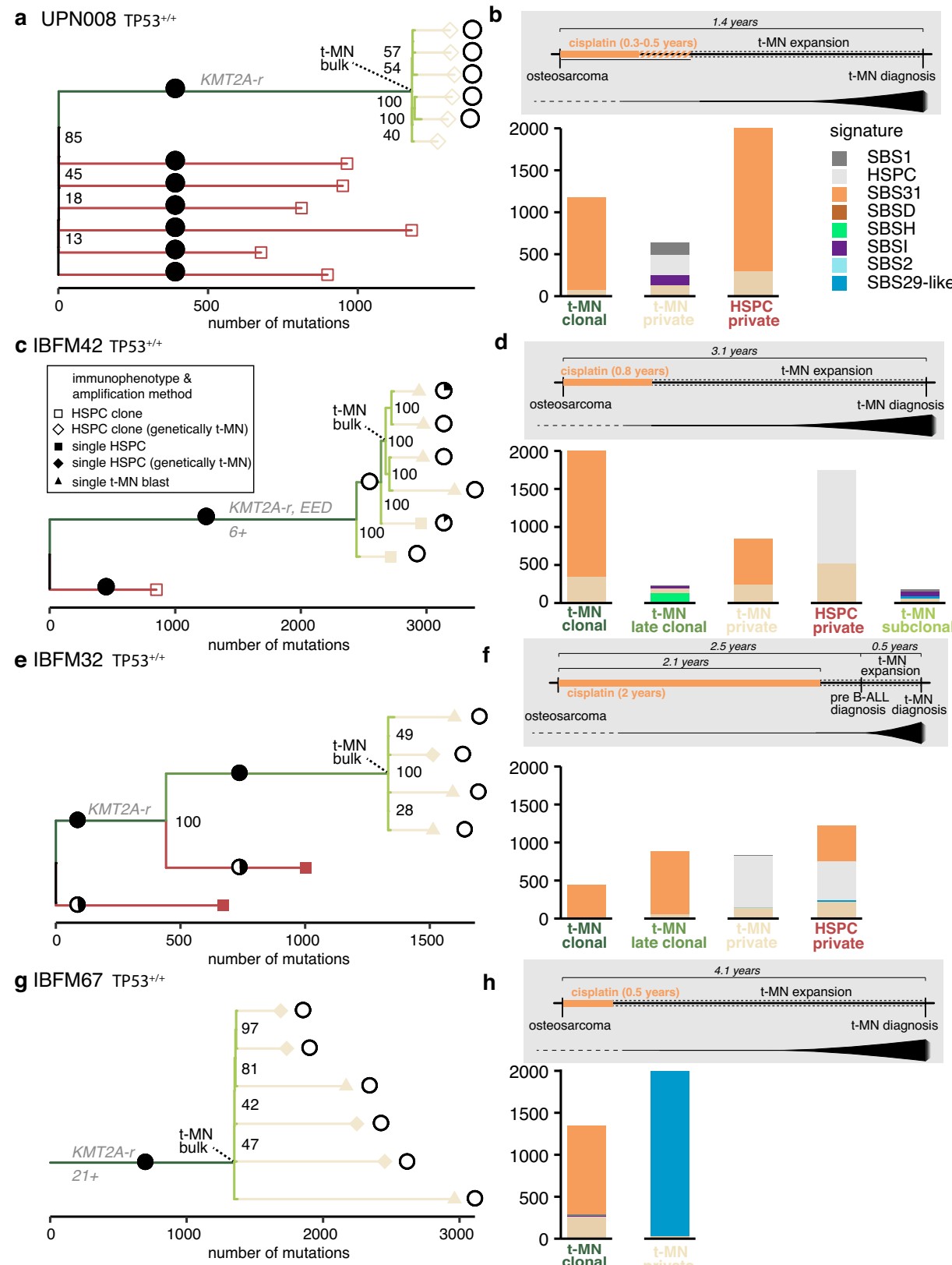

TP53$^{+/+}$ t-MN cases, the *subclonal* branches contained hundreds of mutations. Furthermore, we could observe two branching points that occurred during platinum exposure, as *subclonal* and *private* branches before and after these branching points harbored platinum-related mutations, indicating that the expansion of the t-MN in this case

occurred during treatment. Interestingly, in the private t-MN mutations, another signature (SBSH) is detected, which has a high contribution of T > C mutations. The presence of T > C mutations could be in line with exposure to alkylating agents[37] (Supplementary Fig. 6i). This model was further strengthened by the short latency of 0.7 years

**Fig. 3 | Clonal evolution of t-MN under platinum treatment in patients without germline *TP53* aberrations. a** Phylogenetic tree of clonally expanded HSPCs and bulk t-MN blasts of patient UPN008 (*TP53* wild-type). Pie charts indicate the contribution of SBS31 after strict refitting (max_delta < 0.01). The colors of the branches correspond to the type of mutation, *clonal, subclonal,* or *private,* which is annotated in the same color text. Small black numbers at splits in the trees indicate in what number of CellPhy bootstraps the split was found, out of 100. **b** Signature contribution of the mutations in the corresponding branches in the lineage tree on the left. The *private* HSPC mutations were subsampled to 2000 mutations for visual purposes. Top: schematic overview of the timeline of the different diagnoses

and treatment, including the timing of t-MN development. **c** Similar to (**a**), but for patient IBFM42 (*TP53* wild-type). Also single-cell sequenced HSPCs (black squares) and t-MN blasts (black triangles) are included. SBS31 contribution to the clonal branch was supported by 100/100 bootstraps. In the private branch, SBS31 was found in 40/100 bootstraps. **d** Similar to (**b**), but for patient IBFM42. **e** similar to (**a**), but for patient IBFM32 (*TP53* wild-type). **f** Similar to (**b**), but for patient IBFM32. **g** Similar to (**a**), but for patient IBFM67 (*TP53* wild-type). Single-cell t-MN blasts and single HSPCs were sequenced. All sequenced HSPCs shared the *KMT2A* rearrangement with the t-MN blasts. **h** Similar to (**b**), but for patient IBFM67. *Source data are provided as a Source Data file.*

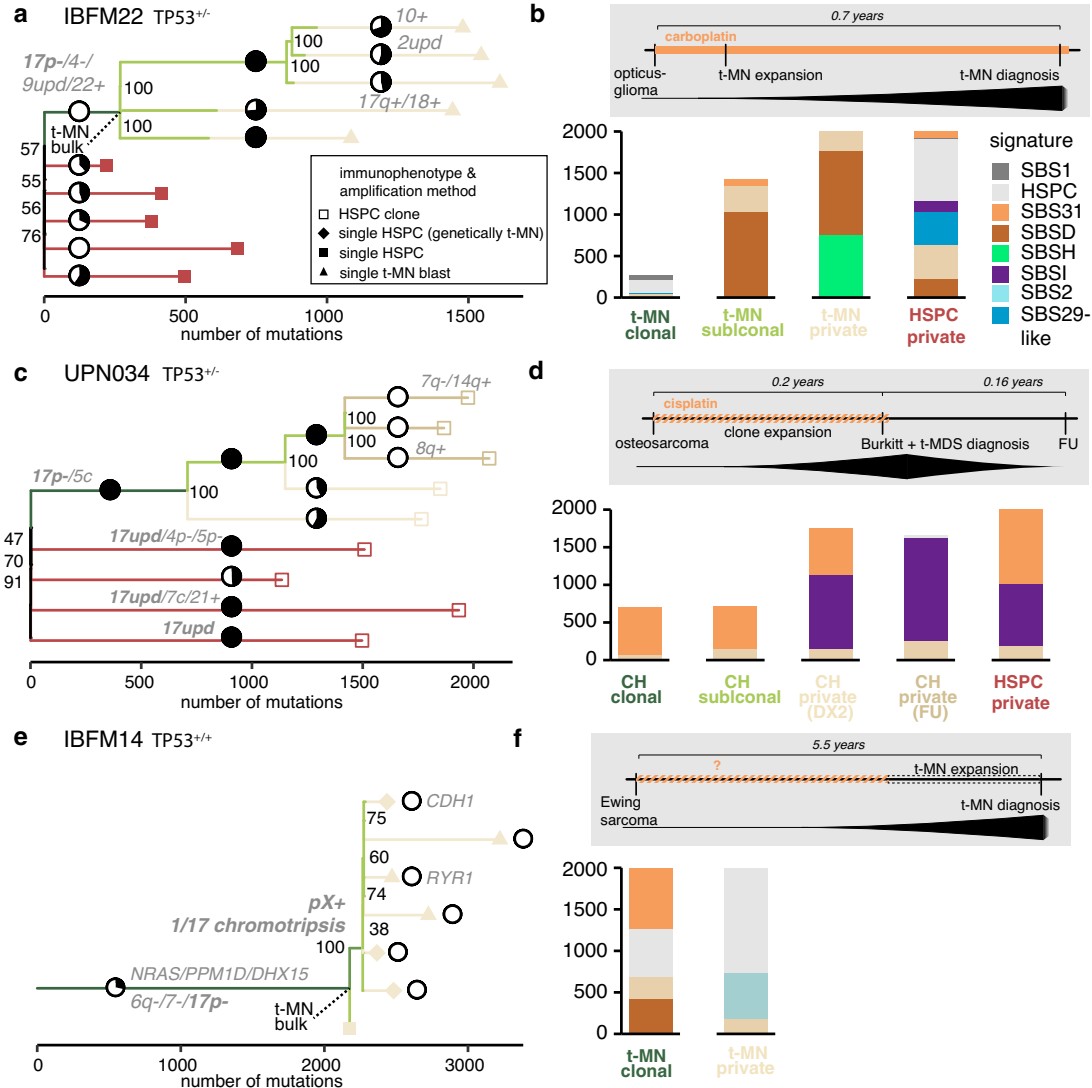

**Fig. 4 | Clonal evolution of t-MN under platinum treatment in patients with germline *TP53* aberrations. a** Phylogenetic tree of single HSPCs (black squares) and bulk and single (black triangles) t-MN blasts of patient IBFM22 who had LFS and a *TP53*[-/-] t-MN. Pie charts indicate the contribution of SBSD after strict refitting (max_delta < 0.01). Small black numbers at splits in the trees indicate in what number of CellPhy bootstraps the split was found, out of 100. UPD: uniparental disomy **b** Signature contribution of the mutations in the corresponding branches in the lineage tree on the left. Top: schematic overview of the timeline of the different diagnoses and treatment, including the timing of t-MN development. **c** Similar to

(**a**), but for patient UPN034 who had LFS. Clonally expanded HSPCs (white squares) were sequenced. Pie charts indicate the contribution of SBS31 after strict refitting (max_delta < 0.01) (**d**) Similar to (**b**), but for patient UPN034. The HSPC *private* mutations were subsampled to 2000 mutations for visual purposes. CH: clonal hematopoiesis, FU: follow-up. In the schematic, the evolution of the t-MN, not the CH, is drawn. 0% blasts were detected in the FU sample by diagnostic MRD measurements. **e** Similar to (**a**), but for patient IBFM14 who had a *TP53*[+/-] t-MN. Pie charts indicate the contribution of SBS31 after strict refitting (max_delta < 0.01) **f** Similar to (**b**), but for patient IBFM14. *Source data are provided as a Source Data file.*

between first diagnosis and t-MN development in patient IBFM22, which was still within the carboplatin treatment window. This latency time was much longer for patients UPN008 (1.4 years), IBFM32 (2.5 years), and IBFM42 (3.1 years). In conclusion, in *TP53*[-/-] t-MN the

timing and speed of the clonal expansion appeared to be different compared to *TP53*[+/+] t-MN.

To confirm the proposed interaction between loss of *TP53* and the evolutionary pressures that are induced by platinum treatment, we

studied the post-treatment bone marrow sample of another LFS patient (UPN034, $n = 9$ HSPC clones), who had been previously treated for osteosarcoma (DX1) with, among other treatments, cisplatin. Less than three months later, during the treatment for osteosarcoma, this patient was diagnosed simultaneously with Burkitt lymphoma and t-MDS with < 5% leukemic blasts in the bone marrow (DX2). We studied nine clonally expanded HSPCs, both from time of DX2 ($n = 6$) and follow-up two months later (FU; $n = 3$). Interestingly, five HSPC clones, including all three FU clones, shared the same complex event involving chromosome 5 and 17 that resulted in the loss of the wild-type *TP53* allele (Supplementary Fig. 8a–d). This is suggestive for the presence of CH at time of DX2 in which one HSPC expanded at a disproportionately higher rate than other HSPCs[42]. Comparable to IBFM22, the *subclonal* branches were hundreds of mutations long and were platinum-related (SBS31), suggesting that also this clone expanded during treatment (Fig. 4c, d, Supplementary Fig. 7f). Similar to the single t-MN blasts of IBFM22, HSPC clones of UPN034 had a high number of T > C mutations, indicating that these mutations are induced by alkylating drugs (signature SBSI, Supplementary Fig. 6i). Interestingly, all cells at time of FU arose from the same clone and the *private* mutations in these cells occurred after the stop of platinum treatment.

Next, we evaluated if the HSPC clones of UPN034 were related to the t-MN. As we were not able to sort and thus perform WGS of the t-MN blasts, we compared the single base substitutions and copy number variants (CNVs) found in the HSPC WGS data with diagnostic data from single nucleotide polymorphism (SNP) and karyotype assays of the t-MN. One clone in the phylogenetic tree that we constructed shared a *RUNX1* aberration and 7q loss with the t-MN, but not the additional 20q loss. These findings imply that this HSPC clone is a pre-leukemic cell. Notably, in total we identified four independent events, with unique breakpoints, by which individual HSPC clones in this patient lost their *TP53* wild-type allele (Supplementary Fig. 8 c-e). This convergent evolution of multiple clones that independently lost the wild-type allele of *TP53* indicates a strong selective pressure to lose TP53 function in this patient[43]. To investigate whether this selective pressure was due to the treatment or already existed before, we investigated WGS data of bulk peripheral blood at the time of primary osteosarcoma diagnosis. CNV analysis of these data revealed that a variety of 17p loss events were already present in the blood before the start of any treatment (at time of DX1). This resulted in the loss of a copy of *TP53* in 38% of the blood cells, indicating that *TP53* wild-type allele loss events were already present before treatment (Supplementary Fig. 8a, b).

Subsequently, we investigated if a clone could also escape platinum-induced inhibition when losing only one allele of *TP53*. Therefore, we performed WGS on single HSPCs ($n = 4$) and t-MN blasts ($n = 3$) of patient IBFM14, whose t-MN had a heterozygous loss of chromosome 17p, involving *TP53*, and no additional *TP53* mutation. Although the patient history is unclear about platinum treatment, mutational signature analysis of the t-MN revealed contribution of both SBS31 and SBSD, indicative of carboplatin exposure. Notably, the *private* mutations of the single t-MN blasts had contribution of SBS2, indicative of APOBEC, which is an uncommon signature in pediatric cancer, including AML[44,45]. Other than that, the t-MN in this patient followed the same pattern as the three *TP53*[+/+] t-MN, with many platinum-related *clonal* mutations, but no platinum-related *private* mutations, indicative of expansion after the end of platinum treatment (Fig. 4e, f, Supplementary Fig. 7g). This confirms that only t-MN clones that have no *TP53* wild-type allele left can escape platinum-induced inhibition. In line with this, 13 out of 15 pediatric t-MN described by Schwartz et al. that harbored a *TP53* alteration, had both a mutation and a copy number loss of this gene[16].

Finally, we validated the effect of *TP53* deficiency on cell proliferation under platinum treatment using *MV4-11*, a pediatric acute monocytic leukemia cell line with a subclonal *TP53* R248W mutation.

We expanded three clones with a *TP53* wildtype allele (*MV4-11*[WT]) and three clones with a homozygous R248W allele (*MV4-11*[R248W])[46,47]. In line with our in vivo findings, *MV4-11*[R248W] was more resistant to carboplatin treatment than *MV4-11*[WT] (IC50 14.3 μM vs. 3.5 μM, $p = 1.6 \times 10^{-27}$, Fig. 5a, b, Supplementary Fig. 9a,b). Next, we used a single pulse of CellTrace™ dye, which is equally distributed over daughter cells during cell divisions, to track proliferation during in vitro treatment. From the fluorescent intensity after 4 days of treatment, we calculated a Pro-liferation Score (Methods). *MV4-11*[R248W] showed a dose-dependent increase in proliferation compared to *MV4-11*[WT], confirming that *TP53*-deficiency leads to platinum resistance by enhancing proliferation during treatment (Fig. 5c, d). To further verify these results, we used CRISPR/Cas9 to induce TP53 knock-out insertions and deletions in CD34+ umbilical cord blood (UCB) cells, which have a minimal mutational background. This resulted in bulk CD34 + UCB populations with an average *TP53* KO score of 38% (UCB[TP53KO], CI 95%: 20%-55%, Supplementary Fig. 9c,d). We treated wildtype UCB cells (UCB[WT]) and UCB[TP53KO] with 22.5 μM carboplatin, the approximate IC50 at day 4 in UCB[WT]. After 4 and 8 days of carboplatin treatment, the viability of UCB[TP53KO] was higher than UCB[WT], although not significantly (Fig. 5e). As the UCB[TP53KO] is a heterogeneous population, we next compared the frequency of KO-inducing deletions at the targeted *TP53* locus. We found that the fraction of TP53 KO-inducing deletions increased during carboplatin treatment compared to untreated UCB[TP53KO] cells, indicating that *TP53* KO cells proliferated more under treatment ($p < 0.0182$, Fig. 5f). Finally, to test the influence of carboplatin on the genome of UCB cells, we sequenced UCB[WT] cells, both untreated, and treated with carboplatin for four days. Carboplatin induced 317 mutations (CI 95%: 67-568, Fig. 5g). Interestingly, the mutational profile of the carboplatin treatment was more similar to SBSD than SBS35 or SBS31 (cosine similarity of 0.82, 0.77, and 0.68 respectively). Possibly, this is caused by the high division rate that is induced by the medium and growth factors that is used for culturing the UCB cells.

## Discussion

Previous studies have described differences in the evolution of t-MN in children and adults[7,14,16]. Whereas in adults the founding cell is often already present before treatment, and subsequently selected during chemotherapy, in children the driving event of the t-MN is likely induced during therapy, and the t-MN blasts probably expand afterwards[7,14]. Indeed, in nine out of eleven t-MN bulk samples of our cohort that were treated with platinum drugs, more than 500 (range 655-2652) platinum-induced *clonal* mutations were detected. This observation indicates that the most common recent ancestor (MCRA) of the t-MN originated at the end or after treatment, as only then the treatment-induced mutations become *clonal*, as previously explained by Landau et al. and Pich et al.[31,48]. Interestingly, we found that all *clonal* mutations in UPN008 and UPN034 were platinum-related, and none were age-related. This is in line with previous research showing that there is no link between the biological age and the number of age-related mutations in patients that are treated with platinum, whereas this link is present in cells of healthy individuals[9]. In addition, we show that the clonal expansion of the t-MN blasts is inhibited by platinum-based treatments. The evolution of *TP53*[+/+] t-MN shows rapid expansion of a single clone, likely in a short period of time, after overcoming a selective pressure, in this case by cessation of platinum-based treatment. On the other hand, pediatric *TP53*[-/-] t-MN are able to expand during platinum exposure and therefore have an evolutionary trajectory that is more similar to those described in adult t-MN[7] and different from pediatric *TP53*[+/+] t-MN. As platinum drugs are usually administered in short intervals during the treatment protocol, it is unclear whether *TP53*[-/-] cells expand in vivo during exposure itself, similar to the MV4-11 cells in vitro, or whether they recover faster and expand during treatment intervals. Overall, more *TP53*[-/-] t-MN cells are able to survive during treatment. The observations that a complete loss of

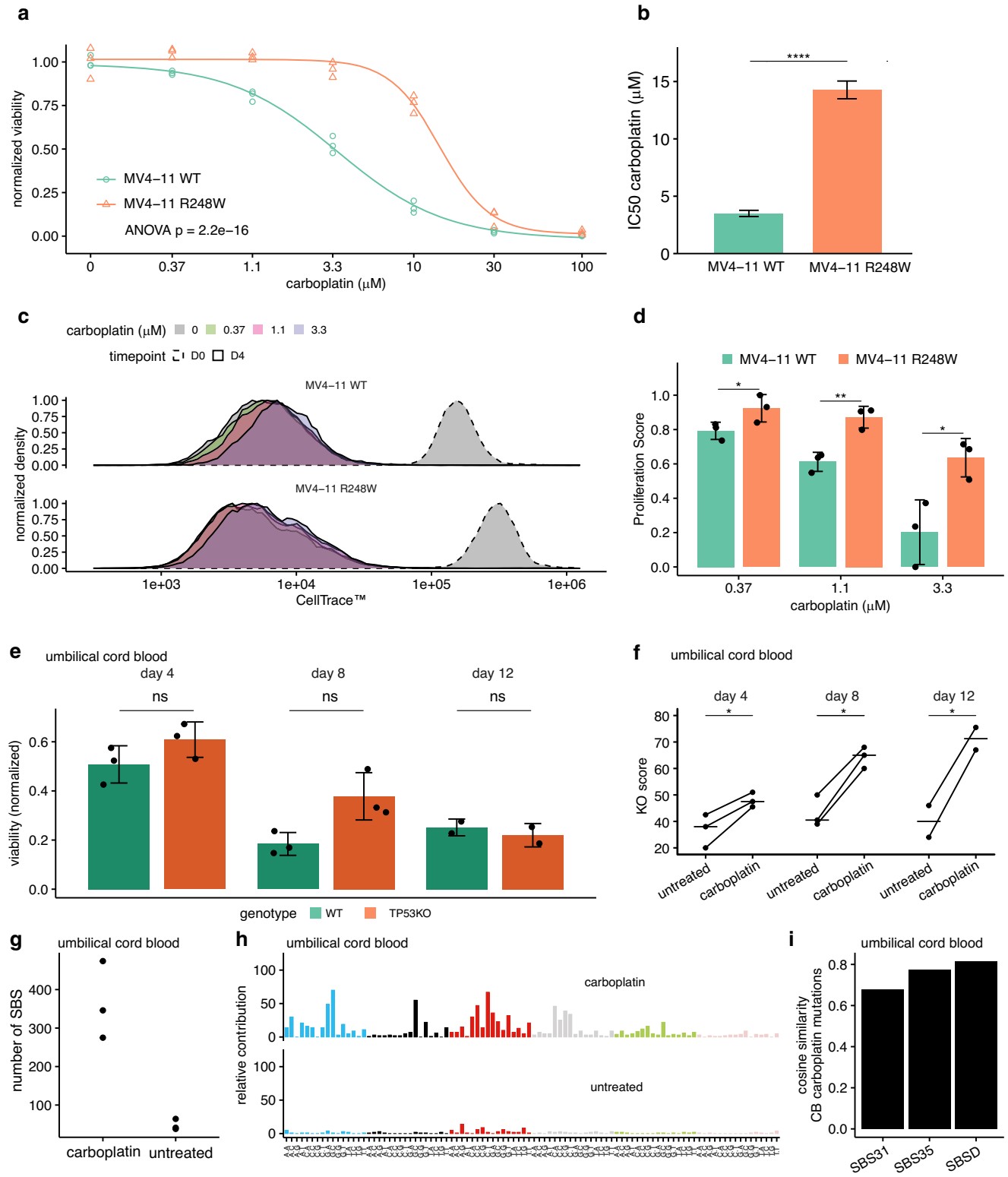

*TP53* is needed for this selective advantage would indicate that also in adult *TP53*-aberrant t-MN the wild-type *TP53* copy is already lost before treatment, opposite to a previous model by Wong et al.[7]. This hypothesis is supported by the events that caused loss of wild-type *TP53* in UPN034 before treatment.

Following our findings, a model arises in which comparable selective pressures are present during platinum-based treatment in adults and children. However, due to a higher number of mutations that accumulated due to aging and potentially environmental mutagenic exposures, adults would have a higher chance to already have *TP53*-mutated cells in their blood before the start of treatment[26,49,50]. The chance would thus be relatively high in adults that one of these cells loses the *TP53* wild-type allele before treatment, and therefore gains a competitive advantage and develops into t-MN. In contrast, in children *TP53* aberrations are hardly present in blood. Thus, only cells in which both *TP53* wild-type alleles are lost prior to or in the beginning of treatment, can then expand during treatment. As this chance is much lower in children, a smaller part of pediatric compared to adult t-MN

**Fig. 5 | TP53 deficiency enables increased proliferation under platinum treatment. a** Dose-response curves of MV4-11$^{WT}$ (circles) and MV4-11$^{R248W}$, based on the DAPI-negative fraction of single cells (gating strategy in Supplementary Fig. 9b), $n = 3$ biological replicates per cell line (average of three clones). The complete dose-response model was tested against the null model, lacking genotype information (ANOVA). **b** The IC50 values of carboplatin treatment per genotype (average of three cells), extracted from the dose-response models depicted in (**a**). The comparison of the IC50 values is based on a z-test and error-bars represent the standard error ($p = 1.58 * 10^{-27}$). **c** CellTrace™ signal per treatment condition, normalized to unit area, for MV4-11$^{WT}$ cells (top) and MV4-11$^{R428W}$ cells (bottom). Representative measurements for a single clone per genotype are shown. **d** Proliferation Score per treatment condition for MV4-11$^{WT}$ and MV4-11$^{R248W}$. The scores of each cell line were compared within treatment conditions using a Holm's corrected one-sided T-test ($p = 0.0428$, $0.00936$, and $0.0366$ for 0.37, 1.1, and 3.3 µM carboplatin, respectively). Error bars represent standard deviation of the mean of three independent experiments. **e** Normalized viability of CD34+ umbilical cord blood cells after 4, 8,

and 12 days of carboplatin treatment (22.5 µM), based on the live cell count per condition. The viability was normalized to each matched untreated condition and compared using a Holm's corrected one-sided T-test. Error bars represent the standard error of the mean of $n = 3$ biological donors, each examined in an independent experiment. **f** The KO score of the TP53 KO conditions based on ICE analysis (Synthego) with and without carboplatin treatment (22.5 µM). The KO score was compared using a one-sided paired T-test ($p = 0.036$, $0.012$, and $0.036$ for D4, D8, and D12, respectively) and each data point represents a biological replicate ($n = 3$ independent experiments, one unique biological donor per experiment). **g** The number of single base substitutions detected in WGS data of untreated and carboplatin-treated clonally expanded umbilical cord blood cells. **h** The 96-trinucleotide mutational profiles of the data shown in (**g**). **i** The cosine similarity between the mutational profile of carboplatin-treated umbilical cord blood cells and the mutational profile of in vivo platinum-based drug exposure (SBS31, SBS35, SBSD). NS not significant, ****$p < 0.0001$, **$p < 0.01$, *$p < 0.05$. Source data are provided as a Source Data file.

harbor *TP53* aberrations. On the other hand, in children with germline *TP53* aberrations this chance is higher, thus the clonal evolution of the t-MN can mimic that of adults. Hence, in these children possibly even more t-MN are driven by *TP53* deficiency and the loss of the *TP53* wild-type allele, as in theory every HSPC in their blood could lose the wild-type *TP53* allele. Finally, for cells with other, non-*TP53*, aberrations, such as *KMT2A* fusions, expansion mostly occurs after the end of platinum-based treatment. When expansion occurs, this seems to happen rather quickly, conceivably because the niche is very sparsely populated. In line with these findings, previous studies on t-MN in adults do not indicate a shorter latency time of t-MN with *TP53* aberrations compared to those without *TP53* aberrations[7].

In addition to a different clonal trajectory, *TP53*$^{-/-}$ tumors accumulate distinct mutational profiles under carboplatin treatment. This signature (SBSD/sbs25) was previously identified in the metastases of treated cancer patients and linked to carboplatin exposure[36]. Here, we confirm this relationship and show enrichment of SBSD in both t-MN and metastases with *TP53* aberrations. It is very likely that more interactions between germline aberrations and treatment exposure exist. Due to the relative rarity of germline aberrations, and the large variety of different treatments that are used, large datasets are needed for the identification of these interactions. Interactions between germline mutations and treatments, such as the *TP53* – carboplatin interaction that we reveal here, could have clinical implications, not only for future choices of treatment regimens, but also for follow-up of these patients after treatment.

## Methods
### Patient samples
Patient samples were collected via the biobank of the Princess Máxima Center for pediatric oncology and via a collaboration with the I-BFM AML SG from Austria and the German AML-BFM study group in accordance with the Declaration of Helsinki. Informed consents were obtained from all participants. For the Dutch samples, ethical approval was granted in the past for treatment according to the respective clinical trials the patients were treated in. These study protocols were approved by the involved ethics committee of the center the patient was treated at. Clinical trial information, if applicable, is summarized in Supplementary Data 4. Samples were collected and banked at the Dutch Childhood Oncology Group (DCOG), after informed consent for study participation was obtained. The DCOG transferred all clinical data and biological specimens with the merge of all pediatric oncology centers in the new facility (the Máxima) in 2018. In the Netherlands when re-using banked samples for wet-lab research that were originally stored with informed consent for additional research, no additional ethics committee approval is required, instead the Institutional Review Board of the Máxima approved this study under proposal

PMCLAB2020.151. For the German patients, ethical approval was granted by the ethical committees of the corresponding Institutional Review Boards of the Medical Association Westphalen-Lippe and Medical Faculty of WWU Münster (AML-BFM Study 2004) and the Ethics Committee of Hannover Medical School (AML-BFM Registry 2012). For IBFM67, the Austrian patient, the ethics committee granted approval within the framework of the paedMyLeu registry. Umbilical cord blood was obtained from the Wilhelmina Children's Hospital after approval by the Biobank Committee of the University Medical Center Utrecht (protocol number 15-341).

Sex has been stated for all included patients. Sex was determined based on clinical data from either the Princess Máxima Center for pediatric oncology, the German Society of Pediatric Oncology and Hematology (GPOH) or St. Anna Children's Cancer Research Institute. If these data were missing, these data were acquired from whole genome sequencing (WGS) results. Sex and or gender were not considered in study design because this study contains samples from a historical cohort of a rare subset of patients.

### Sample work-up
Bone marrow mononuclear cells were stained for fluorescence-activated cell sorting (FACS) after thawing. Hematopoietic stem and progenitor cells (HSPCs) were identified using the following surface markers: Lin-CD11c-CD16-CD34 + , CD38-/CD45RA ("HSPC mix"). t-MN blasts were defined based on diagnostic immunophenotype data if available. If immunophenotype data were unavailable, cells were stained with the same antibody panel including CD33, CD34 and CD38. B-and T-cells were identified using the following surface markers: CD3, CD4, CD20, CD33, CD34 ('mature mix'). Representative gating strategies are shown in Supplementary Fig. 10 and 11.

Blasts and HSPCs were purified on a SH800S Cell Sorter (Sony, software v2.1.6). First, blasts were sorted in bulk for DNA isolation after which single HSPCs and t-MN blasts were index sorted in a 96-well plate prepared with PTA-buffer. Additionally, single HSPCs were sorted in a 384-well plate prepared with 75 µL HSPC culture medium per well. HSPC culture medium consisted of StemSpan SFEM medium (Stemcell technologies, Cat#09650) supplemented with SCF (100 ng/mL; Miltenyi Biotec Cat#130-096-696), Flt3-ligand (100 ng/mL; Miltency Biotec Cat#130-096-480), IL-6 (20 ng/mL; Stem Cell Technologies Cat#78050), IL-3 (10 ng/mL; Stem Cell Technologies Cat#78040), TPO (50 ng/mL; Miltenyi Biotec Cat#130-095-754), UM729 (0.5 µM; Stem Cell Technologies Cat#72332) and Stemregenin (750 nM; Stem Cell Technologies Cat#72344).

HSPCs sorted in HSPC culture medium were cultured for 4−7 weeks at 37 °C, 5% $CO_2$ before harvesting. Mesenchymal stromal cells (MSCs) were cultured from a bone marrow fraction of 500,000 cells/well in 12-well culture dishes with 2 mL DMEM-F12 medium

(Thermo Fisher Scientific, GIBCO, Cat# 12634028), supplemented with 10% FCS (Thermo Fisher Scientific, Cat#10500). Medium was refreshed every other day to remove non-adherent cells and MSCs were harvested when confluent, after ~ 2–3 weeks.

**FACS antibodies.** All antibodies were obtained from Biolegend, except for CD13 (Biosciences). Antibodies used for t-MN blast and HSPC populations: CD34-BV421 (clone 561, 1:20, Cat#343609), lineage (CD3/CD14/CD19/CD20/CD56)-FITC (clones UCHT1, HCD14, HIB19, 2H7, HCD56, 1:20, Cat#348701), CD38-PE (clone HIT2, 1:50, Cat#303505), CD90-APC (clone 5E10, 1:200, Cat#328113), CD45RA-PerCP/Cy5.5 (clone HI100, 1:20, Cat#304121), CD33-PE/Cy7 (clone WM53, 1:100, Cat#303433), CD49f-PE/Cy7 (clone GoH3, 1:100, Cat#313621), CD16-FITC (clone 3G8, 1:100, Cat#302005), CD11c-FITC (clone 3.9, 1:20, Cat#301603), CD123-Pe/Cy7 (clone 6H6, 1:100), CD13-PerCP/Cy5.5 (Biosciences, clone WM15, 1:20, Cat#561361), CD14-APC (clone HCD14, #Cat325607).

For the B- and T-cell sort the following antibodies, obtained from Biolegend, were used: CD3-PE/Cy7 (clone SK7, Cat#344815), CD4-PerCP/Cy5.5 (clone OKT, Cat#317427), CD20-BV421 (clone 2H7, Cat#302329), CD33-APC (clone WM53, Cat#303407), CD34-APC (clone 561, Cat#343607).

## DNA isolation and WGS
DNA was isolated from cell pellets of blasts, MSCs and clonally expanded HSPCs ("HSPC clones") using the DNeasy DNA Micro Kit (Qiagen, Cat#56304), following the manufacturer's instructions. The standard protocol was slightly adjusted by adding 2 μL RNase A (Qiagen, Cat#19101) during the lysis step and eluting DNA in 50 μL low EDTA TE buffer (10 mM Tris, 0.1 mM EDTA, G Biosciences).

DNA from single HSPCs and blasts of all patient samples except IBFM67 was amplified using the ResolveDNA® WholeGenome Amplification Kit (BioSkryb, Cat#100136) according to the manufacturer's instructions. Single HSPCs and blasts of IBFM67 were processed with the ResolveDNA® Whole Genome Amplification Kit v2.0, using a D100 Single Cell Dispenser (HP).

For each sample, DNA libraries for Illumina sequencing were generated from at least 45 ng genomic DNA using standard protocols. For PTA-amplified DNA at least 500 ng genomic DNA was used. The libraries were sequenced on Novaseq 6000 sequencers (2x150bp) at a depth of 15x (clones/single cells) or 30x (bulk t-MN and control samples). Two t-MN bulk blast DNA pallets (15 and 22% purity) were sequenced at a depth of 90x. For IBFM67, three t-MN cells were excluded from further analyses as they had < 70% of the genome covered at 5x depth.

## Read mapping
Sequencing reads were first mapped to genome GRCh38 using Burrows-Wheeler Aligner (bwa) v0.7.17 using "bwa mem −M −c100", then duplicates were marked using Sambamba v0.6.8 and base recalibration was performed using GATK's BaseRecalibrationTable and BaseRecalibration. All GATK tools were from version 4.1.3.0.

## Mutation calling, filtering, annotation
Mutation calling was performed with GATK's HaploTypeCaller, on all samples of a patient combined. Mutation filtering was performed using GATK's SelectVariant with options "*select_type = type == 'SNP'? '−select-type SNP −select-type NO_VARIATION': '−select-type INDEL −select-type MIXED*'" on the resulting multi-sample VCFs. Next, GATK's VariantFiltration was run with the following options: −filter-expression "MQ < 40.0" −filter-expression "FS > 60.0" −filter-expression "HaplotypeScore > 13.0" −filter-expression "MQRankSum <−12.5" −filter-expression "ReadPosRankSum <−8.0" −filter-expression "MQ0 >= 4 && ((MQ0/ (1.0 * DP)) > 0.1)" −filter-expression "DP < 5" −filter-expression "QUAL < 30" −filter-expression "QUAL >= 30.0 && QUAL < 50.0" −filter-

expression "SOR > 4.0" −filter-name "SNP_LowQualityDepth" −filter-name "SNP_MappingQuality" −filter-name "SNP_StrandBias" −filter-name "SNP_HaplotypeScoreHigh" −filter-name "SNP_MQRankSumLow" −filter-name "SNP_ReadPosRankSumLow" −filter-name "SNP_HardToValidate" −filter-name "SNP_LowCoverage" −filter-name "SNP_VeryLowQual" −filter-name "SNP_LowQual" −filter-name "SNP_SOR" -cluster 3 -window 10." And finally −filter-expression "QD < 2.0". For the two bulk-tMN samples that were sequenced 90x (UPN018, UPN023) the final expression was replaced by −filter-expression "QD < 1.0", to prevent filtering out somatic mutations with very high coverage. The additional mutations with a QD between 1 and 2 had a similar mutational profile to those with a QD > 2 (cosine similarity of 0.95 and 0.90 respectively), and all had 40% or more contribution of SBS87 (79% QD < 2 vs 62% QD >= 2, and 47% QD < 2 vs 46% QD >= 2, respectively), which is a thiopurine-related signature, and is not similar to any artifact signature, which confirmed that these additional mutations were not artifacts, but true somatic mutations.

Annotation of variants was done with SNPEffFilter, SNPSiftDbnsfp (dbNSFP3.2a), GATK VariantAnnotator (COSMIC v.89) and SNPSiftAnnotate (GoNL release 5). A full pipeline description is available at https://www.github.com/UMCUGenetics/NF-IAP. The version of NF-IAP used was v.1.3.0.

## Somatic mutation filtering
SMuRF v3.0.0 was used for obtaining high-confidence clonal somatic mutation calls (www.github.com/ToolsVanBox/SMuRF). These were mutations that (A) were positioned on autosomal chromosomes, (B) had a GATK phred-scaled quality score R 100, (C) had a mapping quality of 60 (30x coverage) or 55 or higher (15x), (D) had a base coverage of at least 10 (30x) or 5 (15x), (E) had a GATK genotype quality of 99 (heterozygous) or 10 (homozygous) in both the sample and paired control (if available), (F) had no evidence in the paired control sample if available. Finally, mutations with low variant allele frequencies (VAF) were removed to obtain a set of clonal mutations. The VAF cut-off for 15x sequenced cells and clones was 0.15[14], as mutations below this cut-off could be technical artifacts or mutations acquired in vitro. The cut-off for 30x sequenced bulk t-MN samples with no contamination of healthy cells was 0.3, as mutations below this cut-off are subclonal[51,52]. For 30x sequenced bulk t-MN samples with ~80% and ~50% purity the cut-offs were 0.24 and 0.15. For 90x sequenced bulk t-MN samples with 15% and 22% purity, the cut-off was 0.07. These cut-offs were determine based of the VAF distributions of the somatic mutations (Supplementary Fig. f 3).

## Mutation filtering for bulk t-MN without a paired normal
For bulk t-MN samples without a paired normal, or with evidence of blast contamination in the normal ($n = 2$), all HSPC clones were used for filtering[14]. In these cases, a mutation identified by SMuRF to be in the bulk t-MN was excluded if it was (a) clonally present in all samples that passed QC for that mutation, (b) subclonally present in any sample, or (c) not confidently absent in at least one sample.

## Mutation filtering for PTA single-cell WGS data
For single cells, our in-house developed pipeline PTATO v1.3.3[53] was applied, which takes the VCF files produced by SMuRF v3.0.0 and filters these using germline mutations combined with a random forest and walker to separate real somatic mutations from amplification-induced artifacts. For a full description of this pipeline, see our recent publication[53].

## Mutation set for phylogenetic reconstruction
CellPhy was run on all clones and cells of a patient. The mutation set that was used consisted of the mutations in the clones (from SMuRF) and cells (from PTATO), supplemented with the mutations in the bulk t-MN (from SMuRF).

## Phylogenetic tree construction

Phylogenetic reconstruction was performed using CellPhy v0.9.2[54], which utilizes RAXML-NG[55], a maximum likelihood framework for phylogenetic inference. CellPhy considers the allelic dropout rate and amplification errors per sample and constructs the most likely tree based on these estimates. CellPhy was run on the phred-scaled genotype likelihoods (PL) with standard settings, including the "GT16 + FO + E" model. 100 bootstrap iterations were run. CellPhy was also used to map mutations to the tree (using the "--mutmap" function) with the "--opt-branches off" setting. The CellPhy "--support" function was used to run 100 bootstrap replicates. These number of bootstraps in which a split in the tree is detected, is depicted in Figs. 3 and 4.

The trees were rooted with treeio's "root" function, which is a wrapper for the "root" function from the R package 'ape'. The MSCs were used as the outgroup, or an HSPC which shared no mutations with the other samples, when MSCs were unavailable (UPN008). Finally, mutations of end branches were filtered out if one or more reads supporting the alternative allele was present in more than one cell. In all cells and all shared branches combined, 7.6% of mutations had a missing genotype.

The shape of the trees was validated by assessing the VAF in the bulk t-MN data of the mutations in the tree. Mutations from clonal branches were all found in the bulk t-MN and had a VAF distribution around 0.5, which is in line with the mutations being present in all t-MN blasts. The majority of the mutations in subclonal branches were found at the bulk t-MN, but at a lower VAF, validating that only a subset of t-MN blasts carried these mutations. Most private t-MN mutations were not found in the bulk t-MN blasts, indicating that they were indeed only present in one or very few blasts, and thus not detectable in bulk. Some mutations were found in the bulk t-MN at low VAF, indicating that these were actually subclonal mutations, and thus labeled so. Finally, mutations from HSPC private branches were not found in the t-MN bulk. In addition, the number of mutations missed in the bulk t-MN was evaluated to ensure no major clones were missed (Supplementary Fig. 6).

## Small driver events

Single base substitution and indel driver events were extracted from the output of SMuRF v3.0.0. Only exonic mutations with MODERATE or HIGH impact according to the SnpEff annotation were considered. In addition, mutations in genes from the COSMIC cancer gene consensus v97 and pediatric AML drivers genes were included. Finally, missense, nonsense, frameshift, insertions and deletions were considered as driver events. Driver events (including SVs and CNVs) were visualized using the R package ComplexHeatmap v.2.12.0[56].

## Structural variation calling

The Hartwig Medical Foundation's gridss-purple-linx pipeline v1.3.2 (https://github.com/hartwigmedical/hmftools) was applied on the bulk t-MN blast samples and their paired normal to call somatic structural variants (SVs) and determine copy number alterations (CAN) with options '−amber_tumour_only "true" −cobalt _tumour_only "true" −purple_tumour_only "true"'. All structural variants were checked in IGV and false positives were filtered out. The sub-packages "amber" and "cobalt" that are part of this pipeline were also applied on the bulk blood WGS from time of primary diagnosis of patient UPN034, which was obtained from the diagnostics department.

Germline SVs were extracted from unfiltered GRIDSS VCFs using bcftools filter v1.14[57] in multiple steps using the following filters. (1) -i 'FILTER = = "PASS" && BMQ > 40 && FORMAT/RP > 10 && MQ > 55 && INFO/ASQ > 0', (2) -i 'INFO/ASSR > 20 | INFO/ASRP > 20' $QUAL > $ QUAL2. These variants passing these filters were then overlapped with driver genes from a pediatric cancer WGS dataset from Grobner et al.[23] using "bedtools[58] intersect -header -wa". Finally, the final set of SVs was manually investigated in IGV and only SVs with supporting reads in the matched control and t-MN sample and those overlapping with at least one exon of a cancer gene were reported. Structural variants were visualized using the R package "circlize" v0.4.15. Breakpoints were visualized using the R package "Gviz" 1.40.1[59,60].

## Baseline and mutation load normalization

Somatic mutations were re-called using SMuRF v3.0.0 in samples from a previously published baseline of HSPCs of healthy donors. The number of autosomal mutations in the baseline and t-MN samples were corrected based on GATK's CallableLoci's CALLABLE length. A linear mixed-effects model was fit on the baseline samples and the slope and intercept of this line were used to calculate the expected mutation load for each t-MN sample.

## Mutational signature extraction and refitting of bulk t-MN

MutationalPatterns package v3.6.0 was used for mutational signature extraction and refitting of the bulk t-MN samples. As signature extraction becomes more robust when more samples are included, we performed signature extraction on the single base substitutions of all t-MN bulk samples together with previously published data of 34 healthy HSPC clones of healthy individuals of different ages to ensure that the normal clock-like signatures SBS1, SBS5, and HSPC could be reliably extracted[26]. Extraction was done by applying the *extract_signatures* function with options "rank = 12, nrun = 100". Then, the signatures that correlated to signatures from the COSMIC database v3.2 or previously identified signatures in the Dutch part of this cohort[14], with a cosine similarity of 0.8 or higher were substituted by the highest correlating signature. Signatures that could be reconstructed by three or fewer COSMIC signatures (cosine ≥ 0.85) were substituted by those known signatures. One of the signatures was separated in SBS87, SBS17a and SBS17b.

Refitting was done with the extracted signatures using *fit_to_signatures_bootstrapped* with options "n_boots = 100, max_delta = 0.05". The contributions from the 100 refits were averaged and the difference between the total numbers of mutations in each sample and the sum of refitted mutations was categorized as "unexplained".

## Mutational signature extraction and refitting of phylogenetic trees

Mutational signatures were extracted from the final sets of mutations mapped to the branches of the trees, i.e., for each branch of each tree, the mutations were grouped and used as a "sample" for the mutation extraction. Branches with < 100 mutations were excluded. Eight signatures were extracted, five of which corresponded to SBS1, SBS2, SBS31, SBS18, SBSD, and HSPC (cosine similarity > 0.85) and were replaced with the original signature. In addition, two additional signatures (SBSH and SBSI) were extracted.

Fitting mutational signatures to the mutations of single branches of the tree was done with MutationalPattern's function "fit_to_signatures_strict" with option "max_delta = 0.05". Only branches with 100 mutations or more were considered.

Total signature contributions were estimated by bootstrapped refitting (*n* = 100). For this analysis, branches were assigned to a type (e.g., 'private', 'subclonal', 'clonal') and combined per type per patient. For end branches containing one cell/clone, mutations that were detected in one or more reads in the matching bulk t-MN sample were categorized as 'subclonal', other mutations were categorized as 'private'. Only those branches with 100 mutations or more were considered. Trees were visualized using the R package "ggtree" v3.4.1[61] which uses "ape" v.5.6-2[62].

## Analysis of a WGS cohort of metastases

A cohort of solid tumor metastases previously described by Priestley et al.[36] was used to verify signatures that were not described in the COSMIC database. Somatic and germline mutation calls as well as copy

number status were obtained from the Hartwig Medical Foundation (HMF) that created this dataset. SBSG was refitted to the dataset together with COSMIC signatures v3.2 using the function "fit_to_signatures_strict" from the MutationalPatterns package with option "max_delta = 0.01". Previously, Pich et al.[31] have extracted signatures from this cohort, including signature "sbs25", that was similar to the signature SBSD that we extracted from our cohort. We therefore refitted these signatures, including a signature similar to COSMIC SBS31, together with COSMIC SBS35 to this dataset in the same manner. In this case, only patients that were treated with cisplatin or carboplatin were selected based on the meta data of the data set.

*TP53* mutation status was extracted from three different sources. First, a *TP53* allele was considered to be lost if the BAF of the region including (a part of) *TP53* was > 0.95, as determined by the above mentioned gridss-purple-linx pipeline from HMF. Somatic and germline calls were also obtained from HMF and exonic *TP53* mutations that were annotated in ClinVar as "Pathogenic" or "Likely Pathogenic" were considered driver mutations, as well as all other nonsense and frameshift mutations. For the somatic mutations, also all missense, structural interaction variants, splice site variants, and protein-protein interaction variants were considered driver mutations.

## MV4-11 cell culture

The *MV4-11^bulk* and *MV4-11^R248W* cells were kindly provided by the Frank van Leeuwen and Willem Cox (Princess Máxima Center for pediatric oncology, Utrecht, The Netherlands). The frequency of the *TP53* R248W variant was determined by Sanger Sequencing using the Indigo tool (Gear-Genomics)[63]. MV4-11 cells were cultured at 37 °C, 5%CO₂ and replated biweekly at 1-4 * 10⁵ cells/mL in IMDM (Gibco™, Cat# 12440061), supplemented with 10% FBS (Sigma-Aldrich, Cat# 10270) and 1% Penicillin-Streptomycin (Gibco™, Cat# 15140122). Clonal cultures harboring either wild-type *TP53* or the homozygous R248W *TP53* variant were obtained by single cell sorting using a SH800S Cell Sorter (Sony). The *TP53* genotyping procedure is described below. Cultures were tested negative for mycoplasma every 6 weeks (MycoAlert®, Lonza, Cat# LT07-318) and the cell line identity was confirmed by STR profiling and comparison to the DSMZ CellDive database[64].

## MV4-11 carboplatin resistance assay

Following staining by CellTrace™ Far Red (Invitrogen™, Cat# C34564), MV4-11 cells were seeded at 25.000 cells per mL for a carboplatin (Fresenius Kabi) treatment range and an untreated condition. After four days, cell viability and proliferation were assessed by DAPI staining (Sigma-Aldrich, Cat# D9542) and subsequent flow cytometry (CytoFLEX S (Beckman Coulter), using CytExpert Data v2.4.0.28). For cell viability, we determined the DAPI-negative fraction of single cells in FlowJo™ (v10.8.1), which was normalized to the matched untreated condition. The IC50 (ED50) values were extracted from the summary of the dose-response model using the R package DRC (v.3.0-1)[65]. Representative gating images are available in Supplementary Fig. 9b. The proliferation score was calculated as follows. Per condition, the median fluorescent CellTrace™ intensity (MFI) of DAPI-negative cells was determined. Next, per condition, the MFI was normalized to the MFI at the start of treatment, and then to the untreated condition at day four. Subsequently, the normalized CellTrace™ intensities were rescaled to a range of 0-1 and transformed ($ProliferationScore = 1 - MFI_{scaled}$). Treatment conditions of 10μM and higher of carboplatin were not taken along as the number of viable cells was insufficient (<1.000 cells) for CellTrace™ measurements in the wild-type conditions. Each experiment consisted of the average of three technical replicates per condition of three clonal cultures per genotype and was performed independently three times. Visualization was performed using the R packages DRC (v.3.0-1)[65] and ggpubr (v.0.6.0)[66].

The fitted dose-response curves were tested by ANOVA against the null model, which lacked the genotype factor of MV4-11 WT and

R248W (R package stats v4.2.2[67]). The IC50 values were compared by z-test (default settings DRC::compParm v.3.0-1)[65]. The Proliferation Scores and cell viabilities after drug treatment were compared using a one-sided *t*-test and holm's multiple testing correction (rstatix v.0.7.2). The KO scores were compared using a paired one-sided *T*-test and holm's multiple testing correction (rstatix v.0.7.2).

## CRISPR/Cas9-targeting of TP53 in human CD34+ umbilical cord blood (UCB) cells

For all genome editing experiments, primary human UCB-derived CD34+ cells were purchased from STEMCELL Technologies (single donor) and cultured in HSPC culture medium. Targeting of *TP53* was carried out as follows[68,69]. The ribonucleoprotein complex containing either the Alt-R® CRISPR-Cas9 Negative Control crRNA #1 or TP53 targeting guide Hs.Cas9.TP53.1.AM (5'-TCCATTGCTTGGGACGGCAA-3', IDT) was precomplexed according to manufacturer's instructions in a 10:1 molar ratio with the Alt-R™ S.p. Cas9 Nuclease V3 (6.1 pmol per reaction) and stored at −20 °C until use. Per reaction, 100,000-200,000 cells were transfected using the Neon™ Transfection System (10 µl kit, Invitrogen™, Cat# MPK1025) with electroporation settings 1600 V, 10 ms, and three pulses according to manufacturer's instructions. To validate the loss of p53 function, *TP53*-targeted cells were treated with 1000 µM nutlin-3a (Cayman Chemical, Cat# 18585) for four days and assessed for an increase in KO score (Supplementary Fig. 9d) in two independent experiments using different donors.

## CD34 + HSPC carboplatin resistance assay

To assess the carboplatin resistance in WT versus TP53 KO HSPCs, the genetically engineered cells were cultured at 50.000 cells/mL in HSPC culture medium at 37 °C, 5% CO₂, 5% O₂. Cells were replated in new medium every four days. Four days after transfection, the cells were treated with 22.5 µM carboplatin (Fresenius Kabi), the approximate IC50 for four days of treatment in the wild-type condition. At day zero, four, eight, and twelve of carboplatin treatment, the cell viabilities were monitored through viable cell counting and samples were taken for genotyping of the TP53 locus. The experiment was repeated three times independently using unique donors. In order to assess the effect of carboplatin treatment on the genome of wild-type HSPCs, UCB was obtained from the Wilhelmina Children's Hospital (Utrecht). From UCB, CD34+ cells were isolated by lymphoprep gradient separation (Stem Cell Technologies Cat#07861) and subsequent magnetic-activated cells sorting with anti-CD34 magnetic beads using the MACS system from Miltenyi Biotec (Cat#130-100-453) with LS columns (Cat#130-042-401) according to manufacturer's instructions[70]. CD34+ cells were cultured in the hereabove-mentioned HSPC medium. After overnight incubation at 37 °C, 5% O2 and 5% CO2, these cells were treated with the approximate IC50 concentration of carboplatin (10 µM) for three days. Subsequently, Lin-/CD34 + /CD38-/CD45RA-HSPCs, defined by FACS-sorting, were plated for clonal expansion and WGS analysis, similarly to the patient bone marrow HSPCs.

## Genotyping of *TP53* locus

The presence of the *TP53* R248W variant in the MV4-11 cells and CRISPR-Cas9-induced insertions and deletions in the CD34 + UCB cells were assessed by Sanger sequencing (Macrogen) following cell lysis (DirectPCR, Viagen Biotech, Cat# 301-C) or DNA extraction (Quick-DNA™ MicroPrep, Zymo, Cat# D3020) (Supplementary Fig. 9a). The locus was amplified by PCR (GoTaq® G2 Hot Start Master Mix, Promega, Cat# M7422) using the following primers. MV4-11: Fw – CCTGCTTGCCACAGGTCTC; Rev – GGGGATGTGATGAGAGGTGG. CD34 + UCB: Fw – GTGGATCCATTGGAAGGGCA; Rev - GCTGCCCT GGTAGGTTTTCT (IDT). CD34 + UCB cells: Visualization of genotyping was performed through alignment to ENST00000269305.9 in Benchling (2023). The KO score of the genetically engineered CD34 + UCB cells was assessed by ICE analysis (Synthego)[71].

## Data visualization

All data visualization that was not performed with the packages mentioned above was performed using the R package "ggplot2" v.3.3.6, which is part of the "tidyverse" suite of packages[72].

## Reporting summary

Further information on research design is available in the Nature Portfolio Reporting Summary linked to this article.

## Data availability

The whole genome sequencing (WGS) data generated in this study have been deposited in the European Genome-phenome Archive (EGA) under accession code EGAS00001005141. The raw WGS data are available under restricted access due to privacy laws, access can be obtained via the Princess Máxima Data Access Committee [https://ega-archive.org/dacs/EGAC00001001864]. The processed WGS and other data needed to generate the figures have been deposited in the Mendeley Database [https://data.mendeley.com/datasets/9d7mhxzt9g/1][73]. The raw mutation data of the cohort of solid tumor metastases we obtained are available under restricted access due to privacy laws. Access can be obtained via the Hartwig Medical Foundation [https://www.hartwigmedicalfoundation.nl/en/data/data-access-request/]. Source data are provided with this paper.

## Code availability

The code that is used for this study is available via Mendeley Data: https://data.mendeley.com/datasets/9d7mhxzt9g/1. https://doi.org/10.17632/9d7mhxzt9g.1[73].

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

## Acknowledgements

This work was funded by an ERC Consolidator grant to R. van Boxtel from the European Research Council (ERC; no. 864499). In addition, this work was supported by the Oncode institute, funding E.J.M. Bertrums, J.K. de Kanter, M. Verheul, M.J. van Roosmalen, L.L.M. Derks, L. Trabut and R. van Boxtel. The authors want to thank the Hartwig Medical Foundation for facilitating the low-input whole-genome sequencing. The authors thank Willem Cox, Frank van Leeuwen and Ronald Stam for kindly providing the MV4-11 cell lines. Ruben van Boxtel is a New York Stem Cell Foundation – Robertson Investigator. This research was supported by The New York Stem Cell Foundation. The authors want to acknowledge members of the International Berlin-Frankfurt-Münster AML Study Group (I-BFM AML SG) for the inclusion of patients in this study.

## Author contributions

E.J.M.B. and J.K.d.K were responsible for the experimental design. E.A., D.R. M.N.D. and N.M. included patient samples and assembled clinical data. H.H., D.R., B.F.G., C.M.Z., M.M.v.d.H-E, M.N.D, N.M., and R.v.B. supported the international collaborative study protocol. E.J.M.B. performed the majority of the experimental t-MN work, with support of M.V. L.L.M.D. performed the majority of the experimental in vitro validation work, with support of L.T. J.K.d.K performed all bioinformatic data analyses, with support of M.J.v.R. E.J.M.B. and J.K.d.K were responsible for data interpretation. E.J.M.B., J.K.d.K, L.L.M.D. and R.v.B drafted the manuscript. All authors have proof-read the manuscript.

## Competing interests

The authors declare no competing interests.
