## [Peer Review File · Nature Communications]

Editorial Note: Pages 4, 12, 26 and 57 in this Peer Review File have been amended to follow editorial policy for the ECR Co-Review Scheme.

REVIEWER COMMENTS

Reviewer #1 (Remarks to the Author): Expert in blood cancer genomics and evolution, therapy-related myeloid neoplasms, and cancer therapy

In this study, Bertrums and colleagues utilized single-cell whole-genome sequencing (WGS) to elucidate the evolutionary trajectory of therapy-related myeloid neoplasms (TMN) in pediatric patients who had previously undergone chemotherapy. Their research shed light on the expansion patterns of distinct premalignant clones following platinum exposure, providing insights into the acquisition of key driver mutations during this treatment. Furthermore, the study provides interesting insight regarding the role of TP53 status in determining the specific types of mutational signatures induced by platinum and the timing of premalignant clone expansion.

Overall, the paper is methodologically robust, well-written, and accessible. The Authors are clear experts in the field and the paper represents an important contribution to the definition of TMN pathogenesis and evolution in pediatric patients.

I have only minor comments and suggestions:

Page 2, lines 51-53: Based on recent discoveries in the field and the data presented in this study, I concur with the Authors' conclusion that the acquisition of clonal hematopoiesis (CH) and pre-TMN during treatment is indeed observed in pediatric patients. The findings of Coorens et al. in their study published in *Blood* in 2022 provide further support for this observation. Authors reported that this finding contrasts with what is typically observed in adults, where TMN is primarily attributed to the chemotherapy-driven selection of pre-existing CH. However, a recent paper by Diamond et al. suggests that the acquisition of additional driver mutations and complex genetic events can be induced or facilitated by exposure to certain chemotherapies with direct mutageneses, such as platinum and melphalan. This study proposes that these specific chemotherapies not only enable the selection of existing clones but also contribute to the acquisition of new driver mutations. Therefore, based on these recent findings, the sentence should be revised to reflect the possibility that certain chemotherapies can facilitate both the selection and acquisition of drivers in TMN development.

I couldn't locate Extended Data Table 1 in the provided submission material. I believe this table contains a clinical summary of the series, which would be valuable in understanding the prior exposure, dates, age, and other relevant information for each case. For instance, it would be informative to know if the patients who experienced KMT2A events were also treated with platinum. Additionally, it would be interesting to explore whether the exposure to platinum and TOPi (topoisomerase inhibitors) creates distinct conditions that contribute to the acquisition or selection of KMT2A translocations.

Page 3: Were patients with TP53 loss found to have a shorter latency period before the onset of TMN? Additionally, how do various factors, including different exposures, genetic alterations, and somatic drivers, impact the timeframe for TMN development?

Do the authors observe increased complexity in terms of copy number variations (CNV), structural

variations (SV), and complex events (e.g., chromothripsis) among patients exposed to mutagenic agents? Such an observation would be intriguing and could provide support for the notion that these chemotherapies induce more extensive damage beyond single nucleotide variations (SNV) and Indels.

It would be useful in terms of consistency with prior literature to link these evolutionary models with the concept of single-cell expansion published by Pich et al. Nat Gen 2019 and Landau et al. Nat Comm 2020

Page 6, line 250: Because of the presence of multiple branching and subclones selected over time I don't think we can use the term "neutral" for IBFM22.

The genomic evolution of patients with Li-Fraumeni syndrome is indeed a fascinating aspect that is well-explained and emphasized throughout the paper. It would be informative to know if the Authors investigated whether the CNV and SV profiles of normal and tumor cells in Li-Fraumeni syndrome patients were more complex compared to other patients. Recent studies have suggested that normal cells from individuals with DNA repair deficiencies often exhibit increased complexity and alterations (e.g., PMID: 36450981). Examining this aspect could provide further insights into the genomic characteristics associated with Li-Fraumeni syndrome and contribute to our understanding of DNA repair deficiencies in relation to cancer development.

Figure 2d could benefit from some improvements to enhance the visibility and clarity of the new signatures. One suggestion is to reduce the y-axis limit to allow for better visualization of the signatures. This adjustment would enable a clearer representation of the data and aid in the interpretation of the findings. Additionally, including a table in the supplementary materials with the specific values of the signatures would provide readers with detailed information and facilitate further analysis and comparison. For example, despite is a little bit collapsed, SBSE looks similar to the melphalan signature SBS-MM1 reported in Rustad et al Nat Comm 2020 and Landau et al Nat Comm 2020. It seems from Figure 2a that this patient had Rhabdoid cancer, which is often treated with high-dose melphalan and autologous stem cell transplant. Was this patient exposed to melphalan? Is this SBSE similar to SBS-MM1?

Regarding Figures 3 and 4, if the clock-like signatures SBS1 and SBS5 no longer reflect the age in TMN, it would be appropriate to reconsider the terminology of "molecular time" for the y-axis label.

Figure 4. UPN034 is a very interesting case. It would be interesting beyond TMN to show the size and eventual SV involved in these del17p and the subsequent convergent evolution.

Reviewer #2 (Remarks to the Author): Expert in paediatric leukaemia genomics and evolution, therapy-related myeloid neoplasms, clinical genomics, and single-cell sequencing

The study by Bertrums et al is a follow up to their recent study in Cancer Discovery evaluating the

mutational and evolutionary patterns of tMN in children. Here they specifically focus on the patterns of evolution in children treated with cisplatin and the relationship to TP53 status. Unlike the previous findings in adults from Wong et al in which it was demonstrated that therapy allows for an outgrowth of preexisting TP53 mutant clones, here the authors clearly show that the therapy induces mutations in children. The authors convincingly argue that children with Li-Fraumeni syndrome, by the nature of already having a pre-existing pool of TP53 clones, are more similar to adults with tMN that acquire TP53 mutations as a normal part of aging. Overall this is a strong study that definitely advances our understanding of tMN. However, the MV4;11 studies are incomplete in terms of proving the hypothesis. In addition to generating a homozygous mutant isogenic line, the TP53 mutation should also be corrected to generate a companion wild-type line. Also, the authors need to evaluate the mutational signatures of these isogenic lines when treated with cisplatin to see if they recapitulate the patterns observed in their clinical samples.

Other issues that need resolution:

More clarity is needed regarding the shared samples between this current study and the previous study in Cancer Discovery, especially as it pertains to Figure 1. There appears to be significant similarities with the first figure in both of these studies.

Panels A, C and E in Figures 3 and 4 are not very intuitive and required a lot of time and thought to understand them. These data are clearly complicated, but a different style is recommended.

It is unclear how many single cell genomes (both HSPC and tMN) were generated for each patient in Figures 3 and 4. It was also challenging to understand which data was generated from cultured HSPCs and single cells using PTA. These data appear to be used interchangeably but a comparison of the results within a given sample would be beneficial to see if the expansion skews the clonal structure.

Typo line 307-R238W should be R248W.

Reviewer #3 (Remarks to the Author): Early Career Researcher co-reviewer

[This review has been completed under a Nature Communication initiative involving one early-career researcher. We both signed at the end to encourage transparency in the peer-review process.]

This is an interesting paper in which the authors use mutational signatures to characterize the selective pressures exerted by chemotherapy on pediatric therapy-related myeloid neoplasms (t-MN) and their interplay with germline TP53 mutations. The main result is that platinum induces the t-MN in children, but growth during platinum exposure depends on the TP53 genotype, a finding that might be used for a more efficient treatment.

We value the great effort made by the authors, who have done much work and carried out specific experiments to validate the hypotheses derived from their results. While we might agree with the main conclusions –particularly after seeing the cell line experiments– we have several caveats regarding experimental design, methodology, reproducibility, and interpretation. Namely, the experimental design felt unusual, with different types of samples whose purpose was unclear. Importantly, samples are small within each patient, so caution is needed when interpreting the results. Following the methodology was not always possible, lacking enough detail to reproduce the work; it felt cumbersome, with several analyses consisting of an excess of steps, seemingly arbitrary and often with unjustified thresholds. Regarding inpatient clonal evolution, we noted several inconsistencies. Below we develop these questions in detail.

*** General comments

1. Unusual experimental design

1.1. We found the experimental design unusual, mixing bulk, expanded clones, and single cells, which have distinct characteristics and are subject to different biases. We suggest the authors explain the purpose of each type of dataset. For example, why do you need two types of normal samples (MSC and HSPC)? Or why do you need both expanded HSPC clones and single HSPC cells?

1.2. In addition, we believe there is a need for a better description of the data, including which data sets were obtained for which patients and describing their composition (e.g., number of cells), purity (bulks), average sequencing depth, breadth of coverage, percentage of missing data (particularly important for single cells), and any other details that might be deemed relevant.

1.3. The number of cells or expanded clones per patient is small, less than 10 for the datasets corresponding to the trees shown. Many lineages will go unsampled. Why were so few cells/clones sequenced per patient?

1.4 Furthermore, were phylogenetic trees built for all patients? Why are only six described?

2. Unclear methodology

2.1. It is not straightforward to understand which analysis was performed or which tool was used with each dataset. Similar analyses are presented in distinct sections without a clear logic. For example, the information for t-MN blasts is correctly presented in “Mapping and mutation calling, filtering, and annotation.” At the same time, calling methods for HSPC clones + single cells are included in the “Phylogenetic tree reconstruction” section. We recommend re-arranging the methods section to improve clarity. Specifically, it would be beneficial to present the variant calling + filtering approach for all samples in the same section.

2.2. The filtering scheme requires a more comprehensive explanation as it currently requires more work

to grasp what exactly was done and why. For instance, what is the rationale behind the VAF thresholds? Is this only related to the purity of the bulk t-MN blast samples? If so, was a different threshold used for each patient? And why 0.3 (30x), 0.15 (15x), 0.07 (90x), which looks arbitrary and strange?

2.3. Did the authors apply any filtering steps for the HSPC expanded clones? While, in principle, these samples should have no “healthy” contamination, they can accumulate several mutations during the clonal expansion. Was this accounted for? Importantly, we assume that for the HSPC clones, only the clonal mutations were used for subsequent analyses (signatures and phylogenetic reconstruction), correct? This should be made clearer.

2.4. We had some difficulty understanding exactly how mutational signatures were extracted from the data. While the authors used the somatic mutations found in the t-MN bulk samples from each patient, the level of methodological detail is not enough to fully understand how the data was used. In lines 740-741, the authors state that signature extraction was done using “the mutational matrix of all t-MN bulk substituted with the 34 healthy HSPC samples that were used to construct the baseline”. What does this mean? Also, mutational signatures were assigned to branches with seemingly dozens of mutations. How was the procedure in this case? Did the authors independently use the function “fit_to_signatures_strict” on each set of branch mutations using the same probability threshold as before? Is this reliable?

2.5. Mutations in each branch were assigned to different signatures to identify the processes operating during and after chemotherapy exposure. According to the authors, these mutations were fit to the signatures identified in the bulk t-MN samples. Given that the trees include other sample types (i.e., HSPCs), shouldn't the signatures be re-estimated, or do the authors assume that the active mutational processes should be the same across sample types? Also, how come the complete set of mutations in the HSPC clones in patients UPN008 and UPN034 are attributed to SBS31? Should not a subset of these mutations be attributed to other mutational processes? This should be clarified.

2.6. Structural variants were called with an in-house pipeline. Has this been published? Is it validated? How can a reader reproduce this analysis?

2.7. A tool called PTATO was used to call single-cell variants. As far as we know, this tool has yet to be peer-reviewed and published, so assessing its use is problematic. Indeed, multiple variant callers specifically developed for single-cell DNA sequencing (scDNA-seq) data (doi: 10.1016/j.csbj.2022.06.013) have been benchmarked and published that use sound statistical models to control WGA biases. While we understand that PTA has a different chemistry than other WGA strategies, we are unsure whether this is enough to justify using a novel single-cell variant caller that has not been benchmarked against available tools. Also, it is unclear how PTATO deals with ADO, as it apparently only considers false positives. Is this true? Authors should explain and justify their choice of this tool convincingly.

3. Phylogenetic analysis

3.1. We found the phylogenetic reconstruction strategy clumsy and often challenging to follow. It is also

often mixed with variant calling and filtering when they are different things. After reading this section, we are unsure how these trees were built and whether the methodology was the same for all datasets. Filtering mutations, building trees, and mapping mutations to branches are different things that here seem mixed. We suggest authors explain in different sections distinct sets of steps that should be implemented sequentially: (1) call mutations using appropriate callers, (2) filter unreliable ones out and define the final datasets, (3) apply some of the available methods for phylogenetic reconstruction, (4) map mutations to branches, and (5) identify signatures in branches or clades.

3.2. The strategy for phylogenetic reconstruction here consists of arbitrary steps and multiple filters that are impossible to reproduce. The criterion or algorithm for phylogenetic reconstruction must be mentioned (parsimony, distance, likelihood, etc.). We only read, “A tree was constructed from the resulting mutations.”, which is not informative.

3.3. Mutations and cells are added at different times without an evolutionary justification –no established phylogenetic method works this way. For example, it is unclear why (and how) the authors manually introduced the low-quality samples onto the phylogenetic trees. Most, if not all, phylogenetic methods allow for missing data, and we could not come up with a scenario here where adding tips to the tree a posteriori represents an advantage.

3.4. Established methods exist to build phylogenetic trees from scDNA-seq data; some do it jointly with variant calling. Why did the authors not use well-known tools for phylogenetic tree reconstruction already benchmarked and based on explicit sound statistical models (like SiFit, SCIPHI, or CellPhy)?

3.5. The trees are rooted, but no explanation of how rooting was performed is offered. Rooting is a crucial aspect of phylogenetic reconstruction, also very relevant here, and should be explained in detail.

3.6. Conclusions based on trees can only be as good as the trees themselves. We will always get a tree regardless of the input data. However, the derived conclusions are only reliable if the data support the tree. While the phylogenetic bootstrap is not a p-value, it does help to understand whether the data supports a given branch. Authors should compute and display bootstrap values (or another measure of phylogenetic support) and base their conclusions only on well-supported branches. Bootstrap calculations are straightforward, using multiple software for phylogenetic reconstruction, including R packages.

3.7. Which method was used to assign the mutations to each tree? Does each mutation occur only once in the phylogenies? Statistical methods exist to accomplish this for scDNA-seq data (e.g., [10.1186/s13059-021-02583-w](https://doi.org/10.1186/s13059-021-02583-w))

3.8. We did not understand the role of the MSC samples during variant calling or filtering. For UPN008 and UPN034, why are mutations subclonal in the “MSC control” included? What is this control? Why are mutations subclonal in other samples excluded? Why mutations absent in one sample were included? We do not follow all this. Please define “mutation” (e.g., present in x but absent in y), and explain the rationale of every filter.

3.9. There is an arbitrary manual filter for ADO using hetSNPS. However, models and tools exist to do this in a sound statistical way (mentioned above). Why not use them? Still, this explanation should be in the variant calling section.

4. Interpreting inpatient clonal evolution

4.1 These are (very) small samples of a much larger population. Therefore, the most recent common ancestor (MRCA) of the t-MN cells will never be the initially transformed t-MN cell, as the authors suggest when labeling this node. This needs to be corrected. The t-MN MRCA will be younger than the origin of the t-MN. This also means that mutations shared between all the sampled t-MN blasts did not necessarily accumulate before the expansion of the initial leukemic cell. Many t-MN lineages could go unsampled, so t-MN mutations might look clonal when they are subclonal or look private when in reality, they are subclonal. This should be considered when discussing if the expansion happened during or after treatment. The inference's scope here concerns the *sampled* t-MN clones, not the initial t-MN clone.

4.2. Several trees (UPN008, IBMF22, UPN034) have a big basal polytomy and several short interior branches. This indicates a low phylogenetic signal in the data and is very important when interpreting trees. Once branch support is established (e.g., bootstrap values), authors should avoid making firm conclusions from weakly supported branches.

4.3. There is no temporal information in these trees. Hence, the temporal marks (ticks and length and location of black triangles) in Figure 3 b,d,f, and Figure 4 b,d,f of the origin and expansion of the t-MN are arbitrary, as authors do not really know the absolute timing of the expansion. Please note that the number of mutations is only directly related to time in a strict molecular clock model.

4.4. "Punctuated evolution" is a concept that comes from paleontology to explain the discontinuity of the fossil record and is about *phenotypic* change, with long periods of stasis followed by periods of apparently rapid speciation—in fact, this is known in evolutionary biology as "punctuated equilibrium." For some unknown reason, it has been loosely adopted by the cancer genomics community. We should not forget that selection targets phenotypes, not genotypes. In molecular evolution, we often see changes in the substitution rate among lineages, particularly when comparing distant species, and this is simply described as rate variation among lineages, which can be due to various factors. As simple as that. There is no need to invoke macroevolutionary theories of phenotypic change among species that do not apply in this scenario. There is nothing remarkable about the rates of evolution in patients UPN008, IBFM32, or IBFM42. A long clonal branch and short subclonal branches are not unexpected, considering the small sample sizes and precisely the effect of chemotherapy. Also, please note reliable rooting is essential to trust the branch lengths. We suggest removing the term "punctuated evolution" here and in the discussion.

4.5. The idea that platinum exposure inhibits the expansion of the initial leukemic clone and that this expansion thus starts when the exposure to platinum ends is based only on three patients, or are there more supporting this idea? How were the patients shown selected? If there are more trees, we suggest showing them as supplementary material.

4.6. For patient IBFM42, the pattern observed is unusual, as signature SBS31 seems on and off; if we trust the order of the nodes –which most likely we cannot, bootstrap will tell if this is the case. Please reconsider interpreting this case based only on well-supported nodes.

4.7. For patient IBFM32, the t-MB blasts do not have a common origin. How is that possible? Again this might be due to phylogenetic uncertainty and lack of support. Some measure of support is necessary to modulate the interpretation. Also, the black diamond in this patient indicates, according to the authors, the clonal mutations in the t-MN bulk that were not found in any of the sequenced single blasts. Given this statement, did the authors create an artificial clone with selected mutations? Most likely, such a clone never existed, which might distort the phylogenetic reconstruction. What is the justification for this? Strikingly, if we look at the tree, the branch leading to this clone seems to have zero length, indicating no private mutations. Or is there just a tiny branch there? Please clarify all the rationale.

4.8. For patient IBFM22, signature SBS31 does not appear in the branch leading to the t-MN clade, which the authors interpret as the result of an early expansion during treatment. This might be the case, but the other option is that the t-MN was not the result of the treatment. Have the authors considered this, or does this idea make no sense?

4.9. For UPN034, there are no t-MN blasts, so should we guess t-MN is equivalent to CH? Authors depict an early expansion during cisplatin treatment that declines or stops before FU. Why?

4.10. Importantly, for patient UPN034, one cannot confidently say there were four independent TP53 losses because all lineages involved originate in a polytomy. A polytomy here means uncertainty about the phylogenetic relationships among these clones, so depending on the true but unknown evolutionary relationships among them, the observed losses could be explained as one, two, three, or four independent events. We simply cannot tell the exact number because of the polytomy, so there is no clear evidence of convergent evolution, and this tree cannot be used to support such a hypothesis in the Discussion.

4.11. The tree of patient IBFM14 is very unresolved and will likely have very little support, so that no reliable inferences might be derived from it. As mentioned, the fact that some branches are longer than others has nothing to do with punctuated evolution. This can be natural rate variation among lineages due to coalescent sampling, and for cells might also be explained by different breadth, coverage, and error rates during WGA and scDNA-seq.

5. Discussion

5.1. The model proposed in the second paragraph of the discussion implies that t-MN tumors have lost all their wild-type TP53 copies. This should be easy to check from available bulk data. Is this the case?

5.2. Please remove references to punctuated evolution or neutral evolution. The former does not apply in this context, and the latter cannot be inferred just by looking at tree shapes. And please do not forget

these trees are sample genealogies, so their shape would change if you select other cells from the exact location again. A sample genealogy is related to but is not the same as the tumor history. Sampling error (coalescent) plays a role.

*** Specific comments

200: We guess the authors mean "Fig. 2e".

251: "... the developmental trajectory in this case is indicative for a neutral or more evolution". This statement has no basis. It is impossible to tell whether evolution is neutral just by looking at the tree.

276: "Since we were not able to perform WGS on the bulk t-MN blasts, we compared the HSPC WGS data with diagnostic data from single nucleotide polymorphism (SNP) and karyotype assays." We do not understand the meaning of this statement in this context.

290: At least in our pdf, there is here a stranded "Next,"

697: Mutation calling was performed using GATK for which datasets? Also, which of the different GATK tools was used (Haplotypecaller? or Mutect2?)

717: SMuRF is a method that filters (GATK) variant calls, in this case, to identify clonal mutations in bulk t-MN samples (also HSPC clones), which have low purity, right? This should be made explicit.

726: "In these cases, a mutation was filtered if". We guess it meant "a mutation was kept if."

793: For IBFM22, IBFM32, and IBFM42, we are unsure about the final set of mutations used to build the tree. Is it the overlap among PTATO in single cells and GATK-SMuRF in clones and bulk?

804: "A mutation was filtered out if more than 50% of germline variants had a VAF that differed more than 0.2 from the VAF in the MSC bulk in at least one sample." We do not understand what this means and where the 0.2 comes from. We assume that the first VAF is in the single cells and that mutations become missing data on a cell-per-cell basis.

809: "For both approaches, all shared mutations of branches with 10 or fewer mutations were visually inspected in IGV, and false positives were excluded." What are false positives? Shared mutations among cells assigned to branches with a total number of assigned mutations less than 11? Why less than 11?

810: "A tree was constructed from the resulting mutations." Which mutations? We are pretty lost at this point.

811: "Then, previously excluded mutations that exactly fitted the tree were added back." What is to fit? Please explain and justify.

812: "IBFM42, mutations that did not fit in the tree exactly were still added to the tree if they uniquely fitted to a single branch, assuming that they were missed in one single PTA-amplified cell." So here, the authors sound as if to be manually calling ADOs, assuming reversals are not possible, right? How does this link with the het-SNP approach?

814: "For low-quality samples in which 15% or more of the genome was not sufficiently covered, for each branch, the mean VAF of the mutations with sufficient coverage (>10) was calculated" A VAF from 11 reads is not reliable

816: "A cell was assigned to a shared branch if this average VAF was higher than 0.45.". How is a cell assigned to a shared branch? We do not understand what this means. The mean VAF was estimated from which sample exactly? We are guessing that it must come from the bulk t-MN, but we don't follow the rationale behind this approach and wonder whether it is justified.

817: "For branches with a lower average VAF and end-branches, only the truly shared mutations were considered, and other mutations were kept as single-cell mutations (end branches)." What are truly shared mutations?

Phylogenetic reconstruction: We assume cells are assigned a lot of missing genotypes. How much? Please describe in detail.

Phylogenetic reconstruction: Only clonal mutations should be used throughout for expanded clones. Please confirm.

Phylogenetic reconstruction: For the bulk t-MN, how is/are the clonal sequence/s used for phylogenetic reconstruction determined? (see, for example, patient IBFM22)

Figure 3a: For UPN008, we do not see any t-MN data in the tree. Maybe this is a mistake, and the six HSPC clones (white squares) forming a clade are t-MN blasts and should be black triangles. Or are these immunophenotypically HSPC-like cells that shared all the t-MN drivers and other clonal mutations with bulk t-MN blasts? If this case, they should be relabeled as blasts in the tree.

Figure 3b,f: where does the number of years come from? How did you date the expansion?

Figure 3 legend: "Phylogenetic lineage tree" should be "Phylogenetic tree." "Phylo" and "lineage" are the same thing.

We have made many comments, hoping to increase precision and reproducibility. The general conclusions should be acceptable because of the cell line experiment. However, there is room to improve the methodology and the interpretation of the evolutionary analyses. Still, congratulations for the hard work.

Reviewer #4 (Remarks to the Author): Early Career Researcher co-reviewer

REVIEWER COMMENTS

Reviewer #1 (Remarks to the Author): Expert in blood cancer genomics and evolution, therapy-related myeloid neoplasms, and cancer therapy

In this study, Bertrums and colleagues utilized single-cell whole-genome sequencing (WGS) to elucidate the evolutionary trajectory of therapy-related myeloid neoplasms (TMN) in pediatric patients who had previously undergone chemotherapy. Their research shed light on the expansion patterns of distinct premalignant clones following platinum exposure, providing insights into the acquisition of key driver mutations during this treatment. Furthermore, the study provides interesting insight regarding the role of TP53 status in determining the specific types of mutational signatures induced by platinum and the timing of premalignant clone expansion.

Overall, the paper is methodologically robust, well-written, and accessible. The Authors are clear experts in the field and the paper represents an important contribution to the definition of TMN pathogenesis and evolution in pediatric patients.

I have only minor comments and suggestions:

Comment 1.1

Page 2, lines 51-53: Based on recent discoveries in the field and the data presented in this study, I concur with the Authors' conclusion that the acquisition of clonal hematopoiesis (CH) and pre-TMN during treatment is indeed observed in pediatric patients. The findings of Coorens et al. in their study published in Blood in 2022 provide further support for this observation. Authors reported that this finding contrasts with what is typically observed in adults, where TMN is primarily attributed to the chemotherapy-driven selection of pre-existing CH. However, a recent paper by Diamond et al. suggests that the acquisition of additional driver mutations and complex genetic events can be induced or facilitated by exposure to certain chemotherapies with direct mutageneses, such as platinum and melphalan. This study proposes that these specific chemotherapies not only enable the selection of existing clones but also contribute to the acquisition of new driver mutations. Therefore, based on these recent findings, the sentence should be revised to reflect the possibility that certain chemotherapies can facilitate both the selection and acquisition of drivers in TMN development.

We thank the reviewer for this addition and agree that the statement could have been more nuanced. We have revised the sentence and it now reads as follows. *“Together, these studies propose a model in which chemotherapy exposure induces leukemogenesis mainly by selecting mutated clones that in most cases predated the treatment exposure^{7,10,12}, and only partly by induction of mutations¹²”*. (lines 54-57)

Comment 1.2

I couldn't locate Extended Data Table 1 in the provided submission material. I believe this table contains a clinical summary of the series, which would be valuable in understanding the prior exposure, dates, age, and other relevant information for each case. For instance, it would be informative to know if the patients who experienced KMT2A events were also treated with platinum. Additionally, it would be interesting to explore whether the exposure

to platinum and TOPi (topoisomerase inhibitors) creates distinct conditions that contribute to the acquisition or selection of *KMT2A* translocations.

We apologize for the confusion. Extended Data Table 1 was included in the “main text PDF” at the very end of the document, after all Extended Data Figures, and not as a separate file, like Extended Data Table 2. This table indeed includes the patient diagnoses, treatment (protocol) information (as far as this information was available) and the patient age at time of different diagnoses.

We have tested whether platinum + TOPi exposure was associated with a higher rate of *KMT2A* fusions, compared to TOPi exposure by itself, but this was not the case (12/20, 60% compared to 8/10, 80%, $p = 0.42$, Fisher’s exact test).

Comment 1.3

Page 3: Were patients with TP53 loss found to have a shorter latency period before the onset of TMN? Additionally, how do various factors, including different exposures, genetic alterations, and somatic drivers, impact the timeframe for TMN development?

We thank the reviewer for this question. Unfortunately, we did not have enough *TP53*-loss samples ($n=2$) to perform a reliable analysis on this. In data from Wong et al. 2015 Nature, adult t-AML with a *TP53* alteration did not have a shorter latency than those without a *TP53* alteration (mean 7.50 years compared to 7.26 years, $p = 0.83$, t-test). Similarly, in Schwartz et al. 2021 Nature Communications, pediatric t-MN with a *TP53* alteration did not have a shorter latency compared to *TP53* wildtype t-MN (3.73 compared to 4.47 years, $p = 0.42$, t-test). The latency time of *TP53*-wildtype and *TP53*-loss samples could be influenced by the types of treatment, including hematopoietic stem cell transplantation. We show here that *TP53*-mutated t-MN can likely divide more efficiently under platinum treatment, but whether this difference in proliferation exists for all treatments is unclear. As patients with the same cancer type often get similar treatment, the analysis might be improved by splitting the patients by primary cancer type. In the hereabove mentioned datasets, there are too few samples to split these analyses up per primary tumor type, except for the ALL patients in Schwartz et al. (39/64 samples with information on *TP53* status). For these patients, no significant difference in latency is found ($p = 0.84$, t-test), but ALL patients do not receive platinum-based drugs.

Comment 1.4

Do the authors observe increased complexity in terms of copy number variations (CNV), structural variations (SV), and complex events (e.g., chromothripsis) among patients exposed to mutagenic agents? Such an observation would be intriguing and could provide support for the notion that these chemotherapies induce more extensive damage beyond single nucleotide variations (SNV) and Indels.

We thank the reviewer for the excellent suggestion. When we correlate the number of SVs to each treatment class, we see a trend towards a higher number of SVs after treatment with most of the treatments (although not significant), except for TOPi, platinum, alkaloid, and radiation treatments (**Rebuttal Figure 1A**). The finding of t-MN after TOPi exposure not having an increase in SVs is in line with *KMT2A*-rearranged t-MN having very few other events. We found that the number of total treatments a patient received is not correlated with the number of SVs ($p = 0.108$, **Rebuttal Figure 1B**). However, when we exclude the four negatively correlated treatments mentioned above, we find a significant association between the number of treatments a patient received and the number of SVs in the t-MN (p

= 0.01219, **Rebuttal Figure 1C**). However, we believe that removing treatments solely based on a negative trend to obtain this last result is too biased to include as a conclusion in our manuscript.

Rebuttal Figure 1. Treatment correlation with the number of SVs in t-MN.

- A) The number of SVs in t-MN after exposure to the indicated drug ('y'), or after no exposure to this drug type ('n'). P-values were determined by t-test and *fdr*-corrected. *TP53*-mutated t-MN were excluded.
- B) The correlation between the number of drugs received by a patient before developing t-MN, compared to the number of SVs found in the t-MN.
- C) Similar to B, but the treatments with a negative correlation with the number of SVs (platinum, TOPi, alkaloid, and radiation treatment) were excluded from the analysis.

Comment 1.5

It would be useful in terms of consistency with prior literature to link these evolutionary models with the concept of single-cell expansion published by Pich et al. Nat Gen 2019 and Landau et al. Nat Comm 2020

The models of Pich et al. 2019 and Landau et al. 2020 state that when investigating WGS data of relapsed tumors, a chemotherapy-related mutational signature can only be detected if a bottleneck of sufficient strength occurs during or after treatment, because only then do the originally treatment-related private mutations become sufficiently clonal to be detectable due to their higher VAF in sequencing bulk tissues (Landau/Pich 2019). In our study, this model can be used to state that the expansion of the original t-MN cell occurred

later during, or after treatment in the cases than hundreds to thousands of treatment-related mutations are detected in the bulk t-MN WGS data. We now state the following in the discussion (lines 362-367): *“Indeed, in nine out of eleven t-MN bulk samples of our cohort that were treated with platinum drugs, more than 500 (range 655-2652) platinum-induced clonal mutations were detected. This observation indicates that the most common recent ancestor (MCRA) of the t-MN originated at the end or after treatment, as only then the treatment-induced mutations become clonal, as previously explained by Landau et al. and Pich et al.^{31,48”}*

Comment 1.6

Page 6, line 250: Because of the presence of multiple branching and subclones selected over time I don't think we can use the term “neutral” for IBFM22.

We thank the reviewer for pointing this out. We have now left out all comments in our manuscript regarding a punctuated or neutral evolution, as the terms are not fully applicable our data, which is also pointed out by reviewer 3 and 4 (see comment 3.4.4). In the revised version of the manuscript, we have focused more on the timing of evolution compared to the exposure to chemotherapy.

Comment 1.7

The genomic evolution of patients with Li-Fraumeni syndrome is indeed a fascinating aspect that is well-explained and emphasized throughout the paper. It would be informative to know if the Authors investigated whether the CNV and SV profiles of normal and tumor cells in Li-Fraumeni syndrome patients were more complex compared to other patients. Recent studies have suggested that normal cells from individuals with DNA repair deficiencies often exhibit increased complexity and alterations (e.g., PMID: 36450981). Examining this aspect could provide further insights into the genomic characteristics associated with Li-Fraumeni syndrome and contribute to our understanding of DNA repair deficiencies in relation to cancer development.

We thank the reviewer for this addition. We compared the CNV and SV profiles for the patients treated with platinum-based drugs for which we build phylogenetic trees. We found a trend towards more clonal SVs and CNVs in the t-MN with a *TP53* aberration but this was not significant ($p=0.072$ and $p=0.071$ respectively, t-test, **Rebuttal Figure 2a, b**). For individual cells and clones, translocations could be evaluated in all cells. Here we found no more than 2 translocations per cell, and the fraction of cells that harbored a SVs mostly differed per patient unrelated to the *TP53* status, indicating a higher influence of the treatment and other factors than the *TP53* status (**Rebuttal Figure 2c**). As these numbers were not significant, we did not report them in the manuscript.

Rebuttal Figure 2. Number of clonal and private SVs and CNVs in t-MN separated on TP53 status.

- The number and type of clonal CNVs found in the t-MN (or CH in the case of UPN034) that arose after platinum treatment (shown in Figure 3 and 4).
- The number and type of SVs found in the same t-MN/CH samples as shown in a.
- The fraction of cells with at least one private translocation for the same patients as indicated in panel a and b, separated by TP53 status.

Comment 1.8

Figure 2d could benefit from some improvements to enhance the visibility and clarity of the new signatures. One suggestion is to reduce the y-axis limit to allow for better visualization of the signatures. This adjustment would enable a clearer representation of the data and aid in the interpretation of the findings. Additionally, including a table in the supplementary materials with the specific values of the signatures would provide readers with detailed information and facilitate further analysis and comparison. For example, despite is a little bit collapsed, SBSE looks similar to the melphalan signature SBS-MM1 reported in Rustad et al Nat Comm 2020 and Landau et al Nat Comm 2020. It seems from Figure 2a that this patient had Rhabdoid cancer, which is often treated with high-dose melphalan and autologous stem cell transplant. Was this patient exposed to melphalan? Is this SBSE similar to SBS-MM1?

We thank the reviewer for this comment and interesting thought. We have adjusted the Y-axis in Figure 2 and gave these panels a bit more room within the figure. In addition, we compared SBSE to COSMIC SBS99 (Melphalan). Indeed, the two signatures do share some

peaks, but they also have many differences, and the cosine similarity between the two patterns is 0.65, indicating that they are different (**Rebuttal Figure 3**). In addition, we were able to trace back additional treatment information from this patient, that had a rhabdoid tumor, which we did not have before. This way, we found out that the patient was indeed not exposed to melphalan. We have added the treatment data of this patient to Extended Data Table 1 (IBFM28).

Rebuttal Figure 3. Comparison between the 96-trinucleotide profiles of SBS99 (Melphalan) and SBSE (identified in this manuscript).

Comment 1.9

Regarding Figures 3 and 4, if the clock-like signatures SBS1 and SBS5 no longer reflect the age in TMN, it would be appropriate to reconsider the terminology of "molecular time" for the y-axis label.

We thank the reviewer for this suggestion. We previously named this axis "molecular time" to be able to compare the number of mutations with the years that it costs in healthy live to gain these mutations. As we understand this may be confusing, we have changed the axis title to "number of mutations".

Comment 1.10

Figure 4. UPN034 is a very interesting case. It would be interesting beyond TMN to show the size and eventual SV involved in these del17p and the subsequent convergent evolution. We have visualized all events that led to loss of 17p in the cells of UPN034 and added this information to **Extended Data Figure 8**. The lost regions are between 16.3 and 21.7Mb in size. The three independent losses of heterozygosity (LOH) and loss events in clones 1C3, 1G3, and 1M13 were "simple" events without any detected structural variants (which occurs when the end of a chromosome arm is lost). The 17p loss shared between five other cells was complex with many rearrangements between chromosome 5 and 17 occurring near the break end. We now mention this in the text (lines 286-288): "Interestingly, five HSPC clones, including all three FU clones, shared the same complex event involving chromosome 5 and 17 that resulted in the loss of the wild-type TP53 allele (Extended Data Fig. 8a-d)"

Extended Data Figure 8. Details of the UPN034 17p loss.

a) Copy number plot (top) and B allele frequency (BAF) plot (bottom) of whole genome sequencing (WGS) data of peripheral blood of patient UPN034 at the time of osteosarcoma (DX1). Data from the diagnostics department of our institute.

b) Similar to (a) but for a bone marrow HSPC at time of DX2.

c) A detailed list of breakpoints and copy number regions identified in the cells of patient UPN034.

d) The structural variants in clonal hematopoiesis (CH) clone 1L3 between chromosome 5 and 17, which led to the 17q loss in 5 cells (shown in c).

e) Similar to (d), but for the events between chr17 and chr7 of non-CH clone 1C3.

Reviewer #2 (Remarks to the Author): Expert in paediatric leukaemia genomics and evolution, therapy-related myeloid neoplasms, clinical genomics, and single-cell sequencing

The study by Bertrums et al is a follow up to their recent study in Cancer Discovery evaluating the mutational and evolutionary patterns of tMN in children. Here they specifically focus on the patterns of evolution in children treated with cisplatin and the relationship to TP53 status. Unlike the previous findings in adults from Wong et al in which it was demonstrated that therapy allows for an outgrowth of preexisting TP53 mutant clones, here the authors clearly show that the therapy induces mutations in children. The authors convincingly argue that children with Li-Fraumeni syndrome, by the nature of already having a pre-existing pool of TP53 clones, are more similar to adults with tMN that acquire TP53 mutations as a normal part of aging. Overall this is a strong study that definitely advances our understanding of tMN.

Comment 2.1

However, the MV4;11 studies are incomplete in terms of proving the hypothesis. In addition to generating a homozygous mutant isogenic line, the TP53 mutation should also be corrected to generate a companion wild-type line. Also, the authors need to evaluate the mutational signatures of these isogenic lines when treated with cisplatin to see if they recapitulate the patterns observed in their clinical samples.

Thank you for the suggestions. Based on this comment, we have performed additional experiments, and we believe that these additional data strengthen the message of the manuscript. We have further investigated the cell division rate *in vitro* and related this to TP53 status as well as platinum treatment. Of the initial bulk MV4-11 cells that we used as a control, approximately 5% harbored the TP53 R248W mutation. First, we clonally expanded single cells of this bulk, and selected clones that were homozygous TP53 wildtype and clones that harbored the TP53 R248W mutation (validated by PCR and sequencing). In addition, we aimed to revert the TP53 R248W mutation to the wildtype allele using CRISPR/Cas9-mediated adenine base editing. Unfortunately, the high efficiency of the adenine base editor resulted not only in a correction of R248W to the wildtype allele, but also in a new, likely pathogenic mutation five bases upstream of the R248W mutation leading to an amino acid substitution (M246T). Nevertheless, we repeated the carboplatin treatment experiment with the three types of homozygous clones (R248W, wildtype, and 'corrected' clones with M246T) and averaged the Proliferation Score of three clones per type across three independent experiments (see **Rebuttal Figure 4** below). In addition, we made a dose-response curve of the three independent experiments. The difference that we found in the dose-response of the R248W mutant and wildtype clones was even higher than in our initial experiment. The corrected clone with the additional M246T mutation was less

susceptible to carboplatin than the wildtype clones, but more than the R248W clones. A similar observation could be made for the Proliferation Score. We replaced the initial panels in Figure 5 with the R248W and wildtype clones (see below).

As we were unable to revert the R248W MV4-11 cells to TP53 wildtype, we included an alternative model targeting *TP53* in an independent manner. We used primary human CD34+ umbilical cord blood cells (UCB), which have an extremely low mutational background (clones in this paper: x-x single base substitutions). We used CRISPR/Cas9 gene editing to induce deletions in exon 4 of the *TP53* gene of these cells, which resulted in frameshift mutations. We subsequently treated wildtype and *TP53*-mutated UCB cells from three different donors and saw similar results as in our MV4-11 cell line experiments, based on cell viability and enrichment of mutant alleles after carboplatin treatment.

To validate the mutational pattern of carboplatin treatment in blood cells, we clonally expanded wildtype UCB cord blood cells that had been treated with carboplatin. We show that this exposure resulted in hundreds of mutations. Interestingly, the signature was more similar to SBSD that we observed in t-MN samples, than SBS31 or SBS35. We think this could be a result of the high division rate of the UCB cells that is induced by the medium and growth factors that are used to perform these experiments, and these findings might thus be in line with our previous results.

Based on these results we updated Figure 5 of our manuscripts and the paragraph on these experiments now is as follows (lines 330-356): *“Finally, we validated the effect of TP53 deficiency on cell proliferation under platinum treatment using MV4-11, a pediatric acute monocytic leukemia cell line with a subclonal TP53 R248W mutation. We expanded three clones with a TP53 wildtype allele (MV4-11^{WT}) and three clones with the a homozygous R248W allele (MV4-11^{R248W})^{46,47}. In line with our in vivo findings, MV4-11^{R248W} was more resistant to carboplatin treatment than MV4-11^{WT} (IC50 14.3μM vs. 3.5μM, $p=1.6*10^{-27}$, Fig. 5a, b, Extended Data Fig. 9a,b). Next, we used a single pulse of CellTrace™ dye, which is equally distributed over daughter cells during cell divisions, to track proliferation during in vitro treatment. From the fluorescent intensity after four days of treatment, we calculated a Proliferation Score (Methods). MV4-11^{R248W} showed a dose-dependent increase in proliferation compared to MV4-11^{WT}, confirming that TP53-deficiency leads to platinum resistance by enhancing proliferation during treatment (Fig. 5c, d). To further verify these results, we used CRISPR/Cas9 to induce TP53 knock-out insertions and deletions in CD34+ umbilical cord blood (UCB) cells, which have a minimal mutational background. This resulted in bulk CD34+ UCB populations with an average TP53 KO score of 38% (UCB^{TP53KO}, CI 95%: 20%-55%, Extended Data Figure 9c,d). We treated wildtype UCB cells (UCB^{WT}) and UCB^{TP53KO} with 22.5 μM carboplatin, the approximate IC50 at day 4 in UCB^{WT}. After 4 and 8 days of carboplatin treatment, the viability of UCB^{TP53KO} was higher than UCB^{WT}, although not significantly (Figure 5e). As the UCB^{TP53KO} is a heterogeneous population, we next compared the frequency of KO-inducing deletions at the targeted TP53 locus. We found that the fraction of TP53 KO-inducing deletions increased during carboplatin treatment compared to untreated UCB^{TP53KO} cells, indicating that TP53KO cells proliferated more under treatment ($p < 0.0182$, Figure 5f). Finally, to test the influence of carboplatin on the genome of UCB cells, we sequenced UCB^{WT} cells, both untreated, and treated with carboplatin for four days. Carboplatin induced 317 mutations (CI 95%: 67-568, Figure 5g). Interestingly, the mutational profile of the carboplatin treatment was more similar to SBSD than SBS35 or SBS31 (cosine*

similarity of 0.82, 0.77, and 0.68 respectively). Possibly, this is caused by the high division rate that is induced by the medium and growth factors in the UCB cells.”

Figure 5. TP53 deficiency enables increased proliferation under platinum treatment.

a) Dose-response curves of MV4-11^{WT} (circles) and MV4-11^{R248W}, based on the DAPI-negative fraction of single cells, $n = 3$ biological replicates per cell line (average of three clones). The complete dose-response model was tested against the null model, lacking genotype information (ANOVA). **b)** The IC50 values of carboplatin treatment per genotype (average of three cells), extracted from the dose-response models depicted in (a). The comparison of the IC50 values is based on a z-test and error-bars represent the standard error. **c)** CellTrace™ signal per treatment condition, normalized to unit area, for MV4-11^{WT}

cells (top) and MV4-11^{R248W} cells (bottom). *Representative measurements for a single clone per genotype are shown.* **d)** Proliferation Score per treatment condition for MV4-11^{WT} and MV4-11^{R248W}. The scores of each cell line were compared within treatment conditions using a holm's corrected one-sided T-test. Error bars represent standard deviation of the mean of three independent experiments. **** $p < 0.0001$, * $p < 0.05$. **e)** Normalized viability of CD34+ umbilical cord blood cells after four, eight, and twelve days of carboplatin treatment (22.5 μM), based on the live cell count per condition. The viability was normalized to each matched untreated condition and compared using a holm's corrected one-sided T-test. Error bars represent the standard error of three independent experiments. **f)** The KO score of the TP53 KO conditions based on ICE analysis (Synthego) with and without carboplatin treatment (22.5 μM). The KO score was compared using a one-sided paired T-test and each data point represents a biological replicate (independent experiments). NS = not significant, **** = $p < 0.0001$, ** = $p < 0.01$, * = $p < 0.05$. **g)** the number of single base substitutions detected in WGS data of untreated and carboplatin-treated clonally expanded umbilical cord blood cells. **h)** The 96-trinucleotide mutational profiles of the data shown in (g). **i)** The cosine similarity between the mutational profile

Rebuttal Figure 4. MV4-11 carboplatin treatment including the M246T

a) Dose-response curves of MV4-11^{WT} (circles), MV4-11^{M246T} (crosses) and MV4-11^{R248W} (triangles), based on the DAPI-negative fraction of single cells, $n = 3$ biological replicates per cell line (average of three clones). The complete dose-response model was tested against the null model, lacking genotype information (ANOVA). **b)** The IC50 values of carboplatin

treatment per genotype (average of three cells), extracted from the dose-response models depicted in (a). The comparison of the IC50 values is based on a z-test and error-bars represent the standard error. **c)** CellTrace™ signal per treatment condition, normalized to unit area, for MV4-11^{WT} cells (top) and MV4-11^{R428W} cells (middle), and MV4-11^{M426T} (bottom). *Representative measurements for a single clone per genotype are shown.* **d)** Proliferation Score per treatment condition for MV4-11^{WT}, MV4-11^{R248W}, and MV4-11^{M426T}. The scores of each cell line were compared within treatment conditions using a holm's corrected one-sided T-test. Error bars represent standard deviation of the mean of three independent experiments. **** p<0.0001, * p<0.05.

Other issues that need resolution:

Comment 2.2

More clarity is needed regarding the shared samples between this current study and the previous study in Cancer Discovery, especially as it pertains to Figure 1. There appears to be significant similarities with the first figure in both of these studies.

We thank the reviewer for this question. Indeed, there is an overlap in patients included in our previous study and the current study. To be able to analyze the newly acquired data in a reliable way, we have added as much data as possible. This means that this study comprises both our own in-house patient data, which we have previously published in Cancer Discovery, as well as our newly collected samples from a collaboration with the International Berlin-Frankfurt-Münster AML Study Group (IBFM AML SG). This way, we can obtain more reliable results as the cohort of included patients, analyzed in a similar manner, is as large as possible. We now refer in the text to the study of previously published samples to make sure readers are aware that some patients were re-analyzed in line 92-94 “We included 18 Dutch (Cancer Discovery 2022)¹⁴, one Austrian, and 25 German t-MN patient samples that were obtained via a collaboration with the International Berlin-Frankfurt-Münster AML Study Group (I-BFM AML SG).”.

In addition, we now color-indicated in the panels of Figure 1 and 2 what samples were part of the Cancer Discovery publication.

Comment 2.3

Panels A, C and E in Figures 3 and 4 are not very intuitive and required a lot of time and thought to understand them. These data are clearly complicated, but a different style is recommended.

We understand that the trees can be complicated and contain a lot of information. We have simplified the trees in Figure 3 and 4 by removing the separate branches for the bulk t-MN data (please see 3.4.1 for an elaborate description on this) and the “lower/high coverage” indication being removed. Unfortunately, based on comment 3.3.6, we had to add bootstrapping information. Therefore, we decided to depict the trees in an alternative style in **Extended Data Figure 7**. These do not contain all the information of the drivers and have the signature contribution bar plots incorporated into the branches. In addition, the type of samples is indicated in words instead of symbols. We hope that these new trees are now easier to interpret.

a UPN008

b IBFM32

c IBFM42

d IBFM67

e IBFM22

f UPN034

g IBFM14

Extended Data Figure 7. The phylogenetic trees in an alternative style.

The phylogenetic trees depicted in Figure 3 and 4, but with the signature contributions per branch in bar plots in the tree. In addition, the type of cell/clone is indicated in words instead of symbols. Finally, the information of the drivers is removed.

Comment 2.4

It is unclear how many single cell genomes (both HSPC and tMN) were generated for each patient in Figures 3 and 4. It was also challenging to understand which data was generated from cultured HSPCs and single cells using PTA. These data appear to be used interchangeably but a comparison of the results within a given sample would be beneficial to see if the expansion skews the clonal structure.

We thank the reviewer for bringing up this matter. Indeed, we have used both clonally expanded HSPCs and single cells to generate the lineage trees. We aimed to annotate this in Figures 3 and 4 by open and closed icons at the end of the branches (clones and single PTA cells, respectively). In addition, we do refer in the text to sequenced “clones” and “cells” to make this distinction. We have now clarified this sentence (lines 207-210): “WGS was performed on DNA directly amplified from single cells using primary template-directed amplification (PTA; referred to as “single cells”). Or, when possible, WGS was also performed for clonally expanded HSPCs (referred to as “HSPC clones”; Extended Data Table 3, 4)⁴⁰”. In addition, we sometimes referred to expanded HSPC clones as cells in the previous version of the manuscript. We apologize for this unclarity, and have now rectified/clarified what type was used in, for example, lines 238-239, 262, 302. In addition, we now added Extended Data Table 3 that includes the number of cells and clones sequenced for each tree in Figure 3 and 4.

In total, we have sequenced 29 cells and 22 clones. Unfortunately, it is impossible to perform a direct comparison between sequencing data of clones and PTA-processed single cells, as for only one patient both data types are available, and in that case, only one clone was sequenced. However, in our recent paper that describes the development of PTATO, the processing pipeline of WGS data from PTA-processed cells, we show that the number and type of mutations identified from PTA-derived data, after applying PTATO, is comparable to that of clonally expanded cells. In our manuscript, we also apply PTATO to ensure that we call high-quality mutations from PTA cells. Please find the abovementioned paper via “Middelkamp, S. et al. Comprehensive single-cell genome analysis at nucleotide resolution using the PTA Analysis Toolbox. *Cell Genomics* **3**, 100389, doi:<https://doi.org/10.1016/j.xgen.2023.100389> (2023).” We have also referenced to this paper in text (reference 53).

Comment 2.5

Typo line 307-R238W should be R248W.

We thank the reviewer for this correction. We have changed it in the manuscript accordingly.

Reviewer #3 (Remarks to the Author): Early Career Researcher co-reviewer

[This review has been completed under a Nature Communication initiative involving one early-career researcher. We both signed at the end to encourage transparency in the peer-review process.]

This is an interesting paper in which the authors use mutational signatures to characterize the selective pressures exerted by chemotherapy on pediatric therapy-related myeloid neoplasms (t-MN) and their interplay with germline TP53 mutations. The main result is that platinum induces the t-MN in children, but growth during platinum exposure depends on the TP53 genotype, a finding that might be used for a more efficient treatment.

We value the great effort made by the authors, who have done much work and carried out specific experiments to validate the hypotheses derived from their results. While we might agree with the main conclusions –particularly after seeing the cell line experiments– we have several caveats regarding experimental design, methodology, reproducibility, and interpretation. Namely, the experimental design felt unusual, with different types of samples whose purpose was unclear. Importantly, samples are small within each patient, so caution is needed when interpreting the results. Following the methodology was not always possible, lacking enough detail to reproduce the work; it felt cumbersome, with several analyses consisting of an excess of steps, seemingly arbitrary and often with unjustified thresholds. Regarding intrapatient clonal evolution, we noted several inconsistencies. Below we develop these questions in detail.

*** General comments

Comment 3.1. Unusual experimental design

Comment 3.1.1. We found the experimental design unusual, mixing bulk, expanded clones, and single cells, which have distinct characteristics and are subject to different biases. We suggest the authors explain the purpose of each type of dataset. For example, why do you need two types of normal samples (MSC and HSPC)? Or why do you need both expanded HSPC clones and single HSPC cells?

We apologize for the confusion. In our setup, we used the MSC controls to filter out germline mutations. MSCs are genetically distantly related to the HPSCs and t-MN as they share only very few early embryonal mutations with healthy HSPCs as well as the t-MN clone (see Osorio et al. 2018 <https://doi.org/10.1016/j.celrep.2018.11.014>, and Brandsma et al. 2021 <https://doi.org/10.1158/2643-3230.BCD-21-0010>). Therefore, we can use the MSCs as a germline control to filter inherited single nucleotide polymorphisms (SNPs) and identify somatic mutations in the t-MN and HSPCs except for the few shared early embryonic mutations. In phylogenetics the MSCs have two roles. First, when all HSPCs and t-MN cells share the same ancestor (e.g., IBFM14), the MSCs will be able to distinguish germline mutations from somatic mutations shared by all cells. Second, when sequencing single cells (using PTA), the MSCs are used in PTATO to filter out PTA-induced artifacts using linked-read analyses and germline SNP B-allele frequencies, see “Middelkamp, S. et al. *Comprehensive single-cell genome analysis at nucleotide resolution using the PTA Analysis Toolbox. Cell Genomics* **3**, 100389, doi:<https://doi.org/10.1016/j.xgen.2023.100389> (2023).”.

HSPCs on the other hand are used to compare the type of mutations that accumulate in t-MN blasts and normal cells. In addition, sometimes phenotypic HSPCs share part of the mutations and drivers with the t-MN blasts, making the phylogenetic reconstruction of the t-MN more detailed than when only using single t-MN cells.

The bulk t-MN mutation data is in the new version of the manuscript not incorporated when building the phylogenetic trees. It now serves four purposes. 1) making sure all clonal t-MN mutations are part of the phylogenetic reconstruction mutation set. 2) Separating private t-MN cell mutations from subclonal mutations, 3) to check the validity of the tree by checking the VAF of the mutations in the tree in the bulk t-MN, 4) using the same analysis to identify the MCRA of the bulk t-MN, which we annotate in the trees. All of these are further explained in other comments below.

In the text we state:

Lines 107-109: *“We performed WGS on bulk t-MN blasts and used mesenchymal stromal cells (MSCs) or bulk-sorted B-cells of the same patient as a germline control (Methods, Extended Data Fig. 2, 3).”*

Lines 219-220 *“The HSPCs are used to check if the same mutational processes are active in normal cells.”*

Lines 220-223: *“We validated the shape of the tree, the clonal, subclonal, and private labels for branches, and the most common recent ancestor of the bulk t-MN by investigating the VAF of the tree mutations in the matching bulk t-MN data and by investigating mutations present in the bulk t-MN, but not the tree (Extended Data Fig. 6).”*

Comment 3.1.2. In addition, we believe there is a need for a better description of the data, including which data sets were obtained for which patients and describing their composition (e.g., number of cells), purity (bulks), average sequencing depth, breadth of coverage, percentage of missing data (particularly important for single cells), and any other details that might be deemed relevant.

We thank the reviewer for raising this matter. We understand that it should have been easier to find the data generated per patient, and the WGS metrics of the sequenced samples. We now include Extended Data Table 3 with the number and type of samples sequenced per phylogenetic tree, and Extended Data Table 4 for the WGS metrics of every sample. We mention these in the text in lines 207-210: *“WGS was performed on DNA directly amplified from single cells using primary template-directed amplification (PTA; referred to as “single cells”). Or, when possible, WGS was also performed for clonally expanded HSPCs (referred to as “HSPC clones”; Extended Data Table 3, 4)⁴⁰.”*

Comment 3.1.3. The number of cells or expanded clones per patient is small, less than 10 for the datasets corresponding to the trees shown. Many lineages will go unsampled. Why were so few cells/clones sequenced per patient?

We would have been very happy if we could have sequenced more cells. However, t-MN material is very scarce, and often not many cells are viably frozen. In addition, usually a high percentage of the sample consists of blasts, which makes it even more challenging to reliably filter out the HSPCs. Lastly, not all plated HSPCs will grow out *in vitro*. Therefore,

from some patient samples it was not possible to expand more clones. This is one of the reasons we searched for novel techniques to perform DNA isolation from single cells during our study, and why we started using PTA. This way, we have a higher yield of single cells that we can sequence, and we can use the available material more efficiently.

In regard to the unsampled lineages, we use the bulk AML sequencing data to make sure that we cover all clonal mutations in the AML population. We have included these analyses in **Extended Data Figure 6**. Please see our response to comment 3.4.1 for more details.

Comment 3.1.4 Furthermore, were phylogenetic trees built for all patients? Why are only six described?

We thank the reviewer for this question. Here, we have focused on the effect of platinum-based drugs on the evolution of t-MN. We have included all trees that we generated of patients that were treated with platinum-based drugs. We have previously shown phylogenetic trees for other treatments as well (Bertrums et al 2022, <https://doi.org/10.1158/2159-8290.CD-22-0120>). Furthermore, as we have stated at comment 3.1.3, unfortunately, we did not have enough material from all patient samples to be able to sort cells for single-cell sequencing/clonal expansion. We did want to use all the material that we had, which is why we did use the bulk data from the patient samples for which we were not able to build phylogenetic trees. Via further collaboration with the International Berlin-Frankfurt-Münster (I-BFM) consortium, we have recently obtained material from patient IBFM67, another patient who developed an t-MN after treatment with platinum. The data is now included in **Figure 3**, and the results are described in lines 237-243: *“The phylogenies of patients IBFM32 (n=3 single HSPCs, n=3 blasts), IBFM42 (n=2 single HSPCs, n=1 HSPC clone, n=4 single blasts), and IBFM67 (n=2 single HSPCs, n=5 single blasts), all treated with cisplatin, showed a similar pattern. For these patients, besides HSPCs, we also included single blasts in our analysis. Apart from a single division detected in the middle of the t-MN evolution of IBFM32, all three patients showed long clonal branches and short subclonal branches (Fig. 3c-h, Extended Data Fig. 7b-d).”*.

3.2. Unclear methodology

Comment 3.2.1. It is not straightforward to understand which analysis was performed or which tool was used with each dataset. Similar analyses are presented in distinct sections without a clear logic. For example, the information for t-MN blasts is correctly presented in “Mapping and mutation calling, filtering, and annotation.” At the same time, calling methods for HSPC clones + single cells are included in the “Phylogenetic tree reconstruction” section. We recommend re-arranging the methods section to improve clarity. Specifically, it would be beneficial to present the variant calling + filtering approach for all samples in the same section.

We agree with the reviewers that the method section could have been better structured and apologize for this unclarity. We have now moved all sections on mutation calling and filtering to the beginning and clearly described each step in consecutive paragraphs (lines 806-984). In addition, in response to other questions in this section, we have also simplified the pipeline of building phylogenetic trees. Please see our responses below.

Comment 3.2.2. The filtering scheme requires a more comprehensive explanation as it

currently requires more work to grasp what exactly was done and why. For instance, what is the rationale behind the VAF thresholds? Is this only related to the purity of the bulk t-MN blast samples? If so, was a different threshold used for each patient? And why 0.3 (30x), 0.15 (15x), 0.07 (90x), which looks arbitrary and strange?

We now clarified the steps in this section more thoroughly and **included Extended Data Figure 3** to show why the filters were different for different depths.

- The sequencing depth was different for different types of samples. VAF cut-offs were dependent on the sequencing depth.
 - o 15x sequencing was only used for single cells and clones, as these are pure by definition. There are no subclonal mutations, as any mutation acquired after the 2nd cell division has a VAF of 0.125 or lower. Here the VAF cut-off was 0.15 (see VAF plots in **Extended Data Figure 3a** below).
 - o 30x sequencing was only used for bulk t-MN sequencing. Here the VAF cut-off for pure samples was 0.3. As clonal mutations have a VAF around 0.5, with a general spread between 0.3 and 0.7, we include mutations with a VAF of 0.3 or higher. Mutations lower than 0.3 could be subclonal mutations in these populations and are therefore excluded. We validated these cut-offs in Blokzijl et al. 2016 <https://doi.org/10.1038/nature19768> and Jager et al. 2017 <https://doi.org/10.1038/nprot.2017.111>.
 - o Two t-MN bulk samples (purity 15% and 22%) were received as DNA, not frozen cells, and could not be purified for t-MN blasts like the other samples. They were therefore sequenced to 90x. Because of the impurity and high depth, here a VAF cut-off of 0.07 was used. In **Extended Data Figure 3a**, we now depicted all VAF distributions and cut-offs of the bulk AML samples (also see below).
 - o In **Extended Data Figure 3b** we have plotted the distribution of the different types of samples.
- The VAF cut-off was adjusted in 30x sequenced, purified bulk AML samples for sequencing-based purity estimates
 - o Pure blast samples had a cut-off of 0.3. t-MN bulk samples with an approximate purity of ~80% and ~50% were filtered to an VAF-cut-off of 80% and 50% of the “normal” cut-off (0.24 and 0.15 respectively). This was previously not described and is now clearly mentioned in the Methods.
- **Extended Data Figure 3** is mentioned in lines 840-848: *“Finally, mutations with low variant allele frequencies (VAF) were removed to obtain a set of clonal mutations. The VAF cut-off for 15x sequenced cells and clones was 0.15 as described previously, as mutations below this cut-off could be technical artifacts or mutations acquired in vitro. The cut-off for 30x sequenced bulk t-MN samples with no contamination of healthy cells was 0.3, as mutations below this cut-off are subclonal, as described previously. For 30x sequenced bulk t-MN samples with ~80% and ~50% purity the cut-offs were 0.24 and 0.15. For 90x sequenced bulk t-MN samples with 15% and 22% purity, the cut-off was 0.07. These cut-offs were determine based of the VAF distributions of the somatic mutations (Extended Data Figure 3).”*

Extended data Figure panel 3. VAF distributions and cut-offs

- a)** The VAF distributions of the different types of samples included in the manuscript. The blue numbers indicate the VAF cut-offs used for each type.
- b)** The VAF distributions of the different bulk t-MN samples and the VAF cut-off used.

Comment 3.2.3. Did the authors apply any filtering steps for the HSPC expanded clones? While, in principle, these samples should have no “healthy” contamination, they can accumulate several mutations during the clonal expansion. Was this accounted for? Importantly, we assume that for the HSPC clones, only the clonal mutations were used for subsequent analyses (signatures and phylogenetic reconstruction), correct? This should be made clearer.

We agree that it is important to exclude *in vitro* acquired mutations in clonally expanded HSPCs. As explained in our answer to comment 3.2.2, after the 1st cell division, every mutation that accumulates *in vitro* has a VAF of lower than 0.25, after the 2nd division this is 0.125. This is the reason that we use mutations with a variant allele frequency of 0.15 or higher. We have previously shown that this results in a robust selection of mutations with very few false positives (Blokzijl et al. 2016 <https://doi.org/10.1038/nature19768>, Jager et

al. 2017 <https://doi.org/10.1038/nprot.2017.111>, Brandsma et al. 2021
<https://doi.org/10.1158/2643-3230.BCD-21-0010>)

Textual changes:

We added the sentence: *“The VAF cut-off for 15x sequenced cells and clones was 0.15, as mutations below this cut-off could be technical artifacts or mutations acquired in vitro.”* in lines 841-843.

Comment 3.2.4. We had some difficulty understanding exactly how mutational signatures were extracted from the data. While the authors used the somatic mutations found in the t-MN bulk samples from each patient, the level of methodological detail is not enough to fully understand how the data was used. In lines 740-741, the authors state that signature extraction was done using “the mutational matrix of all t-MN bulk substituted with the 34 healthy HSPC samples that were used to construct the baseline”. What does this mean? Also, mutational signatures were assigned to branches with seemingly dozens of mutations. How was the procedure in this case? Did the authors independently use the function “fit_to_signatures_strict” on each set of branch mutations using the same probability threshold as before? Is this reliable?

We agree that the sentence could have been clearer. Signature extraction becomes more robust when more samples are used for the extraction. To be sure that normal aging-related signatures could be extracted, we included mutation counts extracted from previously published WGS data of clonally expanded HSPCs of healthy individuals (de Kanter et al. 2021, <https://doi.org/10.1016/j.stem.2021.07.012>). We used HSPCs as these are thought to be the cells of origin of t-MN and as they have the same clock-like signatures. For clarity, “mutational matrices” are tables with the number of mutations of each of the 96 mutation types in the rows and samples in the columns. The mutational matrices of the bulk t-MN samples were concatenated to the mutational matrix of the healthy HSPC clones (*Rcbind*). The term “mutational matrix” was unnecessary for the method section and has now been removed. Code for reproducing the figures is uploaded with the rebuttal. In addition, we agree that a minimal number of mutations is needed for refitting. We now only show refitting results for branches (tree pie charts) or groups of branches (bar plots) of more than 100 mutations. We chose this number, as 93% of samples with more than 100 mutations have a cosine similarity of >0.85 of the reconstructed profile after refitting with the original profile, while only 14% of groups with equal or less than 100 mutations reach this similarity. See **Rebuttal Figure 5** below.

Rebuttal Figure 5. Average cosine similarities of reconstructed profiles compared to original mutational profiles for different cut-offs of number of mutations per branch. The numbers at the dots indicate the mutation number cut-off that was used.

Textual changes:

“As signature extraction becomes more robust when more samples are included, we performed signature extraction on the single base substitutions of all t-MN bulk samples together with previously published data of 34 healthy HSPC clones of healthy individuals of different ages to ensure that the normal clock-like signatures SBS1, SBS5, and HSPC could be reliably extracted²⁶.”, lines 927-931.

Comment 3.2.5. Mutations in each branch were assigned to different signatures to identify the processes operating during and after chemotherapy exposure. According to the authors, these mutations were fit to the signatures identified in the bulk t-MN samples. Given that the trees include other sample types (i.e., HSPCs), shouldn't the signatures be re-estimated, or do the authors assume that the active mutational processes should be the same across sample types? Also, how come the complete set of mutations in the HSPC clones in patients UPN008 and UPN034 are attributed to SBS31? Should not a subset of these mutations be attributed to other mutational processes? This should be clarified.

We agree with the reviewers that redoing the refitting on mutations from the phylogenetic trees is a better approach and have therefore now applied this. We identified mostly the same signatures as in the bulk t-MN, but interestingly, we now identified two additional signatures, both of which were mainly present in individual cells of *TP53*-deficient cells of Li-Fraumeni syndrome patients. We briefly discuss this result in the Results section, lines 269-272: *“Interestingly, in the private t-MN mutations, another signature (“SBS1”) is detected, which has a high contribution of T>C mutations. The presence of T>C mutations could be in line with exposure to alkylating agents³⁷ (Extended Data Figure 6i).”*, and lines 292-294: *“Similar to IBFM22, single cells had a high number of T>C mutations, indicating that these mutations are induced by alkylating drugs (signature “SBSH”, Extended Data Figure 6i).”*

Importantly, the conclusion we drew in our manuscript about the timing compared to platinum mutations is not altered by this change.

It is indeed surprising that all mutations in UPN008 and UPN034 are attributed to signature SBS31. It has been reported before that t-MN with platinum-related mutations have no correlation between their biological age, and the number of aging-related mutations (Pich et al 2021, <https://doi.org/10.1038/s41467-021-24858-3>). The cause for this has not been discovered. We have now added in the manuscript (lines 367-370): *“Interestingly, we found that all mutations in UPN008 and UPN034 could be attributed to platinum-exposure. This is in accordance with previous research that showed there is no correlation between the biological age of the patient and the number of clock-like signatures.”*

Comment 3.2.6. Structural variants were called with an in-house pipeline. Has this been published? Is it validated? How can a reader reproduce this analysis?

The pipeline that we have used for structural variant calling has been developed by the Hartwig Medical Foundation, which is not the institute that the authors are employed by. The pipeline is freely available (<https://github.com/hartwigmedical/hmftools>) and has been used extensively in published research by us, and others (e.g. <https://doi.org/10.3389/fonc.2022.919118>, <https://doi.org/10.1038/s41586-020-2080-8>, <https://doi.org/10.1038/s41591-020-1072-4>, <https://doi.org/10.1038/s41467-020-19406-4>). We have added the link to the github page in the Method section, lines 897-898: *“The Hartwig Medical Foundation’s gridss-purple-linx pipeline v1.3.2 (<https://github.com/hartwigmedical/hmftools>) was applied on the bulk t-MN blast samples and their paired normal to call somatic structural variants”*.

Comment 3.2.7. A tool called PTATO was used to call single-cell variants. As far as we know, this tool has yet to be peer-reviewed and published, so assessing its use is problematic. Indeed, multiple variant callers specifically developed for single-cell DNA sequencing (scDNA-seq) data (doi: 10.1016/j.csbj.2022.06.013) have been benchmarked and published that use sound statistical models to control WGA biases. While we understand that PTA has a different chemistry than other WGA strategies, we are unsure whether this is enough to justify using a novel single-cell variant caller that has not been benchmarked against available tools. Also, it is unclear how PTATO deals with ADO, as it apparently only considers false positives. Is this true? Authors should explain and justify their choice of this tool convincingly.

We thank the reviewers for raising this important matter. Previous statistical methods for filtering WGA biases have been based on other protocols and are not optimized for the specific type and relatively low number of artifacts that PTA can result in. PTATO has been peer-reviewed and published in Cell Genomics during the time of review of this article (see <https://doi.org/10.1016/j.xgen.2023.100389>). In this paper, the advantages of PTATO are described and the improvement over SCAN2, the previously preferred method for filtering PTA artifacts, were shown. Based on this published study, PTATO is the most reliable method currently available to analyze PTA-based single cell whole genome sequencing data and results in a low false-negative and false-positive rate.

We now implemented CellPhy for phylogenetic reconstruction which deals with allelic drop out (ADO) (see our answer to comment 3.3.1). We use the t-MN bulk sequencing to show

we miss very few clonal mutations in the tumor. See our answer to 4.3.1.

3. Phylogenetic analysis

Comment 3.3.1. We found the phylogenetic reconstruction strategy clumsy and often challenging to follow. It is also often mixed with variant calling and filtering when they are different things. After reading this section, we are unsure how these trees were built and whether the methodology was the same for all datasets. Filtering mutations, building trees, and mapping mutations to branches are different things that here seem mixed. We suggest authors explain in different sections distinct sets of steps that should be implemented sequentially: (1) call mutations using appropriate callers, (2) filter unreliable ones out and define the final datasets, (3) apply some of the available methods for phylogenetic reconstruction, (4) map mutations to branches, and (5) identify signatures in branches or clades.

We apologize for the confusion. Our approach was to find a set of high-quality mutations that without errors represented a tree. In this case, step (4) is unnecessary, as the mutations inherently form a tree, and mapping is unneeded. In the case that only clones are sequenced, and no single cells are included, the mutations normally form a tree with only a handful of mutations that do not fit, and that can normally clearly be identified as false positives by manual inspection, for example in IGV.

We agree that for data generated by single-cell WGA methods like PTA, the number of artifacts and ADO are higher, and this approach might not be the most appropriate. In addition, we agree that applying CellPhy is easier to reproduce. We therefore now apply CellPhy for all phylogenetic reconstruction and describe in the method section the steps in the appropriate order. We apply these steps.

1. We call mutations as described before, using GATK tools.
2. We use a general somatic mutation filtering approach, SMuRF (which we standardly use and published in the following studies, <https://doi.org/10.1016/j.stem.2021.07.012>, <https://doi.org/10.1158/2159-8290.CD-22-0120>, <https://doi.org/10.1158/2643-3230.BCD-21-0010> and others), to arrive at a set of mutations for clones and bulk t-MN samples.
 - 2.1 We use PTATO (validated and published by Middelkamp et al. 2023, <https://doi.org/10.1016/j.xgen.2023.100389>), which further filters SMuRF-output, to get a set of mutations for single PTA cells.
3. Per patient, we apply CellPhy for phylogenetic reconstruction on the combined set of mutations identified in step 2 and 2.1 in the cells and clones.
4. We map mutations to this tree with Cellphy/RXML-NG.
5. We use CellPhy to map mutations to the tree.
6. We filter mutations in end-branches, i.e., leaves of the tree that only have one sample.
7. We extract mutational signatures from the branches of all trees and perform mutational signature refitting.
8. We validate the tree using the VAF of the tree mutations in the bulk t-MN.

We apply step 6, as mutations mapped to end branches that have supporting reads for the alternative allele in more than two samples are very often mutations that are also assigned

to a shared branch in the trees, and often to multiple end branches, resulting in up to 11 assignments for one mutation (see **Rebuttal Figure 6a** below). For example, a mutation that is found in three cells is more likely to be true than an artifact in two out of those three cells (see Bohrsen et al. <https://doi.org/10.1038/s41588-019-0366-2>). Indeed, the mutational profile of end branches is distinct from the remaining mutations (**Rebuttal Figure 6b**).

The application of CellPhy resulted in longer shared branches with similar mutational signatures compared to what we previously identified, indicating that we previously underestimated the length of these branches, and that CellPhy is a better approach.

Importantly, the conclusions that we drew based on our analyses in the first version of the manuscript still hold, and are now validated with, among others, the bootstrapping from CellPhy (see comment 3.3.6).

Please see lines 863- 888 and 944-962 the Methods section for the restructured and adjusted description of this approach.

Rebuttal Figure 6. Filtering of mutations assigned by CellPhy to end branches

- The number of branches that mutations were assigned to. Only mutations that were assigned to at least one end-branch is indicated.
- The cosine similarity of mutations assigned to end branches by CellPhy. The mutations were divided in three categories. First, those assigned to multiple branches. Second, those assigned to one end branch, but with supporting reads in multiple cells. Last,

those assigned to an end branch and only that cell has supporting reads. Data from patient UPN034 is shown.

c) Similar to (b), but data from patient IBFM42 is shown.

Comment 3.3.2. The strategy for phylogenetic reconstruction here consists of arbitrary steps and multiple filters that are impossible to reproduce. The criterion or algorithm for phylogenetic reconstruction must be mentioned (parsimony, distance, likelihood, etc.). We only read, “A tree was constructed from the resulting mutations.”, which is not informative.

As explained in our response to comment 3.3.1, we have now implemented CellPhy to perform phylogenetic reconstruction. We now mention in the paper that CellPhy uses RXML-NG, a maximum likelihood framework for phylogenetic inference.

See lines 870-971 in the text: *“Phylogenetic reconstruction was performed using CellPhy v0.9.2, which utilizes RXML-NG, a maximum likelihood framework for phylogenetic inference.”*

Comment 3.3.3. Mutations and cells are added at different times without an evolutionary justification –no established phylogenetic method works this way. For example, it is unclear why (and how) the authors manually introduced the low-quality samples onto the phylogenetic trees. Most, if not all, phylogenetic methods allow for missing data, and we could not come up with a scenario here where adding tips to the tree a posteriori represents an advantage.

We now apply Cellphy for phylogenetic reconstruction, which removed many of the steps that could be less straightforward to reproduce, among which this step, and adjusted the Methods section accordingly.

Comment 3.3.4. Established methods exist to build phylogenetic trees from scDNA-seq data; some do it jointly with variant calling. Why did the authors not use well-known tools for phylogenetic tree reconstruction already benchmarked and based on explicit sound statistical models (like SiFit, SCIPHI, or CellPhy)?

We thank the reviewers for this suggestion. We now implement Cellphy as we agree that applying this method is better reproducible than the steps that were previously applied to build the trees. See comments above.

Comment 3.3.5. The trees are rooted, but no explanation of how rooting was performed is offered. Rooting is a crucial aspect of phylogenetic reconstruction, also very relevant here, and should be explained in detail.

We apologize for the lack of information regarding this matter in the previous version of the manuscript. We now root the CellPhy tree using the same function as the R script in the CellPhy package (*treeio::root*, which uses *ape::root* under the hood). We use the WGS data of the mesenchymal stromal cells (MSCs) for rooting in the 6 cases that they are available. In one case MSCs are not available due to contamination with t-MN blasts (UPN008). In this case, we use an HSPC that does not share any mutations with any other cells for rooting. In **Rebuttal Figure 7**, we now show for all trees that the expected cells share the mutations

and support the order of the tree. This process is explained in the method section lines 880-882: “The trees were rooted with treeio’s “root” function, using the MSCs as the outgroup, or an HSPC which shared no mutations with the other samples, when MSCs were unavailable (UPN008).”.

Rebuttal Figure 7. Heatmaps of variant allele frequencies of mutations in the trees.

Comment 3.3.6. Conclusions based on trees can only be as good as the trees themselves. We will always get a tree regardless of the input data. However, the derived conclusions are only reliable if the data support the tree. While the phylogenetic bootstrap is not a p-value, it does help to understand whether the data supports a given branch. Authors should compute and display bootstrap values (or another measure of phylogenetic support) and base their conclusions only on well-supported branches. Bootstrap calculations are straightforward, using multiple software for phylogenetic reconstruction, including R packages.

We thank the reviewers for this comment, as we agree that this approach gives additional confidence in the shape of the trees. We now implement CellPhy and its bootstrap function, to show that most clades in the tree have a high bootstrapping recurrence, and that 16/29 splits after the clonal t-MN branches in the tree are found in 100 out of 100 bootstraps as are many other splits in the tree. Most other splits are found in >50/100 bootstraps. We now include the bootstrapping output in Figure 3 and 4.

Comment 3.3.7. Which method was used to assign the mutations to each tree? Does each mutation occur only once in the phylogenies? Statistical methods exist to accomplish this for scDNA-seq data (e.g., 10.1186/s13059-021-02583-w)

We now use “mutmap” from the CellPhy code to map mutations to the tree as suggested by the reviewers. Please see our response to comment 3.3.1 for more details. Lines 874-876 *“CellPhy was also used to map mutations to the tree (using the “--mutmap” function) with the “--opt-branches off” setting.”*

Comment 3.3.8. We did not understand the role of the MSC samples during variant calling or filtering. For UPN008 and UPN034, why are mutations subclonal in the “MSC control” included? What is this control? Why are mutations subclonal in other samples excluded? Why mutations absent in one sample were included? We do not follow all this. Please define “mutation” (e.g., present in x but absent in y), and explain the rationale of every filter.

We have simplified the inclusion of mutations in building the tree. For completeness, we explain the rationale of all steps we took, and indicate if a step is now excluded.

- The “definition” of a somatic mutation depended on the presence of an MSC control sample.
 - o In the case that uncontaminated MSCs were sequenced, we defined a somatic mutation as any that was clonally present in a cell/clone, but not clonally present in MSCs. MSCs are more distantly related to blood cells than blood cells are among each other, making this population a good germline control. The reason to include subclonal mutations (VAF < 0.3) in MSCs (sequenced 30x), is that these could be very early mutations that arose in utero, and that could be mosaic in bulk MSCs and clonally present in many (blood) cells. These might be essential, in the case that a t-MN is driven by a very early mutation. The used of MSCs was previously published: Osorio et al. 2018 <https://doi.org/10.1016/j.celrep.2018.11.014>, Brandsma et al. 2021 <https://doi.org/10.1158/2643-3230.BCD-21-0010>, Bertrums et al. 2022

<https://doi.org/10.1158/2159-8290.CD-22-0120>, de Kanter et al.

<https://doi.org/10.1016/j.stem.2021.07.012>. However, as we have not identified such mutations in our analyses in this manuscript, we have excluded this step from our methods.

- In contrast, we excluded mutations found with a VAF lower than 0.15 from any cell/clone (sequenced 15x) as these are more likely technical artifacts and could not be confidently placed in the tree. The term “subclonal” for these mutations is therefore incorrect and confusing, as the sequencing depth and VAF cut-off was different from subclonal mutations. We now excluded this filter, as CellPhy uses genotype likelihoods to estimate where in the tree a mutation should be mapped.
- In case of patient UPN008, for which only clones and no cells were sequenced, and for who the MSCs were contaminated with t-MN cells, we needed another method to exclude germline mutations. If for a certain mutation there were reads present in each clone that support the alternative allele, the chances were high that this mutation is germline. Therefore, we only considered mutations to be somatic if they were confidently absent in at least one clone. For clarity, such an approach is only feasible if only clones are sequenced. Due to allelic drop out after direct WGA (PTA-based sequencing data) we need an uncontaminated MSC control to filter out germline mutations.

We state in the text:

Lines 219-220: *“The HSPCs are used to check if the same mutational processes are active in normal cells.”*

Lines 107-109: *“We performed WGS on bulk t-MN blasts and used mesenchymal stromal cells (MSCs) or bulk-sorted B-cells of the same patient as a germline control (Methods, Extended Data Fig. 2, 3).”*

Comment 3.3.9. There is an arbitrary manual filter for ADO using hetSNPs. However, models and tools exist to do this in a sound statistical way (mentioned above). Why not use them? Still, this explanation should be in the variant calling section.

We now use CellPhy to reconstruct the trees. CellPhy uses a RXML-NG with an optimized model for the ADO and FP rates associated with single-cell WGA and subsequent WGS. We explain this in the methods section, lines 871-873 *“CellPhy considers the allelic dropout rate and amplification errors per sample and constructs the most likely tree based on these estimates”*.

3.4. Interpreting inpatient clonal evolution

Comment 3.4.1 These are (very) small samples of a much larger population. Therefore, the most recent common ancestor (MRCA) of the t-MN cells will never be the initially transformed t-MN cell, as the authors suggest when labeling this node. This needs to be corrected. The t-MN MRCA will be younger than the origin of the t-MN. This also means that mutations shared between all the sampled t-MN blasts did not necessarily accumulate before the expansion of the initial leukemic cell. Many t-MN lineages could go unsampled, so t-MN mutations might look clonal when they are subclonal or look private when in

reality, they are subclonal. This should be considered when discussing if the expansion happened during or after treatment. The inference's scope here concerns the *sampled* t-MN clones, not the initial t-MN clone.

It is possible that cells go unsampled with a low number of cells analyzed. We now reconstruct the trees using only the single cells and clones and use the bulk t-MN WGS data to validate the tree and show that no major lineages are missed. These bulk t-MN samples were purified using FACS to only contain blast cells. These analyses revealed that no major populations were missed.

- 1) The mutations in the branch shared between all single t-MN blasts have a mean VAF of 0.5 with an expected distribution, and no enrichment in mutations with a lower VAF. This confirms that this shared branch does not represent a single subclone (which would share many mutations with a lower VAF) (**Extended Data Figure 6a, b**).
- 2) In the trees of IBFM14, IBFM32, IBFM42, and UPN008, no more than 17 mutations with a VAF higher than 0.3 are found in the bulk t-MN, that are not part of the tree. This means that the first branching point in the tree is likely a good representation of the biological MCRA.
 - a. In IBFM32, a *KRAS* driver mutation (p.G12V) is part of the 6 mutations that are not found in the tree. In the initial submission this was visualized in a separate branch, but is now only indicated in the text, lines 243-245: *“Of note, in IBFM32, a KRAS pG12V mutation was present in the bulk t-MN at high VAF, that was not present in the tree, indicating that a t-MN clone went unsampled (Extended Data Fig. 6c).”*
 - b. In patient IBFM22, 789 mutations with a VAF ≥ 0.15 are not part of the tree (set 1). These mutations have a mean VAF of 0.32 in the bulk t-MN (**Extended data Figure 6c**). This is the same as the subclonal mutations only found in cell 2F10 (set 2). These mutations have an average VAF of 0.29 in the bulk t-MN (**Extended data Figure 6c, e**). The coverage of most mutations of the former set (missed in the tree) was 0 in cell 2F10 (**Extended data Figure 6f**). Based on the VAF of germline mutation near the mutations that were not found in 2F10 but with coverage in 2F10, only one of the two alleles is amplified in these areas (**Extended data Figure 6h**). Finally, the mutational profiles of these mutations were the same (**Extended data Figure 6g**). This indicates that 2F10 is part of a major subclone of ~60% of the t-MN blasts and that both sets of mutations are part of this subclone but are not all covered in 2F10.
- 3) The subclonal mutations that are shared between a subset of the single t-MN blasts have a VAF lower than 0.5. The mutations found in individual t-MN blasts, when detected in the t-MN bulk, have a very low VAF in the bulk t-MN. These are likely subclonal, and not private. We now categorized these as “subclonal” in all figures. Of course, it is possible that some of the mutations labeled ‘private’ are subclonal, but the percentage of t-MN cells that carry them is low, and therefore the chance that they occurred very late in the t-MN development is high, making the term “private” for relative timing still an appropriate term.

With this evidence in mind, we are certain the vast majority of the private, subclonal, and clonal mutations are labeled appropriately, and we keep using these terms. We now mention in the text (lines 220-223): *“We validated the shape of the tree, the clonal, subclonal, and private labels for branches, and the most common recent ancestor of the bulk*

t-MN by investigating the VAF of the tree mutations in the matching bulk t-MN data and by investigating mutations present in the bulk t-MN, but not the tree (Extended Data Fig. 6)."

Extended Data Figure 6. Validation of the phylogenetic trees using WGS of bulk t-MN.

- a) The t-MN trees shown in Figure 3 and 4. The number below each branch indicates the percentage of the mutations in the branch that is found in any read in the bulk t-MN WGS data. The number above the branch indicates the average variant allele frequency (VAF) of the mutations that are at least supported by one read in the bulk t-MN WGS data. The color of the dot corresponds with this value.
- b) The t-MN bulk VAF distribution of the mutations in the trees in Figure 3 and 4, grouped by category.
- c) The t-MN bulk VAF of the mutations not found in the phylogenetic trees in Figure 3 and 4. The numbers indicate the number of mutations between 0.15 and 0.3 and above 0.3. “*KRAS*” indicates a *KRAS* p.G12V mutation in IBFM32.
- d) The mutational profiles of the mutations with a VAF higher than 0.15 from (a).
- e) The IBFM22 bulk t-MN VAF of mutations not identified in the corresponding tree (similar to (a)) and of the mutations found in the subclonal/private or cell 2F10.
- f) The sequencing depth in cell 2F10 of the same mutation sets as in (c).
- g) The cosine similarity of the 96-trinucleotide mutational profiles of the two sets of mutations shown in (c) and (d).
- h) Examples of mutations that are found in bulk t-MN, but not present in any cell of the tree (“non-tree” in (c) and (d)) and have reads in cell 2F10 that cover the mutation site, but for which no reads are present in 2F10 that support the mutation (the blue peak between 1 and 2 in panel f). Germline mutations in close proximity indicate that in most cases only one of the two alleles was amplified in 2F10, in most cases the reference allele of the somatic mutation site.
- i) The 96-trinucleotide mutational profiles of the mutational signatures identified in the phylogenetic trees, SBSH and SBSI.

Comment 3.4.2. Several trees (UPN008, IBFM22, UPN034) have a big basal polytomy and several short interior branches. This indicates a low phylogenetic signal in the data and is very important when interpreting trees. Once branch support is established (e.g., bootstrap values), authors should avoid making firm conclusions from weakly supported branches.

We agree with the reviewer that we should not make any firm conclusions about the order of branch points at places in the tree where there is low confidence in this order. Now that we use CellPhy and did bootstrapping, we can see that the order of the tree is often supported in >50/100 bootstraps. However, we do not use the order of these polytomies to draw any conclusions, other than that healthy HSCPs without drivers in many trees do not share any mutations with each other. This is rooted in biology and will inherently result in polytomies in the trees but is still indicative of a very early most recent common ancestor between these healthy blood stem cells and the t-MN initiating cell (see the heatmaps in 3.3.5). As explained in the reply to comment 3.4.1, the labels of the trees are validated by the bulk t-MN sequencing.

Comment 3.4.3. There is no temporal information in these trees. Hence, the temporal marks (ticks and length and location of black triangles) in Figure 3 b,d,f, and Figure 4 b,d,f of the origin and expansion of the t-MN are arbitrary, as authors do not really know the absolute timing of the expansion. Please note that the number of mutations is only directly related to time in a strict molecular clock model.

We have now only indicated the moments with a known temporal difference (e.g., the length between the date of diagnosis of the primary and secondary cancer, or the length of treatment if available) with ticks and the moment of expansion relative to the treatment (as deduced from the signatures in the different branches, not the branch lengths) as a dotted box. The black triangles are now always in the middle of these boxes. Finally, we emphasized in the legend that all of these are schematics without a known exact moment of expansion.

Comment 3.4.4. “Punctuated evolution” is a concept that comes from paleontology to explain the discontinuity of the fossil record and is about *phenotypic* change, with long periods of stasis followed by periods of apparently rapid speciation –in fact, this is known in evolutionary biology as “punctuated equilibrium.” For some unknown reason, it has been loosely adopted by the cancer genomics community. We should not forget that selection targets phenotypes, not genotypes. In molecular evolution, we often see changes in the substitution rate among lineages, particularly when comparing distant species, and this is simply described as rate variation among lineages, which can be due to various factors. As simple as that. There is no need to invoke macroevolutionary theories of phenotypic change among species that do not apply in this scenario. There is nothing remarkable about the rates of evolution in patients UPN008, IBFM32, or IBFM42. A long clonal branch and short subclonal branches are not unexpected, considering the small sample sizes and precisely the effect of chemotherapy. Also, please note reliable rooting is essential to trust the branch lengths. We suggest removing the term “punctuated evolution” here and in the discussion.

We thank the reviewer for pointing out this confusing terminology. We have removed “punctuated” evolution from both parts of our manuscript. Despite the small sample sizes, we do believe the mutational signatures in the different branches of the trees can tell us something about the evolution of the t-MN, and have aimed to describe this more objectively, taking into account that the sampling will never represent all cells in the bone marrow of the patient.

Comment 3.4.5. The idea that platinum exposure inhibits the expansion of the initial leukemic clone and that this expansion thus starts when the exposure to platinum ends is based only on three patients, or are there more supporting this idea? How were the patients shown selected? If there are more trees, we suggest showing them as supplementary material.

We did show all trees that we could generate for patients after platinum treatment. We now added an additional tree of an additionally acquired sample, IBFM67, as we described in the reply to comment 3.1.4.

Comment 3.4.6. For patient IBFM42, the pattern observed is unusual, as signature SBS31 seems on and off; if we trust the order of the nodes –which most likely we cannot, bootstrap will tell if this is the case. Please reconsider interpreting this case based only on well-supported nodes.

We thank the reviewer for raising this concern. With bootstrapping, we show that the order of the nodes is highly robust, with all of the nodes being supported in all 100 bootstraps. There is indeed one individual branch that has 10% of mutations refitted to SBS31. However, when we inspect the 96-trinucleotide profile of this branch, the characteristic peaks of SBS31 are lacking (**Rebuttal Figure 8a**). Indeed, when we perform bootstrapped refitting using the MutationalPatterns R package, we see that SBS31 is identified in the clonal branch in 100% of bootstraps, while SBS31 is only identified in 40% of bootstraps in the individual branch (**Rebuttal Figure 8b**). In addition, while the C[C>T]C and C[C>T]T mutations of the clonal branch are enrichment in the transcribed compared to the untranscribed strand of genes, which is characteristic for SBS31, the mutations of the individual branch do not show this enrichment (**Rebuttal Figure 8c**). We thus conclude that SBS31 is incorrectly refitted to this branch. We now mention the bootstrap values in the legend, lines 468-470 *“SBS31 contribution to the clonal branch was supported by 100/100 bootstraps. In the private branch, SBS31 was found in 40/100 bootstraps.”*

Rebuttal Figure 8. SBS31 mutations in the "sbs31+" end branch of phylogenetic tree of patient IBFM42.

- The mutational profile of the main clonal branch, and the t-MN end branch with a low number of refitted SBS31 mutations, termed 'individual "sbs31+" branch'.
- The bootstrapped refitting results for the mutations of the two branches shown in a.
- A 192-mutational profile indicating the transcription strand bias of the mutations shown in (a) A bias towards the transcribed strand is typical for SBS31 (platinum) mutations, but not for clock-like mutations.

Comment 3.4.7. For patient IBFM32, the t-MB blasts do not have a common origin. How is that possible? Again this might be due to phylogenetic uncertainty and lack of support. Some measure of support is necessary to modulate the interpretation. Also, the black diamond in this patient indicates, according to the authors, the clonal mutations in the t-MN bulk that were not found in any of the sequenced single blasts. Given this statement, did the authors create an artificial clone with selected mutations? Most likely, such a clone never existed, which might distort the phylogenetic reconstruction. What is the justification for this? Strikingly, if we look at the tree, the branch leading to this clone seems to have zero length, indicating no private mutations. Or is there just a tiny branch there? Please clarify all the rationale.

We now include CellPhy's bootstrap function to show the confidence of the tree. Indeed, the two branches that are shared between all t-MN blasts of IBFM32 are found in all 100 bootstraps. The t-MN blasts do have a common origin. We think that the reason that this information is hard to interpret may be to the labels at the tips of the tree. The annotation of "t-MN blast" or "HSPC" clone/cell was made based on cell phenotype (i.e. the cell surface markers the cell expressed as measured by FACS). Sometimes, cells that are phenotypically characterized as HSPCs, by CD34-positivity, CD38-negativity and negativity of blast-specific markers, such as CD33, genetically do carry most, or all, of the driver mutations found in the bulk t-MN blasts. We now mention this more clearly in the text, lines 223-227: *"Of note, HSPCs and t-MN cells were sorted and labeled in the manuscript based on their immunophenotype. In four patients some of the immunophenotypically HSPC-like cells shared all the t-MN drivers and other clonal mutations with the bulk t-MN blasts and were thus in the phylogenetic analysis considered t-MN blasts."*

In addition, we now only use the single cells/clones to reconstruct the tree and use the bulk t-MN blast for validation (see our answer to 3.4.1). As explained in 3.4.1, there are 6 mutations, including a hotspot *KRAS* mutation, found in the bulk t-MN with a high VAF, that are not identified in any of the single t-MN cells. Therefore, the branch is not visible in the tree. We assume that these mutations represent a missed branch that was not sampled. We now excluded these mutations in the clonal branch, and removed the separate branch in the tree that we did show in the previous version of the manuscript. We do mention the mutations in the text and have added them to the Extended Data. Lines (243-245): *"Of note, in IBFM32, a *KRAS* pG12V mutation was present in the bulk t-MN at high VAF, that was not present in the tree, indicating that a t-MN clone went unsampled (Extended Data Fig. 6c)."*

Comment 3.4.8. For patient IBFM22, signature SBS1 does not appear in the branch leading to the t-MN clade, which the authors interpret as the result of an early expansion during treatment. This might be the case, but the other option is that the t-MN was not the result of the treatment. Have the authors considered this, or does this idea make no sense?

The lack of treatment-induced mutations is indeed an intriguing observation. This means that the first detected division was preceded by no, or little, treatment-induced mutations. The following branches in the tree all show a high contribution of platinum-associated mutations. From this, we can conclude that if the expansion started during treatment, it was very early during treatment. As the reviewers suggest, another option is that the expansion

started before treatment. From our observation that in UPN034 the *TP53* wild-type allele was already lost in 40% of the blood before treatment, it seems likely, or at least possible, that in IBFM22, the precursor cell of the t-MN already divided before the start of treatment. However, both of these possibilities stand apart from our conclusion that at least part of the expansion happened during treatment. This can be concluded from the high number of SBS mutations in the subclonal, and private branches. This is something that was not observed in all our samples of *TP53* wild-type t-MN (n=5). What the role of the treatment is in the expansion is difficult to determine, but that the t-MN did divide during treatment can be concluded with very high certainty, and this conclusion is what we use in the manuscript to further build on.

Comment 3.4.9. For UPN034, there are no t-MN blasts, so should we guess t-MN is equivalent to CH? Authors depict an early expansion during cisplatin treatment that declines or stops before FU. Why?

We thank the reviewer for this question. Indeed, we have no blasts available for UPN034, since this sample contained a very low blast percentage. In this sample, we mainly looked at the effect of chemotherapy on the normal HSPCs at time of t-MN and follow-up. We see that there is CH, and that the cell that gave rise to the CH divided during treatment, similar to what we found in *TP53*-mutated t-MN blasts. As both CH and t-MN arise from a normal blood stem cell that acquires driver aberrations, we can assume that the trajectory of CH and t-MN development are similar. The data of this patient does confirm that cells with *TP53* deficiency are able to divide during platinum treatment. In addition, the start of the expansion has now been drawn in the middle of the period between the primary diagnosis, and t-MN detection. The decline between t-MN diagnosis and the follow-up sample was drawn because 0% blasts was detected in the follow-up sample. We now state in the legend lines 488-489: *“In the schematic, the evolution of the t-MN, not the CH, is drawn. 0% blasts were detected in the FU sample by diagnostic MRD measurements.”*

Comment 3.4.10. Importantly, for patient UPN034, one cannot confidently say there were four independent *TP53* losses because all lineages involved originate in a polytomy. A polytomy here means uncertainty about the phylogenetic relationships among these clones, so depending on the true but unknown evolutionary relationships among them, the observed losses could be explained as one, two, three, or four independent events. We simply cannot tell the exact number because of the polytomy, so there is no clear evidence of convergent evolution, and this tree cannot be used to support such a hypothesis in the Discussion.

The tree indeed shows a polytomy and strictly speaking nothing can be concluded about the 17p losses based on the tree. However, the tree was constructed from SNV information only. Later, the tree was annotated with the CNV/SV information. When we look into the output of the SV and CNV pipeline, we can see that the five cells with a shared branch in the SNV tree share **four** breakpoints on chromosome 17, two of which are fusion events with chromosome 5. However, when we look at the other three cells with a loss of 17p, we can observe that none of their breakpoints overlap (see **Extended Data Figure 8 at R1.10**). Therefore, we can state with certainty that these were three independent events. This information was not in the manuscript. We now state in the Results section that the

breakpoints do not overlap, lines 303-305 “*Notably, in total we identified four independent events, based on unique breakpoints of the chromosome arm losses, by which individual HSPC clones in this patient lost their TP53 wild-type allele (Extended Data Fig. 8c-e).*”

Comment 3.4.11. The tree of patient IBFM14 is very unresolved and will likely have very little support, so that no reliable inferences might be derived from it. As mentioned, the fact that some branches are longer than others has nothing to do with punctuated evolution. This can be natural rate variation among lineages due to coalescent sampling, and for cells might also be explained by different breadth, coverage, and error rates during WGA and scDNA-seq.

As the tree is rooted at the MSC control, no bootstrap information can be given about the main clonal branch. However, as this branch contains 2210 SNVs, 1659 of which are found in all 7 cells, and 439 are only missed in one, the existence in this branch is clear (also see the heatmaps at comment 3.3.5). The first subsequent node is supported by 100 of the bootstraps, but the others are found in 38 to 75 iterations. As stated in our response to comment 3.4.4, we have removed any reference to punctuated or neutral evolution. In the case of IBFM14, the conclusion about a short period of expansion stems from the fact that the subclonal nodes cannot be supported by more than a few mutations. The sequencing quality and coverage were good in all t-MN cells; thus, the low number of shared mutations is very unlikely to be caused by ADO. In other words, we think that it is very unlikely that the true number of shared mutations is higher than what we have identified. Under normal circumstances, HSPCs accumulate around 14-16 mutations per year, and divide only once in every 40 weeks (<https://doi.org/10.1158/2159-8290.Cd-22-0120>, <https://doi.org/10.1038/s41586-022-04786-y>). Assuming that mutation rates will never be much lower than this rate, we can state that these divisions most likely occurred within a few months at the most, and are very unlikely to have happened over many years. We can therefore still conclude that the expansion of the t-MN that gave rise to the population of cells present at the time of diagnosis occurred in a relatively short period of time, and we remain with the statements that we made before.

5. Discussion

Comment 3.5.1. The model proposed in the second paragraph of the discussion implies that t-MN tumors have lost all their wild-type TP53 copies. This should be easy to check from available bulk data. Is this the case?

In Schwartz et al. 2021 Nat Comm (<https://doi.org/10.1038/s41467-021-21255-8>), 13 out of 15 patients (87%) had both a TP53 mutation (somatic, germline, or mosaic) and a CN-LOH or deletion event (Schwartz et al. Supplementary Figure 3e). This study is the best resource for WGS of pediatric t-MN next to our own dataset and contains more TP53-mutated cases. So, we can confirm that this is the case, and we now mentioned this in lines 326-328 “*In line with this, 13 out of 15 t-MN from Schwartz et al. that harbored a TP53 alteration, had both a mutation and a copy number loss of this gene¹⁶.*”

Comment 3.5.2. Please remove references to punctuated evolution or neutral evolution. The former does not apply in this context, and the latter cannot be inferred just by looking

at tree shapes. And please do not forget these trees are sample genealogies, so their shape would change if you select other cells from the exact location again. A sample genealogy is related to but is not the same as the tumor history. Sampling error (coalescent) plays a role.

Please see our answer to comment 3.4.4, which addresses the same subject. We have removed references to punctuated/neutral evolution from our manuscript.

We agree with the reviewers that looking at sample genealogy is not the same as knowing the exact tumor history; however, we do believe that by using phylogenetic trees we can come closer to a model of tumor evolution in our patient samples. We do acknowledge that we only sequenced a subset of the cells that are involved in the t-MNs of our patients, and we can therefore not derive a complete tumor history. However, these cells do add direction to our conclusions regarding the time of clonal expansion with regards to the treatment exposure, certainly now that we validated the trees using the bulk t-MN data (see 3.4.1).

*** Specific comments

Comment 3.6.1

200: We guess the authors mean “Fig. 2e”.

We thank the reviewers for noticing this mistake. We have corrected the text accordingly.

Comment 3.6.2

251: “.. the developmental trajectory in this case is indicative for a neutral or more evolution”. This statement has no basis. It is impossible to tell whether evolution is neutral just by looking at the tree.

Please see our answers to comments 3.4.4 and 3.5.2, which address the same subject. We have removed this statement from the current version of the manuscript.

Comment 3.6.2

276: “Since we were not able to perform WGS on the bulk t-MN blasts, we compared the HSPC WGS data with diagnostic data from single nucleotide polymorphism (SNP) and karyotype assays.” We do not understand the meaning of this statement in this context.

We apologize for being unclear with this statement. What we aimed to say with this sentence is that we were not able to perform WGS on the bulk t-MN blasts from this sample ourselves, as the blast percentage was very low. Therefore, we asked for the data that was available at our diagnostics department, which were SNP and karyotype assays. We then compared our generated WGS data from the HSPCs that we sequenced with the diagnostic data of the t-MN blasts. We aimed to clarify this in the edited version of the manuscript (lines 297-300): *“Next, we evaluated if the HSPC clones of UPN034 were related to the t-MN. As we were not able to sort and thus perform WGS of the t-MN blasts, we compared the single base substitutions and CNVs found in the HSPC WGS data with diagnostic data from single nucleotide polymorphism (SNP) and karyotype assays of the t-MN.”*

Comment 3.6.3

290: At least in our pdf, there is here a stranded “Next,”
We thank the reviewers for noticing, and have removed this from the manuscript.

Comment 3.6.4

697: Mutation calling was performed using GATK for which datasets? Also, which of the different GATK tools was used (Haplotypecaller? or Mutect2?)

We apologize for the incomplete information. We now clarify that we use Haplotypecaller and apply the pipeline on all samples of each patient combined, resulting in a multi-sample VCF *“Mutation calling was performed with GATK’s HaploTypeCaller, on all samples of a patient combined.”* (line 807-808). We also state (line 830-831): *“A full pipeline description is available at www.github.com/UMCUGenetics/NF-IAP.”*

Comment 3.6.5

717: SMuRF is a method that filters (GATK) variant calls, in this case, to identify clonal mutations in bulk t-MN samples (also HSPC clones), which have low purity, right? This should be made explicit.

As stated in the Methods, we sort t-MN cells in bulk using fluorescence-activated cell sorting (FACS) to obtain (near) pure samples. For more details on purities, we refer to our answer to comment 3.2.2. We apply SMuRF because the set of mutations outputted by GATK is not sufficiently filtered and contains low-quality mutations and artifacts.

Comment 3.6.6

726: “In these cases, a mutation was filtered if”. We guess it meant “a mutation was kept if.”

Mutations that are (highly likely) germline mutations are filtered out. This process was described in this sentence. If a mutation is clonal in all cells with high sequencing quality, it is considered a germline mutation and is excluded from the final set. For clarity, we changed “filtered” into “excluded” to prevent confusion and extended the sentence. It now reads: *“In these cases, a mutation identified by SMuRF to be in the bulk t-MN was excluded if it was (a) [...]”* lines 852-854.

Comment 3.6.7

793: For IBFM22, IBFM32, and IBFM42, we are unsure about the final set of mutations used to build the tree. Is it the overlap among PTATO in single cells and GATK-SMuRF in clones and bulk?

We now clarified the steps of building the trees in the methods and explained the process applied in the current version of the manuscript in our answer to comment 3.3.1. In addition, we now state in the methods: *“CellPhy was run on all clones and cells of a patient. The mutation set that was used consisted of the mutations in the clones (from SMuRF) and cells (from PTATO), supplemented with the mutations in the bulk t-MN (from SMuRF).”*, lines 864-866.

Comment 3.6.8

804: “A mutation was filtered out if more than 50% of germline variants had a VAF that differed more than 0.2 from the VAF in the MSC bulk in at least one sample.” We do not understand what this means and where the 0.2 comes from. We assume that the first VAF is in the single cells and that mutations become missing data on a cell-per-cell basis.

Now that we implemented CellPhy, this step has been removed from our methods. We have adjusted the methods section accordingly in the new version of our manuscript.

Comment 3.6.9

809: “For both approaches, all shared mutations of branches with 10 or fewer mutations were visually inspected in IGV, and false positives were excluded.” What are false positives? Shared mutations among cells assigned to branches with a total number of assigned mutations less than 11? Why less than 11?

Now that we implemented CellPhy, this step has been removed.

Comment 3.6.10

810: “A tree was constructed from the resulting mutations.” Which mutations? We are pretty lost at this point.

Now that we implemented CellPhy, the process of reconstructing the tree and mapping and filtering mutations has been rewritten in the methods and clarified in our answer to comment 3.3.1 and 3.6.7.

Comment 3.6.11

811: “Then, previously excluded mutations that exactly fitted the tree were added back.” What is to fit? Please explain and justify.

Now that we implemented CellPhy, this step has been removed. Please see comment 3.3.1.

Comment 3.6.12

812: “IBFM42, mutations that did not fit in the tree exactly were still added to the tree if they uniquely fitted to a single branch, assuming that they were missed in one single PTA-amplified cell.” So here, the authors sound as if to be manually calling ADOs, assuming reversals are not possible, right? How does this link with the het-SNP approach?

Now that we implemented CellPhy, this step has been removed. Please see comment 3.3.1.

Comment 3.6.13

814: “For low-quality samples in which 15% or more of the genome was not sufficiently covered, for each branch, the mean VAF of the mutations with sufficient coverage (>10) was calculated” A VAF from 11 reads is not reliable

Now that we implemented CellPhy, this step has been removed. Please see comment 3.3.1.

Comment 3.6.13

816: “A cell was assigned to a shared branch if this average VAF was higher than 0.45.”. How is a cell assigned to a shared branch? We do not understand what this means. The mean VAF was estimated from which sample exactly? We are guessing that it must come from the bulk t-MN, but we don’t follow the rationale behind this approach and wonder whether it is justified.

Now that we implemented CellPhy, this step has been removed. Please see comment 3.3.1.

Comment 3.6.14

817: “For branches with a lower average VAF and end-branches, only the truly shared mutations were considered, and other mutations were kept as single-cell mutations (end branches).” What are truly shared mutations?

Now that we implemented CellPhy, this step has been removed. Please see comment 3.3.1.

Comment 3.6.15

Phylogenetic reconstruction: We assume cells are assigned a lot of missing genotypes. How much? Please describe in detail.

We now use CellPhy to reconstruct tree and map mutations to branches. Private end branches by definition only contained one cell, and the mutations assigned to that branch were found in that cell (see previous answers, mainly 3.3.1, for details). Of all 11201 mutations assigned to a shared branch, 7466 (67%) were detected in all the cells that were part of that branch, 3045 (27.2%) were not present in one cell (i.e., this cell had no reads that supported the alternative allele) and 690 (6.2%) were not present in two or more cells. In other words, in 61984 cases a mutation in a cell was assigned to a shared branch. Of these, 4686 (7.6%) had a missing genotype (no read with that supported the alternative allele was found in that cell). As expected, this was higher for trees with PTA cells (8.4%) compared to the two trees with only expanded clones (0.9%). **Rebuttal Figure 9** (below) visualizes the number of missing genotypes, and additional genotypes (a cell has one or more reads supporting the alternative allele but is not part of the branch to which that mutation was assigned). In addition, we now mention the following in the Methods (lines 883-884): “*In all cells and all shared branches combined, 7.6% of mutations had a missing genotype.*”

Rebuttal Figure 9. Missing and additional genotypes in the phylogenetic trees.

Each bar in a plot represents the mutations in one branch of a tree (divided by patient). Each mutation is colored based on two factors; first the number of additional genotypes (cells that had one or multiple reads supporting the alternative allele, but were not part of this branch), and second the number of missing genotypes (cells part of that branch, but for which no reads supported the alternative allele of the mutation).

Comment 3.6.16

Phylogenetic reconstruction: Only clonal mutations should be used throughout for expanded clones. Please confirm.

We can confirm that we only use clonal mutations. This has been answered in our response to comment 3.2.2 and 3.2.3.

Comment 3.6.17

Phylogenetic reconstruction: For the bulk t-MN, how is/are the clonal sequence/s used for phylogenetic reconstruction determined? (see, for example, patient IBFM22)

We now only use the bulk t-MN data to validate the tree and to determine the most common recent ancestor of the sequenced bulk t-MN cells and do not include the bulk mutation data in the tree itself. For a more detailed explanation please see our answer to comment 3.4.1.

Comment 3.6.18

Figure 3a: For UPN008, we do not see any t-MN data in the tree. Maybe this is a mistake, and the six HSPC clones (white squares) forming a clade are t-MN blasts and should be black triangles. Or are these immunophenotypically HSPC-like cells that shared all the t-MN drivers and other clonal mutations with bulk t-MN blasts? If this case, they should be relabeled as blasts in the tree.

We discuss the annotation of the tips of the trees in our answer to comment 3.4.7. In short, the HSPCs that are used for the “HSPC clones” and “single HSPC” are annotated based on their immunophenotype. The cells express cell surface markers characteristic of HSPCs, while lacking the canonical or patient specific t-MN blast markers, but still harbor all genetic t-MN drivers. Thus, genetically we could call them blast. We validate this by using the VAF of the tree mutation in the bulk t-MN. Please see our response to comment 3.4.1. This is mentioned in the manuscript lines 225-227: *“In four patients some of the immunophenotypically HSPC-like cells shared all the t-MN drivers and other clonal mutations with the bulk t-MN blasts and were thus in the phylogenetic analysis considered t-MN blasts.”* and indicated in the legends of Figure 3 and 4 that the labels are for the cell immunophenotype.

Comment 3.6.19

Figure 3b,f: where does the number of years come from? How did you date the expansion?

We would like to refer to our answer to comment 3.4.3, where the same issue is raised. In short, we now indicated more clearly in the panels that the exact moment of the expansion is unknown. The only exact data that we have is the time of sampling of the DX1, DX2 and FU samples. Time of treatment from treatment history was also added, as were the ranges of treatment time based on treatment protocols, if exact times were not known. These times are annotated in the figures accordingly.

Comment 3.6.20

Figure 3 legend: “Phylogenetic lineage tree” should be “Phylogenetic tree.” “Phylo” and “lineage” are the same thing.

We now removed the word “lineage” in the legend.

We have made many comments, hoping to increase precision and reproducibility. The general conclusions should be acceptable because of the cell line experiment. However, there is room to improve the methodology and the interpretation of the evolutionary analyses. Still, congratulations for the hard work.

Reviewer #4 (Remarks to the Author): Early Career Researcher co-reviewer

REVIEWERS' COMMENTS

Reviewer #1 (Remarks to the Author):

The authors fully addressed my comments and provided detailed answers to all my queries.

Reviewer #2 (Remarks to the Author):

I applaud the research team for addressing the comments and they have satisfactorily addressed by concerns. I especially like the new data with the R248W mv4;11 clone.

Reviewer #3 (Remarks to the Author):

[This re-review has been completed under a Nature Communication initiative involving one early-career researcher. We both signed at the end to encourage transparency in the peer-review process.]

We are very satisfied with the substantial revisions made to the original draft. The quality of the study has notably increased, with the authors effectively addressing the majority of our comments and concerns. Specifically, the authors have (1) re-analyzed all data within a more robust evolutionary framework, (2) moderated some of their initial claims, and (3) significantly enhanced the methods section by thoroughly detailing the variant calling pipeline and filtering criteria. Furthermore, the completion of additional cell-line experiments has solidified the core argument of their work, further enriching the study's overall impact.

While we have only a few minor questions and observations, mainly seeking clarification on certain points before publication, we have also identified a few typos, which are detailed below. It is important to note that these comments do not detract from the overall quality of the work. In our assessment, this manuscript is ready for publication in this journal, pending the resolution of these small issues.

==Comments==

1. Given the mislabeling of HSPC cells, authors might consider labeling them as t-MN blasts in the phylogenetic trees.
2. Line 292: When describing the results for UPN034, the authors wrote: "Similar to IBFM22, single cells had a high number of T>C mutations, indicating that these mutations are induced by alkylating drugs (signature "SBSH", Extended Data Fig. 6i)." However, there are no single cells in UPN034, and there is no contribution of signature SBSH. Perhaps the authors mean "HSPC

clones” and signature “SBS1”?

3. Lines 367-370: the authors state that all mutations from UPN008 and UPN034 could be attributed to platinum exposure. This statement doesn't align with the data shown in Fig.3a/ and Fig.4c/d where some private and subclonal mutations are, instead, attributed to other signatures. Perhaps the authors meant to say that all clonal mutations from the t-MN blasts were attributed to SBS31. In relation to this, the legend in Fig.4c needs to be adjusted as the authors indicate that the pie charts correspond to the contribution of SBS31. However, we believe the authors meant to say SBS31. Still, we do not understand the logic underlying this paragraph (check the English also): “Interestingly, we found that all mutations in UPN008 and UPN034 could be attributed to platinum-exposure. This is in accordance with previous research that showed there is no correlated between the biological age of the patient and the number of clock-like signatures.”

4. Line 880: “The trees were rooted with treeio’s “root” function”. Please explain where this function comes from.

5. Line 886: “The shape of the trees was validated by the VAF of the tree mutations in the bulk t-MN data.” Perhaps you could explain this a bit better here.

6. The axis labels in Rebuttal Figure 5 have to be wrong, but we believe we understand what you want to show here.

7. We do not completely follow the logic behind removing mutations in terminal branches before extracting mutational signatures, nor do we understand Rebuttal Figure 6b., but we will not insist on this.

8. Please note that in phylogenetics, a bootstrap value of 50 is considered very poor. Depending on who you ask, a “good” bootstrap value starts at 80 or even 90.

Minor typos:

Line 248: replace “blasts or IBFM67” with “blasts of IBFM67”

Line 369: change “there is no correlated between” to “there is no correlation between”

Line 870: replace “RXML-NG” with “RAXML-NG”

Line 873: replace “phenotype likelihood (PL)” with “phred-scaled genotype likelihoods (PL)”

Line 954: change “Only mutations with 100 mutations” to “Only branches with 100 mutations”

Line 957: change “branchers” to “branches”

Congratulations once again on your comprehensive work.

Reviewer #4 (Remarks to the Author):

REVIEWERS' COMMENTS

Reviewer #1 (Remarks to the Author):

The authors fully addressed my comments and provided detailed answers to all my queries.

We thank the reviewer for their time and feedback.

Reviewer #2 (Remarks to the Author):

I applaud the research team for addressing the comments and they have satisfactorily addressed by concerns. I especially like the new data with the R248W mv4;11 clone.

We thank the reviewer for their time and feedback.

Reviewer #3 (Remarks to the Author):

[This re-review has been completed under a Nature Communication initiative involving one early-career researcher. We both signed at the end to encourage transparency in the peer-review process.]

We are very satisfied with the substantial revisions made to the original draft. The quality of the study has notably increased, with the authors effectively addressing the majority of our comments and concerns. Specifically, the authors have (1) re-analyzed all data within a more robust evolutionary framework, (2) moderated some of their initial claims, and (3) significantly enhanced the methods section by thoroughly detailing the variant calling pipeline and filtering criteria. Furthermore, the completion of additional cell-line experiments has solidified the core argument of their work, further enriching the study's overall impact.

While we have only a few minor questions and observations, mainly seeking clarification on certain points before publication, we have also identified a few typos, which are detailed below. It is important to note that these comments do not detract from the overall quality of the work. In our assessment, this manuscript is ready for publication in this journal, pending the resolution of these small issues.

==Comments==

1. Given the mislabeling of HSPC cells, authors might consider labeling them as t-MN blasts in the phylogenetic trees.

We thank the reviewer for this suggestion. We understand that the cells with an HSPC phenotype and leukemic blast genotype can be confusing. However, we do make the distinction between phenotypic blasts and phenotypic HSPCs within this population, also because flowcytometry is an often-used diagnostic technique. To make the distinction clearer, we have now labeled these cells as "HSPC (genetically t-MN)".

2. Line 292: When describing the results for UPN034, the authors wrote: "Similar to IBFM22, single cells had a high number of T>C mutations, indicating that these

mutations are induced by alkylating drugs (signature “SBSH”, Extended Data Fig. 6i).” However, there are no single cells in UPN034, and there is no contribution of signature SBSH. Perhaps the authors mean “HSPC clones” and signature “SBSI”? We thank the reviewers for noticing this mistake, and we have rectified the text as suggested.

3. Lines 367-370: the authors state that all mutations from UPN008 and UPN034 could be attributed to platinum exposure. This statement doesn’t align with the data shown in Fig.3a/ and Fig.4c/d where some private and subclonal mutations are, instead, attributed to other signatures. Perhaps the authors meant to say that all clonal mutations from the t-MN blasts were attributed to SBS31.

We apologize for this unclarity. We indeed meant all clonal mutations from UPN008 and UPN034 and we have now added “clonal” to the sentence in the manuscript.

In relation to this, the legend in Fig.4c needs to be adjusted as the authors indicate that the pie charts correspond to the contribution of SBS31. However, we believe the authors meant to say SBS31.

The reviewers are correct, and we apologize for the mistake in this legend. Now the correct signature is indicated in the legend at both figure 4c and 4e.

Still, we do not understand the logic underlying this paragraph (check the English also): “Interestingly, we found that all mutations in UPN008 and UPN034 could be attributed to platinum-exposure. This is in accordance with previous research that showed there is no correlated between the biological age of the patient and the number of clock-like signatures.”

We have now changed this sentence to more clearly state that all mutations were platinum-related, and thus no clonal age-related mutations were found. “Interestingly, we found that all clonal mutations in UPN008 and UPN034 were platinum-related, and none were age-related. This is in line with previous research showing that there is no link between the biological age and the number of age-related mutations in patients that are treated with platinum, whereas this link is present in cells of healthy individuals.”

4. Line 880: “The trees were rooted with treeio’s “root” function”. Please explain where this function comes from.

We now state in the text that treeio’s root function uses the root function from the ape package.

5. Line 886: “The shape of the trees was validated by the VAF of the tree mutations in the bulk t-MN data.” Perhaps you could explain this a bit better here.

We thank the reviewer for this comment. We have replaced this sentence make the method clearer. It is now as follows: “The shape of the trees was validated by assessing the VAF in the bulk t-MN data of the mutations in the tree. Mutations from clonal branches were all found in the bulk t-MN and had a VAF distribution around 0.5, which is in line with the mutations being present in all t-MN blasts. The majority of the mutations in subclonal branches were found at the bulk t-MN, but at a lower VAF, validating that only a subset of t-MN blasts carried these mutations. Most private t-MN mutations were not found in the bulk t-MN blasts, indicating that they were indeed only present in one or very few blasts, and thus not detectable in bulk. Some mutations were found in the bulk t-MN at low VAF, indicating that these were

actually subclonal mutations, and thus labeled so. Finally, mutations from HSPC private branches were not found in the t-MN bulk.”

6. The axis labels in Rebuttal Figure 5 have to be wrong, but we believe we understand what you want to show here.

Indeed, the x-axis should have read: “fraction of branches with more mutations than cut-off having a reconstructed vs original cosine > 0.85 ”, and the y-axis should have read: “fraction of branches with fewer mutations than cut-off having a reconstructed vs original cosine > 0.85 ”. We apologize for the mistake.

7. We do not completely follow the logic behind removing mutations in terminal branches before extracting mutational signatures, nor do we understand Rebuttal Figure 6b., but we will not insist on this.

We apologize that this point was unclear. In Rebuttal Figure 6a, we tried to show that in all phylogenetic trees many (and in three cases the majority) of mutations that were mapped to end branches (leaves of the tree with 1 individual cell), were often detected in 2 or more cells. We hypothesized that these are likely actually shared mutations, and the mapping to end branches was erroneous. A mutation detected in multiple cells, belonging to an end branch can only be explained if that mutation was truly present in one cell, but a technical artifact in the other cell(s). The artifact rate of PTA is estimated at around 300 per cell, and these artifacts occur throughout the genome. It is thus highly unlikely that these artifacts will so often be shared. In addition, we Rebuttal Figure 6b, we wanted to show that the mutations mapped to end branches that were truly found in one cell, had a different mutational profile (as seen in the heatmap by a low cosine similarity) compared to those mutations that were mapped to end branches, but that were detected in multiple cells. We argue that if the latter were truly belonging to end branches, they should have the same mutational profile. Finally, the mutations mapped to end branches were in almost all cases also mapped to a shared branch in the tree, which we would argue is a more likely scenario. Therefore, we filtered end branches to contain only those mapped mutations that were only detected in one cell.

8. Please note that in phylogenetics, a bootstrap value of 50 is considered very poor. Depending on who you ask, a “good” bootstrap value starts at 80 or even 90.

We thank the reviewers for this comment. We are aware that in some of our phylogenetic trees the bootstrap numbers are below 80. However, we do believe that these lower bootstraps value do not influence any of the conclusions drawn from our study as these splits are not the ones that we draw our main conclusions from, i.e. the splits in the tree that define splits during platinum treatment (with branches with SBS31 present after the split) were all of high confidence.

Minor typos:

Line 248: replace “blasts or IBFM67” with “blasts of IBFM67”

Line 369: change “there is no correlated between” to “there is no correlation between”

Line 870: replace “RXML-NG” with “RAXML-NG”

Line 873: replace “phenotype likelihood (PL)” with “phred-scaled genotype likelihoods (PL)”

Line 954: change “Only mutations with 100 mutations” to “Only branches with 100 mutations”

Line 957: change “branchers” to “branches”

We thank the reviewers for pointing out these typos and have corrected them accordingly.

Congratulations once again on your comprehensive work.